# Unitarity cuts of the worldsheet

**Lorenz Eberhardt⋆ and Sebastian Mizera†**

Institute for Advanced Study, Einstein Drive, Princeton, NJ 08540, USA

⋆ elorenz@ias.edu, † smizera@ias.edu

## Abstract

We compute the imaginary parts of genus-one string scattering amplitudes. Following Witten's $i\varepsilon$ prescription for the integration contour on the moduli space of worldsheets, we give a general algorithm for computing unitarity cuts of the annulus, Möbius strip, and torus topologies exactly in $\alpha'$. With the help of tropical analysis, we show how the intricate pattern of thresholds (normal and anomalous) opening up arises from the worldsheet computation. The result is a manifestly-convergent representation of the imaginary parts of amplitudes, which has the analytic form expected from Cutkosky rules in field theory, but bypasses the need for performing laborious sums over the intermediate states. We use this representation to study various physical aspects of string amplitudes, including their behavior in the $(s, t)$ plane, exponential suppression, decay widths of massive strings, total cross section, and low-energy expansions. We find that planar annulus amplitudes exhibit a version of low-spin dominance: at any finite energy, only a finite number of low partial-wave spins give an appreciable contribution to the imaginary part.



## Contents



# 1 Introduction

A pivotal moment in the development of string theory was the theoretical discovery of the Veneziano amplitude [1]. It gives a concise formula for the tree-level scattering of four massless gluon excitations and can be written as

$$\mathcal{A}_{\text{tree}}(s, t) = t_8 g_s^2 \frac{\Gamma(-\alpha' s)\Gamma(-\alpha' t)}{\Gamma(1 - \alpha' s - \alpha' t)}, \tag{1.1}$$

where $\alpha'$ is the inverse string tension, $g_s$ is the string coupling, $t_8$ carries the information about the polarizations, and $(s, t)$ are the Mandelstam invariants. We also suppressed color structure and normalizations for simplicity.

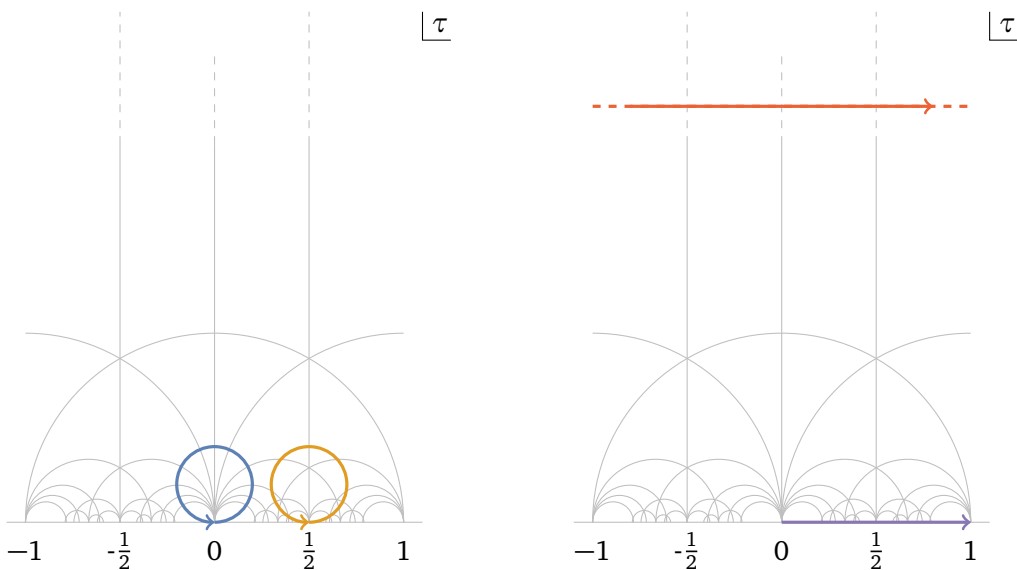

Figure 1: Integration contours in the $\tau$ upper half-plane computing unitarity cuts of genus-one amplitudes. Left: Cuts of planar annulus and Möbius strip topologies come from the blue and orange circles starting and ending at $\tau = 0$ and $\tau = \frac{1}{2}$ respectively. Their radii are irrelevant, but they cannot be shrunken to points because of essential singularities on the real axis. Similar contours exist for the non-planar annulus topology. Right: Cuts of the closed string come from the integration of the analytically-continued modular parameter $\tau = \tau_x + \tau_y$, where $\tau_x = \operatorname{Re}\tau$ runs along the purple contour from 0 to 1, and $\tau_y = i\operatorname{Im}\tau$ along the red contour parallel to the real axis. The vertical displacement of this contour does not matter.

Almost all properties of tree-level scattering are made manifest with this formula. It is crossing-symmetric in $s \leftrightarrow t$. There is an infinite number of integer-spaced resonances in the $s$- and $t$-channels. The coefficient of the residue at $s = n$ tells us about the presence of spin $\leqslant n + 1$ exchanges. In the low energy limit, $\alpha' \to 0$, we recover the super Yang–Mills amplitude $\frac{t_8 g_s^2}{st}$ and its higher-derivative corrections. In the high-energy fixed-angle scattering, $\alpha' \to \infty$, the amplitude is exponentially suppressed (modulo the poles). In the Regge limit, $s \to \infty$ with $t$ fixed, it grows as $s^{1-t}$. Even though the worldsheet no-ghost theorem gives an indirect proof of the unitarity of the amplitude, it is very hard to demonstrate this directly on the level of the amplitude [2].

An extension of the Veneziano formula to loop level remains elusive. But before discussing why, let us stop to ask what makes (1.1) so much more appealing than, say, its worldsheet representation $-\frac{t_8 g_s^2}{\alpha' t} \int_0^1 z^{-\alpha' s - 1}(1-z)^{-\alpha' t} \mathrm{d}z$. This integral does not converge in the physical kinematics, e.g., the $s$-channel with $-s < t < 0$. In order to compute it, one is therefore forced to define it through an analytic continuation of the result evaluated in unphysical kinematics. In any case, what makes (1.1) special is precisely that it can be evaluated without additional hassle: Gamma functions have fast-convergent sum representations (for example the Lanczos approximation [3]), making numerical evaluation straightforward. We will use this criterion of practicality as a guiding principle for searching for a higher-genus generalizations. As a concrete challenge, we will aim to be able to plot the amplitude in physical kinematics.

At the core of the problem is the fact the integration contours for string amplitudes are in general not known. The textbook prescription defining them as integrals over the moduli space of punctured Riemann surfaces (for closed strings) or middle-dimensional contours in those moduli spaces (for open strings) is not entirely well-defined for the same reason as in the

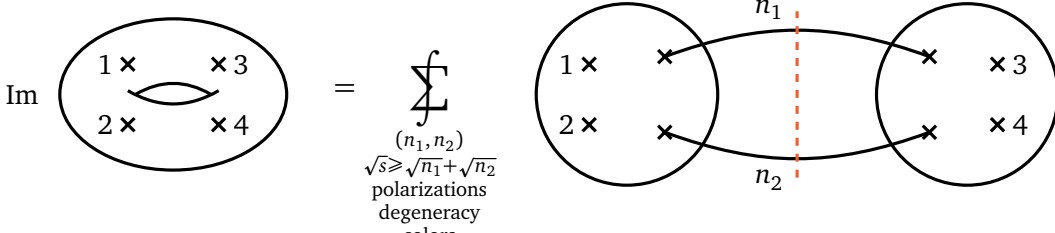

Figure 2: Imaginary parts of genus-one four-point amplitudes in the $s$-channel come entirely from the normal thresholds at $s \geqslant (\sqrt{n_1} + \sqrt{n_2})^2$, where a pair of integers $(n_1, n_2)$ labels the mass squares of the intermediate states. In this work we show how the integration contours from Figure 1, for either open or closed strings, reproduce unitarity cuts without the need for laborious sums over the intermediate states.

Veneziano amplitude: they give rise to divergent amplitudes. For instance, at genus one, these integrals give rise to manifestly real, but divergent results. These cannot be correct since physical loop-level amplitudes need imaginary parts for consistency with unitarity! These difficulties have usually been treated either by regularization of analytic continuation in the literature. The underlying issue turns out to be the fact that Riemann surfaces are Euclidean, which we use to compute observables in the Lorentzian target space (the reason why Euclidean metrics are used in the first place is to manifest the absence of ultraviolet divergences). Pioneering work on analytic continuation of genus-one closed-string amplitudes was undertaken by D'Hoker and Phong [4–6].

At this stage one might reasonably ask why this issue has not caused severe problems before. After all, there is an enormously rich literature on scattering amplitudes in string theory, see, e.g., [7–13] for reviews. However, most of those results have been obtained either at tree-level or in the low-energy expansion at loop-level. Both of these are very forgiving when it comes to analytic continuation. Tree-level amplitudes only feature meromorphic functions of the kinematics (with isolated poles corresponding to propagators going on-shell), while in the low-energy limit the answers can be usually matched with the field-theory intuition for placement of branch cuts. By contrast, in this work we are primarily interested in computing string amplitudes at finite energy, where extreme care is required when it comes to the integration contours, since infinitesimal deformations can lead to being on different sides of branch cuts in the kinematic space.

A prescription for the integration contour on the moduli space was put forward by Witten [14] (see also [15] for earlier work), who pointed out that it should be sufficient to analytically continue worldsheets to Lorentzian signature whenever they develop long tubes and hence are approximated by worldlines. This happens close to the compactification divisors of the moduli space (separating and non-separating). In this way, moduli space contours are consistent with the Feynman $i\varepsilon$ in field theory. We elaborate on this prescription and turn it into a practical algorithm for computing imaginary parts of string amplitudes at genus one. In an upcoming work [16] we will discuss the application of this formalism beyond the imaginary part. The relevant contours in the modular-parameter space are illustrated in Figure 1.

Recall that on-shell poles of string amplitudes are associated with the divisors of the moduli space corresponding to the Riemann surface pinching, or equivalently, developing a long tube looking like a field-theory propagator. Simultaneous poles can occur when the Riemann surface is pinched multiple times. Based on this intuition, one would expect unitarity cuts at genus one to comes from *two* pinches: one for each string state going on-shell, cf. Figure 2. We nevertheless show that they originate from codimension-*one* boundaries (real codimension for open string and complex for closed string) of the moduli space (the non-separating divisor). It

seems to be enough to pinch one cycle of the Riemann surface to force two propagators to go on-shell simultaneously.

We find that string integrands drastically simplify on integration contours in Figure 1, but which specific terms in the integrand survive depends discontinuously on the value of the kinematic invariants $s$ and $t$. This is related to the observation that the Deligne–Mumford compactification [17] does *not* seem to provide a proper model for string amplitudes, because the actual compactification near the moduli space divisors requires more data to describe, including the external kinematics and the remaining moduli. Anyway, to turn this into a systematic procedure, we apply a version of tropical analysis, which automatically tells us which contributions to keep.[1] We show that a new set of terms needs to be taken into account whenever the tropicalization of the string action develops a new saddle point. This happens in the $s$-channel only if

$$\sqrt{s} \geqslant \sqrt{n_1} + \sqrt{n_2}, \tag{1.2}$$

for a pair of non-negative integers $(n_1, n_2)$. The natural physical interpretation is that of a new two-particle threshold being kinematically allowed once the above condition is satisfied, where $\sqrt{n_i}$ are the masses in the string spectrum, see Figure 2. This intuition perfectly matches with the physical interpretation of thresholds as worldline saddle points in field theory [24]. The aforementioned contour therefore implements a cutting procedure for worldsheets (cutting rules in string field theory were recently studied by Pius and Sen [25–28]). We further extend this analysis to anomalous thresholds, or Landau singularities [29–31] in Section 5. The worldsheet cuts are illustrated on many genus-one topologies in type I and type II superstring theory throughout Section 4.

To be more concrete, the chief result of this work is the following manifestly-convergent representation of the imaginary part. Let us quote it in the case of the planar annulus topology in the $s$-channel:

$$\operatorname{Im} A_{\mathrm{an}}^{\mathrm{p}}(s,t) \propto t_8 g_s^4 \frac{\Gamma(1-s)^2}{\sqrt{stu}} \sum_{n_1 \geqslant n_2 \geqslant 0} \theta\big[s - (\sqrt{n_1} + \sqrt{n_2})^2\big] \int_{P_{n_1,n_2} > 0} \mathrm{d}t_{\mathrm{L}}\, \mathrm{d}t_{\mathrm{R}}\, P_{n_1,n_2}(t_{\mathrm{L}}, t_{\mathrm{R}})^{\frac{5}{2}}$$

$$\times Q_{n_1,n_2}(t_{\mathrm{L}}, t_{\mathrm{R}}) \frac{\Gamma(-t_{\mathrm{L}})\Gamma(-t_{\mathrm{R}})}{\Gamma(n_1 + n_2 + 1 - s - t_{\mathrm{L}})\Gamma(n_1 + n_2 + 1 - s - t_{\mathrm{R}})}, \tag{1.3}$$

where we set $\alpha' = 1$ for readability and omitted an inconsequential numerical normalization factor. This formula takes exactly the same form as the result of unitarity cut of any one-loop S-matrix into two tree-level amplitudes exchanging two on-shell particles. Let us explain all the features in turn.

First, the sum is responsible for every two-particle thresholds opening up once the center-of-mass energy $\sqrt{s}$ reaches the value $\sqrt{n_1} + \sqrt{n_2}$ for any pair of two non-positive integers $(n_1, n_2)$, as explained above. Next, we have a two-dimensional integral over the on-shell phase space of the cut, labelled by the momentum transfers $t_{\mathrm{L}}$ and $t_{\mathrm{R}}$ of the two tree-level amplitudes we glue together. It is convolved with the kernel $P_{n_1,n_2}$ which originates as a certain Gram determinant of the internal kinematics. The second line contains all the stringiness of the amplitude. The Gamma functions come essentially from two copies of the appropriately-shifted Veneziano amplitudes. The polynomial $Q_{n_1,n_2}$ summarizes all the intricate information about the coupling to the intermediate states at the levels $(n_1, n_2)$ and we determine it explicitly for the first few values of $(n_1, n_2)$. For example we simply have $Q_{0,0} = 1$. All the integrands are bounded and integrated over a finite region, meaning that they manifestly converge. Finally, we pulled out the polarization dependence $t_8$ and the overall factor $\Gamma(1-s)^2$, which is responsible for

---

[1]Tropical geometry appeared previously in the analysis of the strict $\alpha' \to 0$ limit at higher-genus [18], tree-level amplitudes [19], and loop integrands [20], as well as individual Feynman diagrams [21–23], but here it plays a rather different role since we work at finite $\alpha'$.

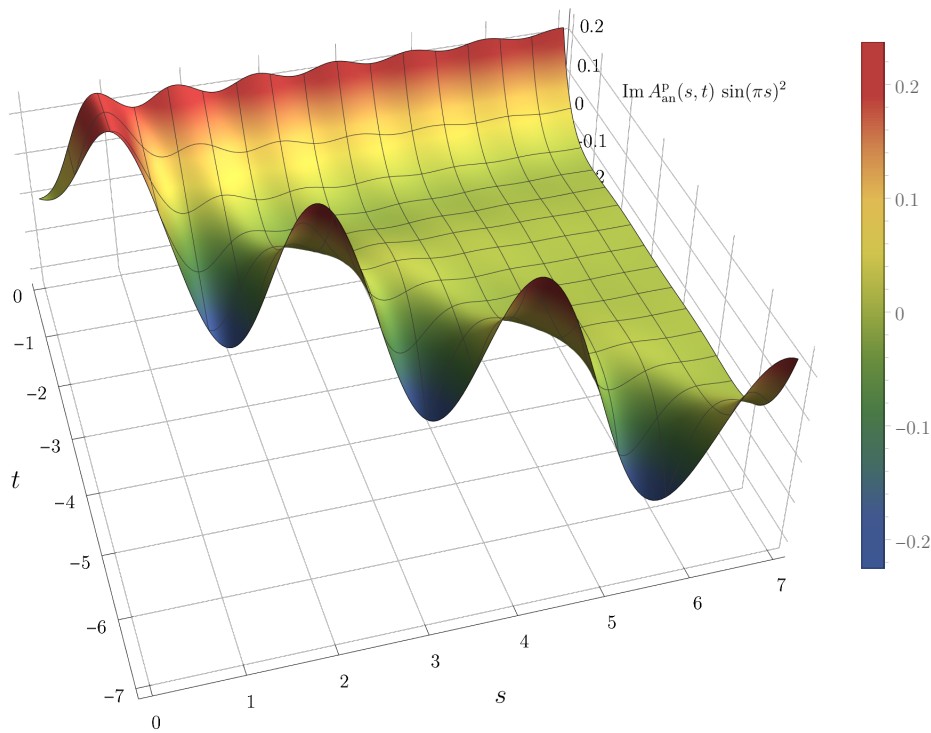

Figure 3: Plot of the imaginary part of the planar annulus amplitude $\mathrm{Im}A_{\mathrm{an}}^{\mathrm{p}}(s,t)$ in the $s$-channel kinematic region ($-s < t < 0$), multiplied by $\sin(\pi s)^2$ in order to remove double poles at $s \in \mathbb{Z}_{>0}$.

double poles at every positive integer $s$ (analogue of bubble diagrams in field theory), which are expected to resum to Breit–Wigner distributions together with higher-genus contributions. This is the only source of divergences of the imaginary part. Contributions from other topologies (Möbius strip, non-planar annulus, and torus) can be incorporated analogously, see Section 4.

At this stage, (1.3) gives us a manifestly-convergent representation of the imaginary part, which can be evaluated with arbitrary accuracy. In Figure 3 we plot it in the $s$-channel kinematic region, where we used the normalization $\sin(\pi s)^2$ in order to remove the double poles.

Let us emphasize that the above computation could have been performed using traditional unitarity methods by gluing together two tree-level amplitudes involving higher and higher mass levels. We illustrate this in Section 3 at the lowest level $(n_1, n_2) = (0,0)$. However, application of this method becomes unfeasible at higher levels because we would have to first enumerate the allowed massive states, construct the vertex operators, and compute their tree-level amplitudes, followed by summing over all contributions and their polarization running along the unitarity cuts. This whole procedure is automatically taken into account from the genus-one worldsheet perspective and amounts to the polynomials $Q_{n_1,n_2}$ in (1.3).

A number of physical properties can be studied using the new representation (1.3). We already mentioned that the only physical region singularities arise from the explicit prefactor $\Gamma(1-s)^2$ coming from the double-pinch degenerations of the worldsheet. As can be read off from Figure 3, and will be illustrated more clearly in Section 6.4, the imaginary part of the

amplitude (modulo the double poles) decays exponentially for larger and larger energies in fixed-angle scattering, consistent with the prediction of [32]. A qualitatively different behavior occurs in the fixed-momentum transfer scattering. The $t \to 0$ limit is non-singular and can be used to compute the planar annulus contribution to the total cross-section, see Section 6.2. We illustrate that for energies up to $s < 7$, it appears to grow linearly with the center-of-mass energy squared $s$, again up to the double-poles. In Section 6.3 we use the coefficients of the double poles at $s = n$ to read-off decay widths of massive string states. At $s = 1, 2$, this computation is exact, because all the level 1 and 2 states are related by supersymmetry, and in the $s > 2$ case we only estimate the decay widths in an "averaged" sense over different species at level $n$. Finally, the $(n_1, n_2) = (0, 0)$ term in the sum (1.3) is enough to reproduce the (asymptotic) $\alpha'$-expansion of the imaginary part of the amplitude. A simple change of variables converts this problem into a straightforward residue computation which can be used to expand it to an arbitrary order in $\alpha'$. We check it against known results of the $\alpha' \to 0$ expansions [33, 34].

Perhaps the most intriguing new feature of the imaginary part of the planar annulus amplitude is its *low-spin dominance*, see Section 6.5. After expressing it in terms of the partial-wave coefficients $\text{Im} f_j(s)$ at spin $j$, we find that the whole imaginary part can be recovered almost entirely from the scalar ($j = 0$) partial-wave coefficient up to $s \lesssim 1$. Following this trend, we find that keeping spins up to $j$ is enough to accurately approximate the imaginary part up to $s \lesssim j+1$. In other words, low spins dominate the imaginary part. We checked up to $s < 1$ that low-spin dominance also occurs for the Möbius strip, non-planar annulus, and torus topologies. A version of low-spin dominance was previously observed at tree-level and in field theory amplitudes, albeit with only spin-0 dominance [35, 36], see also [37].

This paper is organized as follows. In Section 2 we review different degenerations of the worldsheet and how to define the integration contour in their neighborhoods, leading up to the definition of the integration contours computing the imaginary parts of different genus-one topologies. In Section 3 we apply the standard unitarity cut methods to derive a representation of the imaginary parts of amplitudes below the first massive threshold and hint at the generalization to arbitrary masses running through the cuts. In Section 4 we use our proposal for the integration contour in the moduli space to arrive at the same formula for unitarity cuts directly from the worldsheet, and describe an algorithm to compute it for arbitrarily-high energy. In Section 5 we explain how to use tropical analysis to determine discontinuities of genus-one string amplitudes, thus recovering Landau equations known from field theory. In Section 6 we analyze physical properties of the imaginary parts, including its fixed-angle and fixed-momentum transfer behavior, total cross section, decay widths, and the low-spin dominance. We conclude in Section 7 with a discussion and outlook. In Appendix A we give a few longer formulae for the decay widths of string states.

## 2 Overview

Let us start by reviewing some basic facts about string amplitudes. We place particular importance on the choice of integration contour and how to extract the imaginary part of amplitudes.

### 2.1 Amplitudes as integrals over moduli spaces

We will consider perturbative string theory in this paper, which expresses string amplitudes as certain integrals over the moduli space of (super) Riemann surfaces. For simplicity, we will start by discussing the closed bosonic string in flat 26-dimensional space and indicate the necessary changes to other types of string theory below. We will use a mostly plus convention for the

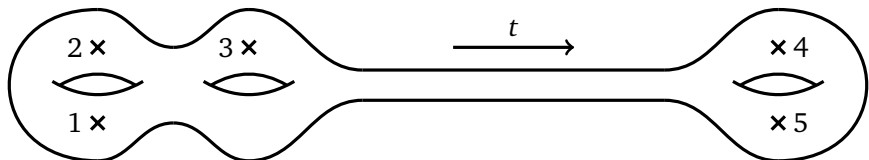

Figure 4: The worldsheet close to a degeneration for a five-point function where a long tube forms. Along the tube, one can consistently define a Lorentzian metric with worldsheet time $t$ along the tube.

metric. Colloquially speaking, an $n$-point tachyon amplitude takes the form

$$\mathcal{A}(p_1, p_2, \ldots, p_n) \sim \sum_{g=0}^{\infty} g_s^{2g+n-2} \int_{\mathcal{M}_{g,n}} \mathcal{I}_g(p_1, p_2, \ldots, p_n). \tag{2.1}$$

We wrote $\sim$ since this equality will be made more precise below. Here $\mathcal{I}_g(p_1, p_2, \ldots, p_n)$ is a certain integrand that can be computed from the worldsheet theory. It takes the general form [38]

$$\mathcal{I}_g(p_1, p_2, \ldots, p_n) = \frac{|\det'(\partial_{bc})|^2}{\det'(\Delta)^{13} (\det \operatorname{Im} \Omega)^{13}} \exp\left( 2 \sum_{1 \leqslant i < j \leqslant n} p_i \cdot p_j \, G(z_i, z_j) \right). \tag{2.2}$$

Here $G(z_i, z_j)$ is the relevant Green's function for the scalar Laplacian $\Delta$ on the Riemann surface under consideration and $|\det'(\partial_{bc})|^2$ is the ghost partition function. Since we removed the ghost zero modes that parametrize moduli, $\det'(\partial_{bc})$ takes values in the canonical bundle of $\mathcal{M}_{g,n}$ and hence $\mathcal{I}_g$ represents a top-form on $\mathcal{M}_{g,n}$ that can be integrated. We refer to [8] for details. We will also use conventions in which mass squares of physical string states are integer, i.e., $\alpha' = 4$ for the closed string and $\alpha' = 1$ for the open string.

The integrals over moduli space are a little bit formal, since they are usually all divergent. At tree-level this is not a big obstacle since the integrals may for example be defined by analytic continuation in the external kinematics. This is clearly unsatisfactory and one would like to give a more direct definition of the amplitudes that does not require one to analytically continue to unphysical kinematics. An explanation and cure for these divergences was given in [14].[2] The basic problem is that perturbative string theory treats the worldsheet as Euclidean, whereas any sensible physical string should treat it as Lorentzian. However, most worldsheet topologies do not admit global Lorentzian metrics and thus there is no such thing as a 'moduli space of Lorentzian surfaces'. The right thing is to notice that all divergences in the integral over $\mathcal{M}_{g,n}$ come from the regions where some part of the worldsheet becomes very long. Since we are talking about conformal structure on the worldsheet, it is equivalent to say that a cycle of the surface is pinching. For a very long tube inside the worldsheet, it is well-defined to choose a Lorentzian metric on this portion on the surface with time going along the tube. This is illustrated in Figure 4.

The length of the tube (compared to its width) is one of the moduli of the surface. Thus if we want to continue the worldsheet integrand from Euclidean signature to Lorentzian signature we should replace the Euclidean length $\tau$ by its Lorentzian counterpart $t$. Concretely this amounts to integrating over $\tau$ up to some cutoff $\tau_*$ and then the contour turns into the complex plane

$$\tau = \tau_* + it, \tag{2.3}$$

with $t \geqslant 0$. Here $\tau$ is defined to be the Schwinger parameter of the corresponding degeneration, which means that $q = e^{-\tau}$ is a well-defined local coordinate for the compactified moduli space

---

[2]See also [15] for an alternative approach at loop level and [39] at tree level.

$\overline{\mathcal{M}}_{g,n}$ in the vicinity of the degeneration in moduli space. Thus (2.1) should be modified to

$$\mathcal{A}(p_1, p_2, \ldots, p_n) = \sum_{g=0}^{\infty} g_s^{2g+n-2} \int_{\Gamma \subset \mathcal{M}_{g,n}^{\mathbb{C}}} \mathcal{I}_g(p_1, p_2, \ldots, p_n), \qquad (2.4)$$

where $\mathcal{M}_{g,n}^{\mathbb{C}}$ is the complexification of $\mathcal{M}_{g,n}$ and $\Gamma$ the contour that we just described in words. Since $\Gamma$ is always close to the real slice $\mathcal{M}_{g,n} \subset \mathcal{M}_{g,n}^{\mathbb{C}}$, we do not need to specify the precise complexification. However it can be explicitly constructed as a cover of two copies of moduli space.

For example, in the tree-level tachyon four-point function, the integrand simply takes the form

$$\mathcal{I}_0 = |z|^{-2s-4}|1-z|^{-2t-4}, \qquad (2.5)$$

with $z \in \mathbb{C}$ and $s = -(p_1+p_2)^2$, $t = -(p_2+p_3)^2$ are the usual Mandelstam variables. Then there are three degenerations corresponding to $z=0$, $z=1$, and $z=\infty$. At $z=0$, it is convenient to parameterize $z = r\,e^{i\phi}$. While $r$ and $\phi$ are initially real, they are allowed to have complex values in the compactification. The local parameter is simply $q = r$ and thus $\tau = -\log|r|$. This means that the relevant integration contour near $z=0$ is given by $r = e^{-\tau_*-it}$, while $\phi$ continues to be integrated from 0 to $2\pi$. The contour in $r$ encircles the origin clockwise.

## 2.2 Separating divisors and on-shell poles

Before moving to the unitarity cuts, we will first discuss the simpler singularities coming from poles in the internal propagators. Consider again Figure 4. From a low-energy perspective, we cannot tell that the long tube is in fact a string as opposed to a field theory propagator. Thus we should expect that it behaves analogously to a field theory propagator. Let us suppose that cutting the tube leads to a disconnected worldsheet as in the above figure. For a genus-$g$ amplitude $\mathcal{A}_g$, the locus in compactified moduli space $\overline{\mathcal{M}}_{g,n}$ where the surface is pinched in known as a separating divisor, since it separates the surface into two components. In particular when we tune $s_{123} = s_{45} \in \mathbb{Z}_{\geqslant -1}$ in the Figure, then there is a corresponding particle in the string spectrum that goes on-shell and leads to a pole in the string amplitude. Thus these regions in moduli space are associated to the on-shell poles in the amplitudes.

This perspective is made manifest in string field theory, where one associates to such a tube a propagator $(L_0 + \bar{L}_0)^{-1}\delta_{L_0,\bar{L}_0}$ (suppressing $b$-ghosts). We can see this also directly from equation (2.2). For a separating degeneration parametrized by a Schwinger parameter $q \to 0$, the worldsheet integrand in the case of tachyon scattering behaves as $q^{-2s_I-3}(1 + \mathcal{O}(q))$, where $s_I$ is the suitable Mandelstam variable.[3] More precisely, a separating divisor separates the vertex operators into two groups with momenta $p_i$, $i \in I$ and the complement. Then $s_I = -\left(\sum_{i \in I} p_i\right)^2$. This behaviour of the integrand follows directly from the corresponding behaviour of the Green's function. As in the tree-level example above, we are hence instructed to integrate over the contour $q = e^{-\tau_*-it}$. Clearly, this integral is oscillatory,

$$-i \int_0^{\infty} dt \; e^{(2s_I+2)(\tau_*+it)}\left(1 + \mathcal{O}(e^{it})\right), \qquad (2.6)$$

but much better convergent than its Euclidean contour part. We could add any small convergence factor and can remove it after the calculation which leads to the expected poles for $s_I \in \mathbb{Z}_{\geqslant -1}$.

---

[3]Often the parameter $q$ is defined to be complex in the literature, even before complexification. In this case, the argument tells us about the twist of the long tube in Figure 4, but we will take to be $q$ real in this paper. We denote the complex version as $\mathfrak{q}$, so that $q = |\mathfrak{q}|$.

One can use this additional structure to give a fully convergent integral representation. We notice that since the monodromy around $q = 0$ just produces the factor $\mathrm{e}^{-4\pi i s_I}$, it suffices to integrate once around the origin if we accompany it with a factor

$$\sum_{n=0}^{\infty} \mathrm{e}^{-4\pi i n s_I} = \frac{1}{1 - \mathrm{e}^{-4\pi i s_I}}. \tag{2.7}$$

Thus one can give a fully compact integration contour near the separating degenerations. At tree level, this approach is well-developed in the literature and is closely related to twisted homology [40–42], see also [43–47] for progress at higher genus.

There are also special separating divisors corresponding to wave-function renormalization and the tadpole. In the former, only one puncture is on one side of the surface. Say this is $p_1$. Then the corresponding Mandelstam variable is automatically on-shell since the external particles are on-shell. Thus the amplitude seemingly sits on the pole. In general, one has to introduce an IR cutoff in the worldsheet integral to deal with these singularities, see, e.g., [48]. However when we look at the superstring, we will study the graviton (or gluon for the open string) amplitude, which is protected against wave-function renormalization thanks to supersymmetry. Thus we will not get into further details about this here. A similar comment applies to the tadpole, where all punctures are on one side of the separating tube. In this case, $s_\emptyset = 0$ and since there are massless particles in the spectrum we are also seemingly sitting on a pole. Again due to supersymmetry, the tadpole cancels in the superstring.

We should also mention that each separating divisor is itself isomorphic to a product of two moduli spaces $\overline{\mathcal{M}}_{h,|I|+1} \times \overline{\mathcal{M}}_{g-h,n-|I|+1}$, where the additional puncture on both sides of the pinched cycle is where the components of the surface are attached. Let us denote by $\mathfrak{q}$ the gluing parameter of the surface, which includes both the twist and the length of the long tube. We have $\mathfrak{q} = q\,\mathrm{e}^{i\theta}$ with $\theta \in [0, 2\pi]$ the twist. Then in terms of $\mathfrak{q}$, the integrand behaves like $|\mathfrak{q}|^{-2s-4}(1 + \mathcal{O}(\mathfrak{q}, \overline{\mathfrak{q}}))$. It then follows from the contour prescription that

$$\operatorname*{Res}_{s_I=n} \mathcal{A}_g(p_1, p_2, \ldots, p_n) = -\pi \int_{\Gamma_1 \times \Gamma_2 \subset \overline{\mathcal{M}}_{h,|I|+1} \times \overline{\mathcal{M}}_{g-h,n-|I|+1}} \operatorname*{Res}_{\mathfrak{q}=0} \operatorname*{Res}_{\overline{\mathfrak{q}}=0} \mathcal{I}(p_1, p_2, \ldots, p_n). \tag{2.8}$$

Taking the double residue produces an integrand on the divisor that is itself isomorphic to the product of two moduli spaces. This property makes factorization of string amplitudes manifest.

The main takeaway of this brief review is that one can be relatively careless about the separating divisors. Since they lead to meromorphic singularities in the string amplitude, it is alright to define them in essentially any way one likes. For this reason we will often ignore the Feynman $i\varepsilon$ prescription that leads to the actual physical spiralling integration contour for the separating divisors and just work with the naive contour where we integrate over the real moduli space.

## 2.3 Non-separating divisor and the imaginary part

The behaviour near the non-separating divisor in moduli space is much more subtle and interesting. A picture of a non-separating divisor at genus 3 is given in Figure 5. For the closed string, there is actually only a single such non-separating divisor in $\overline{\mathcal{M}}_{g,n}$, which itself is isomorphic to $\overline{\mathcal{M}}_{g-1,n+2}/\mathbb{Z}_2$ with the two additional indistinguishable nodes glued together.

As we already discussed, separating divisors lead to on-shell poles in the amplitudes and hence all other types of singularities such as both normal and anomalous thresholds should originate from the non-separating degeneration. In particular, the imaginary part of the amplitude gets picked up entirely at the non-separating divisor. Let us hence discuss in more



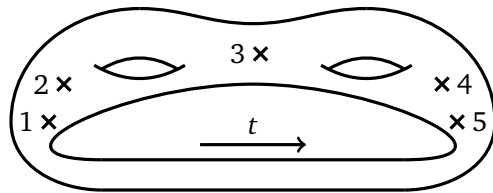

Figure 5: Worldsheet near the non-separating degeneration of a genus-three five-point function.

detail how to extract the imaginary part of the string amplitude from this knowledge.[4] The integration contour in $q$ takes the form that we discussed earlier and is depicted in Figure 6. One can then subtract from it the contour where we apply the $i\varepsilon$-prescription in the opposite way, i.e., the spiralling of the contour is opposite. Due to the reality of the integrand, this computes precisely $2 \operatorname{Im} \mathcal{A}_g$. Turning around the direction of the contour to match the standard counterclockwise direction, we learn that

$$\operatorname{Im} \mathcal{A}_g(p_1, p_2, \ldots, p_n) = -\frac{1}{2} \int_{\circlearrowleft} \mathcal{I}_g(p_1, p_2, \ldots, p_n). \tag{2.9}$$

Here $\circlearrowleft$ stands for the contour in $\mathcal{M}_{g,n}^{\mathbb{C}}$ that encircles $q = 0$ forever, as depicted in the second picture of Figure 6. The contour in all other variables is unchanged.

In field theory, unitarity relates the imaginary part of the amplitude to all possible ways of cutting the Feynman diagram under consideration. For example for a scalar bubble diagram, the Cutkosky cutting rules simply state that

$$\operatorname{Im} \ \text{} \ . \tag{2.10}$$

The definition of the cut, indicated with a dashed line, is that every cut propagator $i/(p^2 + m^2)$ is replaced by $\delta^+(p^2 + m^2) = \delta(p^2 + m^2)\theta(p^0)$, putting the corresponding particle on-shell and imposing positivity of its energy. Thus, the right-hand side reduces to an integral over the remaining loop momentum phase space of the product of two four-point functions. In order for a given cut to contribute to the imaginary part of the amplitude, it has to be possible to simultaneously put all the cut propagators on-shell, which can only happen if the external kinematics satisfies certain conditions. In the bubble diagram example, this is the normal threshold for particle production, $s \geqslant (m_1 + m_2)^2$, where $m_1$ and $m_2$ are the masses of the internal propagators.

In field theory amplitudes, we can also have *anomalous* thresholds or Landau singularities, which arise when we cut the diagram into more than two pieces. For example, one can cut all four propagators of the box diagram

$$\tag{2.11}$$

---

[4] We are ignoring here delta-function contributions to the imaginary part that come from the poles in the amplitude.

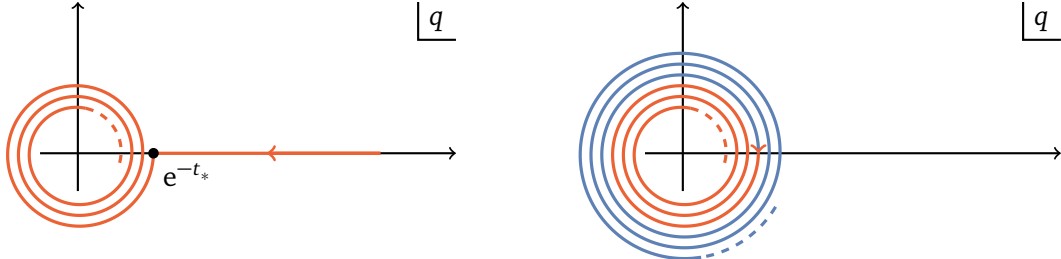

Figure 6: The integration contour in the neighborhood of the non-separating divisor. Only the spiralling part of the contour contributes to the imaginary part of the amplitude. The contour for the imaginary part of the amplitude is given by $\frac{1}{2}$ of the right figure.

which leads to an anomalous threshold at

$$st + 4m^2 u - 4M^4 = 0\,, \tag{2.12}$$

where $M$ is the mass of the external particles (that we assumed to be all equal) and $m$ is the mass of the internal particles. One can show that presence of such cuts is a direct consequence of unitarity, see, e.g., [49]. However, if all the external states are stable against decay, anomalous thresholds do not contribute for physical kinematics. In particular, in the string theory context, we are mostly interested in scattering amplitudes of massless external particles, which are stable. However, the anomalous thresholds can still show up as singularities of the amplitude once we analytically continue it in the $s$- and $t$-variables, for example to the region $s, t > 0$.

This begs the question how (2.9) encodes correctly the cutting rules in string theory. Field theory intuition suggests that (2.9) should be equal to a sum over all possible cuttings of the string diagram. For given Mandelstam variables, there are finitely many such cuttings. For example, in the one-loop four-point function, the only cuts that would contribute in field theory in the $s$-channel are those associated to the production of two intermediate particles, see the earlier Figure 2. For given $s$, there are finitely many string states that can be produced, namely those with $(\sqrt{n_1} + \sqrt{n_2})^2 \leqslant s$ and thus there is still a large number of physical processes contributing to the imaginary part. Note that in Figure 2, it is not necessary to conjugate the part of the amplitude to the right of the cut, precisely because no further propagators can be put on-shell and hence the $i\varepsilon$-prescription (which would have been the only source of an imaginary part) is no longer necessary after taking the normal-threshold cut. We will further elaborate on the generalization of this point beyond four-point amplitudes in Section 7.

The emergence of this structure from (2.9) is clearly quite non-trivial. From the point of view of the conformal structure on the worldsheet, we only put *one* propagator on-shell, namely the one associated to the long tube in Figure 5. However, for the imaginary part our discussion implies that we are automatically putting one more propagator on-shell. The right-hand side of (2.9) does not obviously decompose into a finite sum of products of tree-level amplitudes. The resolution to this is that the integrand $\mathcal{I}_g(p_1, p_2, \ldots, p_n)$ does *not* actually extend to a meromorphic function (or a top form) on the Deligne–Mumford compactification $\overline{\mathcal{M}}_{g,n}$, in which case we would get a single term on the right-hand side.

To explain this a bit more in detail, let us focus on the case of the one-loop amplitude in closed string theory. Denote by $q$ again the local parameter that degenerates at the non-separating degeneration, which is equal to $q = e^{-2\pi \operatorname{Im}\tau}$ in the standard parametrization of the torus moduli space. Then we can expand the integrand $\mathcal{I}_g(p_1, p_2, \ldots, p_n)$ locally in this

parameter, which leads to an equation of the form

$$\operatorname{Im}\mathcal{A}_g(p_1, p_2, \ldots, p_n) = -\frac{1}{2}\sum_T \int_\circlearrowright d\operatorname{Im}\tau \int_{\substack{\text{other} \\ \text{moduli}}} (\operatorname{Im}\tau)^{-\frac{D}{2}} C_T\, q^{\operatorname{Trop}_T}, \qquad (2.13)$$

where $T$ labels different terms in this expansion, and D is the spacetime dimension (26 for the bosonic string and 10 for the superstring). Now, the crucial difference to the separating divisor is that the exponent $\operatorname{Trop}_T$ generally depends on the other moduli (while for separating divisors, the leading exponent was just dependent on the corresponding Mandelstam variable). We call the exponent $\operatorname{Trop}_T$, since it is the appropriate tropicalization of the integrand. $C_T$ is a function that depends on all the moduli other than $q$. There are now finitely many terms in this expansion for which $\operatorname{Trop}_T$ is negative for some choice of moduli. These are the only terms that can contribute to the contour integral since otherwise the integrals decays exponentially if we let the contour wind tighter and tighter around $q = 0$. In the bulk of this paper, we will work this out explicitly for various one-loop amplitudes and show that the different terms indeed correspond to the different unitarity cuts as expected from an effective field theory perspective. We will be able to show in these examples that each term in this expansion can be rewritten in terms of the Baikov representation [50] of cut Feynman diagrams.

## 2.4 Other types of string theories

For other string theories such as the closed or open superstring, the behaviour is very similar with small changes. In an open string, the separating and non-separating divisors are real codimension 1. Thus the integration contour is somewhat easier to describe, since there is no analogue of the twist variable as in eq. (2.8). Correspondingly, one only has to take a single residue of the integrand in (2.8). For open strings, one has to moreover perform an appropriate sum over color indices. The different color structures are precisely reproduced by orientable and unorientable open string surfaces.

Finally for superstrings, one has to deal in general with the fact that the relevant moduli space is really a moduli space of super Riemann surfaces [48, 51].[5] However the fermionic directions of the contour are unaffected by the local modification near the divisors. For example in the case of type II strings, the integration contour is given by

$$\Gamma \subset \mathfrak{M}_{g, n_{\text{NS}}, n_{\text{R}}} \times \mathfrak{M}_{g, n_{\text{NS}}, n_{\text{R}}}. \qquad (2.14)$$

Here $\mathfrak{M}_{g, n_{\text{NS}}, n_{\text{R}}}$ is the moduli space of super Riemann surfaces with $n_{\text{NS}}$ NS-punctures and $n_{\text{R}}$ R-punctures of superdimension $3g - 3 + n_{\text{NS}} + n_{\text{R}} \,|\, 2g - 2 + n_{\text{NS}} + \frac{1}{2}n_{\text{R}}$. The product of two such moduli space corresponds to the complexification $\mathcal{M}_{g,n}^{\mathbb{C}} \cong \mathcal{M}_{g,n} \times \mathcal{M}_{g,n}$ that we considered in the bosonic case.[6] The integration contour in the fermionic directions is full-dimensional and hence one only has to describe the integration contour on the reduced space that is the moduli space of bosonic spin curves, where the contour is completely analogous to the contour that we described before.[7] We refer the reader to [14] for more details.

In the cases studied in this paper, we can get away by talking only about the bosonic moduli, even in the superstring case. For low genera, there are various accidents that allow one to reduce the string integrand from super moduli space to bosonic moduli space by integrating out the fermionic directions and summing over spin structures – in general no natural way

---

[5]Alternatively one needs to insert appropriate PCO's [52, 53]. The two approaches were recently shown to be equivalent [54].

[6]Strictly speaking, the complexification is identified with a cover of the product of two moduli spaces. But since the contour runs close to the diagonal of the two moduli spaces, the distinction does not matter.

[7]This reduction is not canonical and there is no natural description of $\Gamma$, only a natural description up to homology.

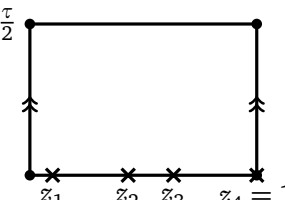 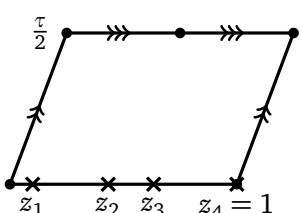 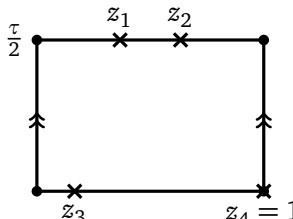

Figure 7: The geometries of the three open string one-loop amplitudes. The first picture is the planar annulus, the second the Möbius strip and the third the non-planar annulus. They are all obtained by orientifolding a torus under the action $z \to \bar{z}$. In the annulus cases the modular parameter is purely imaginary, whereas $\tau \in \frac{1}{2} + i\mathbb{R}$ in the case of the Möbius strip.

for doing so exists [55]. In particular, we will consider one-loop amplitudes for which simple integral formulas exist.[8] For reference, the one-loop four-point functions of graviton scattering in type II [61,62] and gluon scattering in type I string theory [62,63] are

$$\mathcal{A}_{\text{II}} = 2^{-5}\pi^4 g_s^4 t_8 \tilde{t}_8 \int_\Gamma \frac{d^2\tau\, d^2 z_1\, d^2 z_2\, d^2 z_3}{(\operatorname{Im}\tau)^5} \prod_{j<i} |\vartheta_1(z_{ij},\tau)|^{-2s_{ij}} e^{2\pi s_{ij}\frac{(\operatorname{Im} z_{ij})^2}{\operatorname{Im}\tau}} , \tag{2.15a}$$

$$\mathcal{A}_{\text{an}}^{\text{p}} = 2^9 \pi^2 g_s^4 N\, t_8 \operatorname{tr}(t^{a_1} t^{a_2} t^{a_3} t^{a_4}) \frac{(-i)}{32} \int_\Gamma d\tau\, dz_1\, dz_2\, dz_3 \prod_{j<i} \vartheta_1(z_{ij},\tau)^{-s_{ij}} , \tag{2.15b}$$

$$\mathcal{A}_{\text{Möb}} = 2^9 \pi^2 g_s^4\, t_8 \operatorname{tr}(t^{a_1} t^{a_2} t^{a_3} t^{a_4})\, i \int_\Gamma d\tau\, dz_1\, dz_2\, dz_3 \prod_{j<i} \vartheta_1(z_{ij},\tau)^{-s_{ij}} , \tag{2.15c}$$

$$\mathcal{A}_{\text{an}}^{\text{n-p}} = 2^9 \pi^2 g_s^4\, t_8 \operatorname{tr}(t^{a_1} t^{a_2})\operatorname{tr}(t^{a_3} t^{a_4}) \frac{(-i)}{32} \int_\Gamma d\tau\, dz_1\, dz_2\, dz_3 \prod_{j=1}^2 \prod_{i=3}^4 \vartheta_4(z_{ij},\tau)^{-s_{ij}}$$
$$\times \big(\vartheta_1(z_{21},\tau)\vartheta_1(z_{43},\tau)\big)^{-s}. \tag{2.15d}$$

We define the relevant Jacobi theta functions as

$$\vartheta_1(z,\tau) = i\sum_{n\in\mathbb{Z}} (-1)^n e^{2\pi i(n-\frac{1}{2})z + \pi i(n-\frac{1}{2})^2\tau} , \tag{2.16a}$$

$$\vartheta_4(z,\tau) = \sum_{n\in\mathbb{Z}} (-1)^n e^{2\pi i n z + \pi i n^2 \tau} . \tag{2.16b}$$

We are using conventions in which $\alpha' = 1$. Recall that $s_{ij} = -(p_i + p_j)^2 = -2p_i \cdot p_j$ in the massless case, with the short-hands $s = s_{12}$ and $t = s_{23}$. It is understood that the string coupling is the closed/open string coupling in the respective cases. The open string amplitudes have an imaginary prefactor, since $\tau$ runs upwards over the imaginary axis $i\mathbb{R}$ for the annulus and $\tau \in \frac{1}{2} + i\mathbb{R}$ for the Möbius strip (with suitable modifications near the non-separating divisor $\tau = 0$). We also follow the convention to take all $z_i$'s to be real for the open string and $z_i$ in the fundamental domain of the torus in the case of the closed string. The open string geometry in the three cases is pictured in Figure 7. In all cases, the worldsheet without punctures has a translation symmetry that allows us to fix one of the vertex operators arbitrarily and hence the integral only runs over three punctures. We often follow the conventions $z_4 = 1$ for the open string and $z_4 = 0$ for the closed string. In the open string, the vertex operators are ordered and

---

[8]Similar formulas have also been derived for two-loop four-point [56,57] and five-point [58] functions and conjecturally also for three-loop four-point functions [59,60].

hence the integration region in $z_i$ is given by

$$0 \leqslant z_1 \leqslant z_2 \leqslant z_3 \leqslant 1 \tag{2.17}$$

in the planar cases and

$$0 \leqslant z_3 \leqslant 1, \qquad 0 \leqslant z_2 \leqslant 1, \qquad z_2 - 1 \leqslant z_1 \leqslant z_2 \tag{2.18}$$

in the non-planar case. The non-planar annulus diagram is invariant under orientation reversal and thus should be counted in the full amplitude with an additional factor of $\frac{1}{2}$. This factor is not yet included in eq. (2.15d). For the full open string amplitudes, we should also sum over the different color orderings. $N$ denotes the rank of the SO($N$) gauge group and tr the color traces. Both $\mathcal{A}_{\mathrm{an}}^{\mathrm{P}}$ and $\mathcal{A}_{\mathrm{M\ddot{o}b}}^{\mathrm{P}}$ are logarithmically divergent from the region $\tau \to i\infty$. This singularity does not have an analogue in closed string theory and is cancelled by combining the two diagrams. Famously, cancellation happens only when $N = 32$ and thus SO(32) is the only permissible gauge group in open string theory [64]. The factor $t_8$ (or $t_8 \tilde{t}_8$, since the polarization structure double copies in the closed string) contains all the polarizations of the external states. It is discussed further in Section 3, see in particular eq. (3.9) for the definition.

We will denote by $A_{\mathrm{II}}$, $A_{\mathrm{an}}^{\mathrm{P}}$ etc. the amplitudes with the factor $2^{-5}\pi^4 g_s^4 t_8 \tilde{t}_8$ taken out in the closed string and the factor $2^9 \pi^2 g_s^4 t_8$ and the color trace taken out in the open string. The normalizations in the amplitudes will actually be fixed from our analysis. The color traces are taken in the fundamental representation of so($N$).

Let us also note for future reference that the contour for extracting the imaginary part of the amplitude in the case of genus 1 open string amplitudes reads

$$\mathrm{Im}\,\mathcal{A}_{\mathrm{I}} = -\frac{1}{2} \int_{\circlearrowleft} \mathrm{d}\tau \int \mathrm{d}z_1 \,\mathrm{d}z_2 \,\mathrm{d}z_3 \,\mathcal{I}(\tau, z_i). \tag{2.19}$$

Here $\mathcal{I}$ is the corresponding integrand and $\circlearrowleft$ denotes a circular contour that touches the real axis at $\tau = 0$ for the annulus and $\tau = \frac{1}{2}$ for the Möbius strip as in Figure 1. This is the contour described above, where $q = e^{-\frac{2\pi i}{\tau}}$. The contour in the $z_i$'s is the same as for the full amplitude. Similarly, the formula for the closed string amplitude reads

$$\mathrm{Im}\,\mathcal{A}_{\mathrm{II}} = -\frac{1}{2} \int_{\longrightarrow} \mathrm{d}\tau_y \int_0^1 \mathrm{d}\tau_x \int \prod_{i=1}^3 \mathrm{d}x_i \,\mathrm{d}y_i \,\mathcal{I}(\tau_x, \tau_y, x_i, y_i). \tag{2.20}$$

Here we set set $\tau_x = \mathrm{Re}\,\tau$, $\tau_y = i\,\mathrm{Im}\,\tau$, so that $\tau = \tau_x + \tau_y$ and also write $z_i = x_i + \tau y_i$ with $x_i$ and $y_i$ real. Then the original contour in the $z_i$'s get mapped to $0 \leqslant x_i, y_i \leqslant 1$. This parametrization makes the analytic continuation of the integrand convenient. Here $\longrightarrow$ denotes a horizontal contour in the upper half plane in $\tau_y$.

# 3  Imaginary parts from unitarity cuts

In this section we illustrate how to compute the imaginary parts of the one-loop amplitudes at low energies using unitarity cuts. For massless cuts, this is a straightforward computation following standard steps and has been done multiple times in the past (see, e.g., [33,34,65–67]). However, repeating these steps at finite $\alpha'$, where cuts of massive states need to be included, will not going to be feasible, since the string spectrum becomes very complicated, see, e.g., [68]. Therefore, the goal of this section is only to establish the general form of the answer expected from unitarity cuts, which in Section 4 will be reproduced directly from the worldsheet, bypassing the need for summing over the intermediate states.

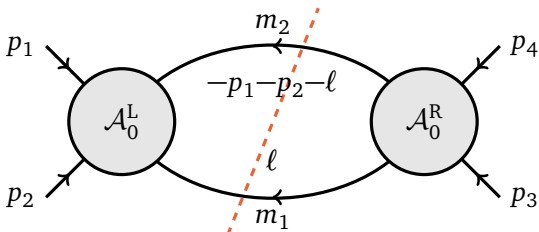

Figure 8: Conventions for the one-loop unitarity cut in the $s$-channel.

At this level, only the elastic amplitudes contribute, i.e., those where a $2 \to 2$ genus-1 amplitude $\mathcal{A}_1$ is obtained by gluing two copies of $2 \to 2$ genus-0 amplitudes $\mathcal{A}_0$, cf. Figure 2. Above the massless threshold but below the first massive one in the $s$-channel, i.e., when $0 < s < 1$ and $t, u < 0$, unitarity instructs us to sum over all ways of exchanging two massless on-shell string states with positive energy. In equations, we have up to a normalization factor:

$$\text{Im}\,\mathcal{A}_1(1234)\big|_{s<1} = \int \mathrm{d}^D\ell \; \delta^-(\ell^2)\,\delta^-((-p_1-p_2-\ell)^2) \sum_{\substack{\text{species}\\\text{colors}\\\text{polarizations}}} \mathcal{A}_0^{\mathrm{L}}(1256)\,\mathcal{A}_0^{\mathrm{R}}(34\overline{56})\,, \qquad (3.1)$$

where the on-shell delta functions are $\delta^\pm(q^2+m^2) = \theta(\pm q^0)\delta(q^2+m^2)$, see, e.g., [69, Section 6-3-4]. We assume that all external states are gluons (for the open string) or gravitons (for the closed string).

Recall that we use the all-incoming conventions, which means we assign the momentum and polarization $(p_i, \epsilon_i)$ to incoming and $(-p_i, \overline{\epsilon}_i)$ to outgoing states, so that the overall momentum conservation reads $\sum_{i=1}^4 p_i = 0$. As we already mentioned, we generally use $\mathcal{A}$ for the full amplitude with the momentum conserving $\delta$-function $-i\delta^D(\sum_{i=1}^4 p_i)$ stripped off. In this notation, the momenta of the particles 5 and 6 in $\mathcal{A}_0^{\mathrm{L}}$ are expressed in terms of the loop momentum $\ell$ as

$$p_5 = \ell\,, \qquad p_6 = -p_1 - p_2 - \ell\,, \qquad (3.2)$$

and their orientations are reversed in $\mathcal{A}_0^{\mathrm{R}}$ (denoted with a bar), see Figure 8. In particular, in those conventions $\delta^-$ imposes the causal energy flow required by unitarity. Note, also, that the factor $\mathcal{A}_0^{\mathrm{R}}$ is already a result of complex conjugation. It actually does not have any effect other than conjugating the polarizations, because $\mathcal{A}_0^{\mathrm{R}}$ does not have any poles or branch cuts within the phase-space we are integrating over, and hence one does not even have to be careful about the $i\varepsilon$ prescription.

The above sums go over all species, colors (for open string), and polarizations of the intermediate string states. Since all the external species are identical, analogous equations in the $t$- and $u$-channels can be obtained simply by relabelling.

**Open string.** In type I superstring theory we only have two possible massless states: gluons $g$ and gluinos $\lambda$, each with 8 polarizations. Since gluinos always come in pairs, the sum in (3.1) boils down to

$$\sum_{\text{colors}} \left( \sum_{\text{pol}} \mathcal{A}_0^{\mathrm{L}}(1_g 2_g 5_g 6_g)\mathcal{A}_0^{\mathrm{R}}(1_g 2_g \overline{5}_g \overline{6}_g) - \sum_{\text{pol}} \mathcal{A}_0^{\mathrm{L}}(1_g 2_g 5_\lambda 6_\lambda)\mathcal{A}_0^{\mathrm{R}}(1_g 2_g \overline{5}_\lambda \overline{6}_\lambda) \right)\,, \qquad (3.3)$$

where $\mathcal{A}_0^{\mathrm{L}}(1_g 2_g 5_{g/\lambda} 6_{g/\lambda})$ are the tree-level amplitudes for scattering of two gluons to two gluons/gluinos, and similarly for $\mathcal{A}_0^{\mathrm{R}}$. The relative minus sign comes about because of the fermion loop. Note that we have absorbed some spinor factors into the definition of the tree

amplitude involving two gluinos so that the cut in (3.1) takes the same form for bosons and fermions.

The evaluation of (3.1) will therefore involve three steps: determining which color structures give rise to which one-loop amplitudes, performing the polarization sums in (3.3), and finally massaging the loop integration to a manifestly on-shell form. We tackle these three problems in turn.

**Closed string.** At the massless level, the spectrum type IIA or IIB string theory is a "double copy" [70] of the open string, for example gravitons, the B-field, and the dilaton can be treated as the symmetric traceless, antisymmetric, and trace parts of the polarization $\epsilon_i^\mu \tilde{\epsilon}_i^\nu$, where $\epsilon_i^\mu$ and $\tilde{\epsilon}_i^\nu$ are two copies of a gluon polarization vector. Because of this, the steps in deriving the imaginary part will be essentially identical to those in the open-string case.

## 3.1 Color sums

We start by determining which tree-level contributions need to be glued together to obtain the planar, non-planar annulus, and Möbius strip topologies. Recall that treating the adjoint color index $a = AB$ as a pair of antisymmetrized fundamental indices, we can write the generators of $so(N)$ as

$$[t^{AB}]_{CD} = -i(\delta^{AC}\delta^{BD} - \delta^{AD}\delta^{BC}), \tag{3.4}$$

where $A, B, C, D = 1, 2, \ldots, N$ etc. It gives rise to the following identity for sums over colors:

$$\sum_a t_{CD}^a t_{EF}^a = -2(\delta^{CE}\delta^{DF} - \delta^{CF}\delta^{DE}). \tag{3.5}$$

The rules for gluing Chan–Paton factors are therefore obtained by all ways of joining the color lines, with a minus sign for every twisting. In equations, we will need the identities

$$\sum_a \mathrm{tr}(At^a)\mathrm{tr}(Bt^a) = 2\left(\mathrm{tr}(AB) - \mathrm{tr}(AB^\mathsf{T})\right), \tag{3.6}$$

and

$$\sum_a \mathrm{tr}(At^aBt^a) = 2\left(\mathrm{tr}(A)\mathrm{tr}(B) - \mathrm{tr}(AB^\mathsf{T})\right), \tag{3.7}$$

where $\mathrm{tr}(\varnothing) = N$. Here, A and B are any strings of generators, i.e., elements of the universal enveloping algebra. Note the because of the asymmetry of the color matrices, we have $(t^{a_1}t^{a_2}\cdots t^{a_m})^\mathsf{T} = (-1)^m t^{a_m}\cdots t^{a_2}t^{a_1}$, which in particular means that every trace of an odd number of colors vanishes.

To simplify the notation, we write the full tree-level amplitudes are given by[9]

$$\mathcal{A}_0^\mathrm{L}(1_g 2_g 5_{g/\lambda} 6_{g/\lambda}) = 4g_s^2 t_8^{b/f}(1256)\Big(\mathrm{tr}(t^{a_1}t^{a_2}t^{a_5}t^{a_6})A_0(s, t_\mathrm{L}) \tag{3.8}$$
$$+ \mathrm{tr}(t^{a_1}t^{a_2}t^{a_6}t^{a_5})A_0(s, u_\mathrm{L}) + \mathrm{tr}(t^{a_1}t^{a_6}t^{a_2}t^{a_5})A_0(t, u_\mathrm{L})\Big),$$

where the information about the external states enters only through the color-independent prefactor $t_8^{b/f}$. In the purely bosonic case, it is the famous $t_8$ tensor:

$$t_8^b(1256) = \mathrm{tr}_v(F_1 F_2 F_5 F_6) + \mathrm{tr}_v(F_1 F_5 F_2 F_6) + \mathrm{tr}_v(F_1 F_2 F_6 F_5) \tag{3.9}$$
$$- \tfrac{1}{4}\Big(\mathrm{tr}_v(F_1 F_2)\,\mathrm{tr}_v(F_5 F_6) + \mathrm{tr}_v(F_1 F_5)\,\mathrm{tr}_v(F_2 F_6) + \mathrm{tr}_v(F_1 F_6)\,\mathrm{tr}_v(F_2 F_5)\Big),$$

---

[9]We follow the normalization conventions of [71], see eq. (12.4.22), although our definition of $t_8^b$ differs by a factor of 2. Here, $g_s$ is the open-string coupling.

where the linearized field strengths are $F_i^{\mu\nu} = p_i^\mu \epsilon_i^\nu - \epsilon_i^\mu p_i^\nu$ and every trace is taken over the Lorentz vector indices. The fermionic counterpart will be spelled out in due time. Finally, each $A_0$ is the scalar Veneziano amplitude:

$$A_0(s, t_L) = \frac{\Gamma(-s)\Gamma(-t_L)}{\Gamma(1-s-t_L)}, \tag{3.10}$$

which is symmetric in $s \leftrightarrow t_L$ and depends on the Mandelstam invariants (recall that we are using a mostly plus convention)

$$s = -(p_1 + p_2)^2, \qquad t_L = -(p_2 + p_5)^2 = -2p_2 \cdot \ell, \qquad u_L = -(p_1 + p_5)^2 = -2p_1 \cdot \ell, \tag{3.11}$$

satisfying $s + t_L + u_L = 0$. The right amplitude $\mathcal{A}_0^R$ is defined by relabelling and reversal of the orientations of the legs 5 and 6, i.e., $t_R = 2p_3 \cdot \ell$ and $u_R = 2p_4 \cdot \ell$.

Because the polarization dependence is factored out in (3.8), the polarization sums can be performed independently of the color sums. Therefore, the factor (3.3) becomes

$$16g_s^4 \mathcal{P} \sum_{a_5, a_6} \left( \text{tr}(t^{a_1} t^{a_2} t^{a_5} t^{a_6}) A_0(s, t_L) + \ldots \right) \left( \text{tr}(t^{a_3} t^{a_4} t^{a_5} t^{a_6}) A_0(s, t_R) + \ldots \right), \tag{3.12}$$

where the polarization dependence $\mathcal{P}$ will be analyzed in the next section. One can then use the color sum identities (3.6) and (3.7). Various contributions can be read-off as coefficients of different trace structures. In addition, we can identify the terms related by relabeling of the intermediate particles $5 \leftrightarrow 6$ (exchanging $(t_L, t_R) \leftrightarrow (u_L, u_R)$) and also the incoming to outgoing particles $12 \leftrightarrow 43$ and simultaneously reversing the loop momentum (exchanging $(t_L, u_L) \leftrightarrow (t_R, u_R)$). The coefficients of $N\text{tr}(t^a t^b t^c t^d)$ are the terms relevant for the planar annulus contributions:

$$128g_s^4 N \mathcal{P} \left[ \text{tr}(t^{a_1} t^{a_2} t^{a_3} t^{a_4}) A_0(s, t_L) A_0(s, t_R) + \text{tr}(t^{a_1} t^{a_2} t^{a_4} t^{a_3}) A_0(s, t_L) A_0(s, u_R) \right]. \tag{3.13}$$

Notice that $\text{tr}(t^{a_1} t^{a_3} t^{a_2} t^{a_4})$ does not have any allowed cuts in the $s$-channel. For the Möbius strip contributions, we read-off coefficients of $\text{tr}(t^a t^b t^c t^d)$ (without the $N$), giving:

$$128g_s^4 \mathcal{P} \left[ 2\text{tr}(t^{a_1} t^{a_2} t^{a_3} t^{a_4}) A_0(s, t_L)(A_0(t_R, u_R) - A_0(s, t_R)) \right.$$
$$+ 2\text{tr}(t^{a_1} t^{a_2} t^{a_4} t^{a_3}) A_0(s, t_L)(A_0(t_R, u_R) - A_0(s, u_R))$$
$$\left. - \text{tr}(t^{a_1} t^{a_3} t^{a_2} t^{a_4}) A_0(t_L, u_L) A_0(t_R, u_R) \right]. \tag{3.14}$$

Finally, the non-planar contributions are of the form $\text{tr}(t^a t^b)\text{tr}(t^c t^d)$ and are given by

$$128g_s^4 \mathcal{P} \left[ \text{tr}(t^{a_1} t^{a_2})\text{tr}(t^{a_3} t^{a_4}) A_0(s, t_L)(A_0(s, t_R) + A_0(s, u_R)) \right.$$
$$\left. + \frac{1}{2}\left( \text{tr}(t^{a_1} t^{a_3})\text{tr}(t^{a_2} t^{a_4}) + \text{tr}(t^{a_1} t^{a_4})\text{tr}(t^{a_2} t^{a_3}) \right) A_0(t_L, u_L) A_0(t_R, u_R) \right]. \tag{3.15}$$

All contributions are displayed graphically in Figure 9. For concreteness, below we will first treat the case of the annulus $N\text{tr}(t^{a_1} t^{a_2} t^{a_3} t^{a_4})$ contribution (3.13) step-by-step and then give the final answers in the remaining cases. In this case, we have

$$\text{Im}A_{an}^p \big|_{s<1} = \frac{s^2}{8\pi^2} \int d^D\ell \, \delta^-(\ell^2) \delta^-((-p_1 - p_2 - \ell)^2) A_0(s, t_L) A_0(s, t_R). \tag{3.16}$$

We defined a straight $A$ that has the polarizations, color traces and coupling constants as well as various constant factors stripped off. This convention will turn out to match the one on the

Planar annulus

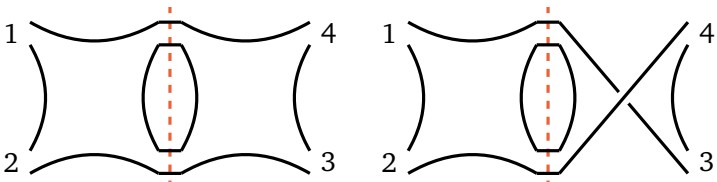

Möbius strip

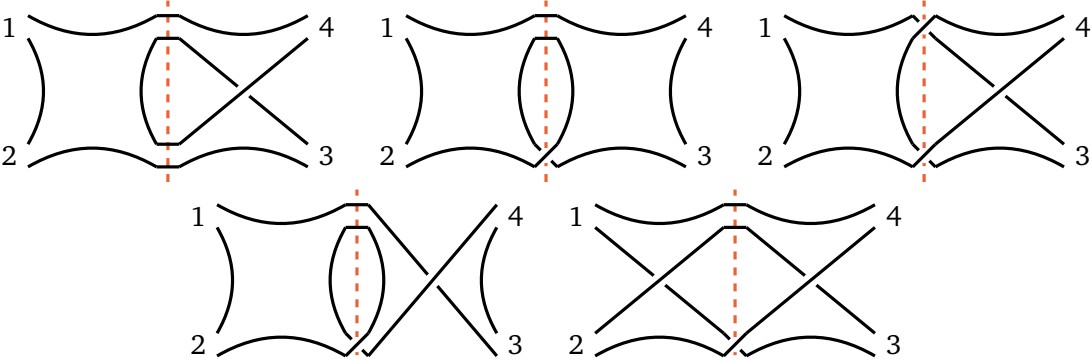

Non-planar annulus

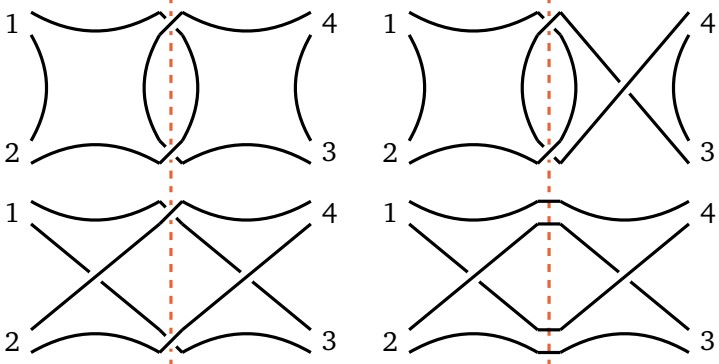

Figure 9: Different possibilities for the color contractions of two tree-level four-point functions in the $s$-channel, corresponding to the terms in the planar annulus (3.13), Möbius strip (3.14), and non-planar annulus (3.15) contributions, before using symmetry relations.

worldsheet integrals with all prefactors stripped off. We will compute below that $\mathcal{P} = \frac{s^2}{2} t_8$ and hence

$$\text{Im}\,\mathcal{A}_{\text{an}}^{\text{p}} = 2^9 \pi^2 g_s^4 N t_8 \text{tr}(t^{a_1} t^{a_2} t^{a_3} t^{a_4}) \text{Im}\,A_{\text{an}}^{\text{p}} + 2 \text{ other color orderings}. \tag{3.17}$$

It will be actually slightly more illuminating to restate the results as always computing the same trace structure, but in different kinematic channels, which can be always obtained by rebelling of the external momenta. For example, the $s$-channel cut of the Möbius strip with ordering (1324) is the same as the $u$-channel cut of (1234) after relabelling.

## 3.2   Polarization sums

In this section we compute the polarization sums. We first treat the open string case and later compute the closed case by double copy.

**Open string.**   Since all the polarization dependence is enclosed in the $t_8^{b/f}$ tensors, here we will be interested in evaluating the sum

$$\mathcal{P} = \sum_{\text{pol}} \left[ t_8^b(1256)\, t_8^b(34\overline{56}) - t_8^f(1256)\, t_8^f(34\overline{56}) \right], \tag{3.18}$$

where the relative minus sign arises because of the fermion loop. Recall that the two particles we cut have the momenta $p_5 = \ell$ and $p_6 = -(p_1 + p_2 + \ell)$ respectively, and the bar represents reversing the polarizations and momenta of the particles to match our all-incoming conventions.

Anticipating a large amount of cancellations between bosons and fermions due to supersymmetry, the first goal is to bring the expressions for $t_8^{b/f}$ to a similar form manifesting these cancellations. Following the literature on scattering equations [72, 73], we can rewrite these functions in the following way:

$$t_8^{b/f}(1256) = \sum_{A \in \{\varnothing, 1, 2, 12, 21\}} c_{5A6}\, W_{5A6}^{b/f}, \tag{3.19}$$

where the dependence on the polarizations of the special particles 5 and 6 enters only through the combinations:

$$W_{5A6}^b = \epsilon_5 \cdot F_{A_1} \cdot F_{A_2} \cdots F_{A_{|A|}} \cdot \epsilon_6, \tag{3.20a}$$

$$W_{5A6}^f = \tfrac{1}{\sqrt{2}} \chi_5 F\!\!\!/_{A_1} F\!\!\!/_{A_2} \cdots F\!\!\!/_{A_{|A|}} \chi_6, \tag{3.20b}$$

with the same coefficients $c_A$ for the bosonic and fermionic versions. We use $\chi_i$ to denote appropriately-normalized Weyl spinors of the particles and 5 and 6, as well as the conventions in which the bosonic/fermionic linearized field strengths are given by

$$F_i^{\mu\nu} = p_i^\mu \epsilon_i^\nu - \epsilon_i^\mu p_i^\nu, \qquad F\!\!\!/_i = \tfrac{1}{8} F_i^{\mu\nu}[\gamma_\mu, \gamma_\nu] = \tfrac{1}{2} p\!\!\!/_i \epsilon\!\!\!/_i. \tag{3.21}$$

In (3.20a) and (3.20b), the contractions are performed by Lorentz and spinor indices respectively. It will turn out that all the factors except $c_{5126}$ and $c_{5216}$ will drop out. However, let us list them for completeness

$$c_{56} = 2\epsilon_{1,\mu}\epsilon_{2,\nu}[-p_3^\mu p_1^\nu p_2\cdot p_5 - p_2^\mu p_3^\nu p_1\cdot p_5 + p_3^\mu p_3^\nu p_1\cdot p_2 + \eta^{\mu\nu} p_1\cdot p_5\, p_2\cdot p_5], \tag{3.22a}$$

$$c_{516} = 2\epsilon_{2,\mu}[-p_5^\mu p_1\cdot p_2 + p_1^\mu p_2\cdot p_5], \qquad c_{526} = 2\epsilon_{1,\mu}[-p_3^\mu p_1\cdot p_2 + p_2^\mu p_1\cdot p_5], \tag{3.22b}$$

$$c_{5126} = -2 p_2\cdot p_5, \qquad\qquad\qquad c_{5216} = -2 p_1\cdot p_5. \tag{3.22c}$$

Notice that gauge each term is gauge-invariant on its own.

At this stage $\mathcal{P}$ is a bilinear in the $W$'s. Since the individual $t_8$'s are already gauge-invariant, there is no need to project out the unphysical components, and the polarization sums simply amount to making the replacements:

$$\sum_{\text{pol}} \epsilon_j^\mu \overline{\epsilon}_j^\nu \to \eta^{\mu\nu}, \qquad \sum_{\text{pol}} \chi_j^\alpha \overline{\chi}_j^\beta \to \delta^{\alpha\beta}, \tag{3.23}$$

for $j = 5, 6$. Note that the above normalization of spinors $\chi_i$ is such that fermionic unitarity cuts can be treated at the same footing as the bosonic ones, $\delta^+(p_i^2)$. In the bosonic case, this leads to

$$\sum_{\text{pol}} W_{5A6}^b \overline{W}_{5B6}^b = (-1)^{|B|} \operatorname{tr}_\nu(F_{A_1} F_{A_2} \cdots F_{A_{|A|}} F_{B_{|B|}} \cdots F_{B_2} F_{B_1}), \tag{3.24}$$

where the bar on the left-hand side denotes the fact that $\epsilon$'s of the loop momenta are complex-conjugated. Note that the order of $F$'s in the set $B$ is transposed and the trace is taken over the Lorentz indices, which we indicate with $\mathrm{tr}_\nu$. Similarly, the sums over fermions give

$$\sum_{\text{pol}} W^f_{A56} \overline{W}^f_{B56} = \tfrac{1}{2}(-1)^{|B|} \mathrm{tr}_s(\slashed{F}_{A_1}\slashed{F}_{A_2}\cdots\slashed{F}_{A_{|A|}}\slashed{F}_{B_{|B|}}\cdots\slashed{F}_{B_2}\slashed{F}_{B_1}). \tag{3.25}$$

Here, the trace is taken over the spinor indices, indicated with $\mathrm{tr}_s$.

In the special case when both $A$ and $B$ are empty, we have

$$\mathrm{tr}_\nu(\varnothing) = D - 2, \qquad \mathrm{tr}_s(\varnothing) = 2^{(D-2)/2}. \tag{3.26}$$

Here we take the spacetime dimension D as a variable (with even D to have Weyl spinors) to illustrate how the cancellations happen in D $= 10$. In addition, by antisymmetry we have

$$\mathrm{tr}_s(\slashed{F}_i) = \mathrm{tr}_\nu(F_i) = 0. \tag{3.27}$$

Using gamma-matrix algebra, one can further expand every spinor trace $\mathrm{tr}_s(\cdots)$ in terms of combinations of $\mathrm{tr}_\nu(\cdots)$ (see [73, App. C] for a general formula), with the next two cases giving

$$\mathrm{tr}_s(\slashed{F}_i\slashed{F}_j) = 2^{(D-8)/2}\,\mathrm{tr}_\nu(F_iF_j), \tag{3.28}$$

$$\mathrm{tr}_s(\slashed{F}_i\slashed{F}_j\slashed{F}_k) = 2^{(D-8)/2}\,\mathrm{tr}_\nu(F_iF_jF_k). \tag{3.29}$$

Finally, in the most complicated case we have

$$\mathrm{tr}_s(\slashed{F}_i\slashed{F}_j\slashed{F}_k\slashed{F}_l) = 2^{(D-10)/2}\Big(\mathrm{tr}_\nu(F_iF_jF_kF_l) - \mathrm{tr}_\nu(F_iF_kF_jF_l) - \mathrm{tr}_\nu(F_iF_jF_lF_k)\Big)$$
$$+ 2^{(D-14)/2}\Big(\mathrm{tr}_\nu(F_iF_j)\mathrm{tr}_\nu(F_kF_l) + \mathrm{tr}_\nu(F_iF_k)\mathrm{tr}_\nu(F_jF_l) + \mathrm{tr}_\nu(F_iF_l)\mathrm{tr}_\nu(F_jF_k)\Big). \tag{3.30}$$

Note that the first trace has a sign opposite to the following two.

We see that in D $= 10$, all the terms where the total length of $A$ and $B$ is less than four cancel out between bosons and fermions, and we are only left with those where both $A$ and $B$ has two elements each. More specifically,

$$\mathcal{P} = \sum_{\substack{A \in \{12,21\} \\ B \in \{34,43\}}} c_{5A6}\,\overline{c_{5B6}}\Big(\underbrace{\mathrm{tr}_\nu(F_{A_1}F_{A_2}F_{B_2}F_{B_1}) - \tfrac{1}{2}\mathrm{tr}_s(\slashed{F}_{A_1}\slashed{F}_{A_2}\slashed{F}_{B_2}\slashed{F}_{B_1})}_{\tfrac{1}{2}t_8(1234)}\Big). \tag{3.31}$$

Finally, one notices that the above traces organize themselves into the $t_8$ tensor of the external kinematics $t_8(1234)$ for any $A$ and $B$. We are thus left with

$$\mathcal{P} = \tfrac{1}{2}\underbrace{(c_{5126} + c_{5216})}_{-s}\underbrace{(c_{5346} + c_{5436})}_{-s}t_8(1234) = \frac{s^2}{2}t_8(1234). \tag{3.32}$$

**Closed string.** The closed-string case can be treated entirely analogously. The sums over species and polarizations in (3.1) amount to

$$\sum_{\text{pol}}\Big[t^b_8(1256)\,t^b_8(34\overline{56}) - t^f_8(1256)\,t^f_8(34\overline{56})\Big] \tag{3.33}$$

$$\times \Big[\tilde{t}^b_8(1256)\,\tilde{t}^b_8(34\overline{56}) - \tilde{t}^f_8(1256)\,\tilde{t}^f_8(34\overline{56})\Big], \tag{3.34}$$

where $\tilde{t}^{b/f}_8$ are obtained from $t^{b/f}_8$ by replacing all the polarizations $\epsilon_i \to \tilde{\epsilon}_i$. We also define[10]

$$\mathcal{A}_0(s,t,u) = \pi^2 g^2_s t^b_8 \tilde{t}^b_8 A_0(s,t,u), \tag{3.35}$$

---

[10]This corresponds again to the convention used in [71] with $\alpha' = 4$ so that the closed string spectrum is integer spaced. Here $g_s$ denotes the closed string coupling constant.

with

$$A_0(s,t,u) = \frac{\Gamma(-s)\Gamma(-t)\Gamma(-u)}{\Gamma(1+s)\Gamma(1+t)\Gamma(1+u)} = \frac{1}{\pi}\frac{\sin(\pi s)\sin(\pi t)}{\sin(\pi(s+t))}A_0(s,t)^2. \tag{3.36}$$

The sums over the tilded and untilded polarizations can be performed independently. Recognizing that (3.33) involves two copies of (3.18), it equals

$$\pi^4 g_s^4 \mathcal{P}\tilde{\mathcal{P}}A_0(s,t_L,u_L)A_0(s,t_R,u_R). \tag{3.37}$$

Finally using (3.31), the formula for the imaginary part of the closed-string amplitude is given by

$$\text{Im}\,\mathcal{A}_{\text{II}}\big|_{s<1} = \frac{\pi^4 s^4 g_s^4}{4}t_8\tilde{t}_8 \int d^D\ell\,\delta^-(\ell^2)\,\delta^-((-p_1-p_2-\ell)^2)A_0(s,t_L,u_L)A_0(s,t_R,u_R). \tag{3.38}$$

We define

$$\mathcal{A}_{\text{II}} = 2^{-5}\pi^4 t_8\tilde{t}_8 A_{\text{II}}, \tag{3.39}$$

for convenience. This is again consistent with the normalization of the worldsheet integrals.

## 3.3 Loop integration

The loop integration in (3.1) already uses only the on-shell data, but not manifestly so. The goal of this subsection is to express it purely on-shell. We have that $s = s_L = s_R$ by momentum conservation. The integral over the on-shell phase space thus has to amount to convolving the on-shell amplitudes $\mathcal{A}_0^L(t_L)$ and $\mathcal{A}_0^R(t_R)$ over the physically-allowed range of $t_L$ and $t_R$'s:

$$\text{Im}\,\mathcal{A}_1(s,t) = \int dt_L\,dt_R\,\mathcal{K}(t,t_L,t_R)\,\mathcal{A}_0^L(s,t_L)\,\mathcal{A}_0^R(s,t_R), \tag{3.40}$$

for some kernel $\mathcal{K}(t,t_L,t_R)$. From this perspective, the goal is to find an explicit formula for $\mathcal{K}$. Note that at this stage we are making a tacit assumption that within $\mathcal{A}_0^L$ and $\mathcal{A}_0^R$ the dependence on the loop momentum enters only through the invariants $t_L$ and $t_R$, which we verified above to be the case. To illustrate the logic, we first treat the simplest case of the forward limit, $t = 0$, before moving on to the most general case.

### 3.3.1 Forward limit

The forward limit can be realized by going to the Lorentz frame in which $p_3 = -p_2$ and $p_4 = -p_2$, giving the momentum transfer squared $t = -(p_2 + p_3)^2 = 0$. The problem then depends only on four independent kinematic invariants: $\ell^2$, $\ell \cdot p_1$, $\ell \cdot p_2$, and $p_1 \cdot p_2$. They are related to the on-shell data of the constituent amplitudes by

$$s = -2p_1 \cdot p_2, \qquad t_L = -(p_2 + \ell)^2 = -\ell^2 - 2\ell \cdot p_2, \qquad t_R = -(p_3 - \ell)^2 = t_L. \tag{3.41}$$

We thus expect to arrive at an expression involving a single integration over $t_L = t_R$ in the physical region $-s < t_L < 0$.

As the first step, let us decompose the D-dimensional loop momentum $\ell = \ell_\parallel + \ell_\perp$ into two components $\ell_\parallel$ in the span of the external momenta $p_1$, $p_2$ and the remaining D−2 orthogonal components $\ell_\perp$. In other words, $p_i \cdot \ell = p_i \cdot \ell_\parallel$, which allows us to trade the integration over $\ell_\parallel$ into that over $\ell \cdot p_1$ and $\ell \cdot p_2$ at a cost of the Jacobian $1/\sqrt{|\det \mathcal{G}_{p_1 p_2}|} = 2/s$ which is given by the determinant of the Gram matrix

$$\mathcal{G}_{q_1 q_2 \ldots q_m} = [-q_i \cdot q_j]_{i,j=1,2,\ldots,m}. \tag{3.42}$$

Moreover, the vector $\ell_\perp$ only ever appears in the combination $\ell_\perp^2$, which means we can integrate out all its angular components, giving $\pi^{(D-2)/2}/\Gamma(\frac{D-2}{2})$, and be left with only the integral over $\ell_\perp^2$. These manipulations result in the loop-momentum measure being expressed as

$$\int d^D\ell = \int d^2\ell_\| \, d^{D-2}\ell_\perp = \frac{2\pi^{(D-2)/2}}{s\Gamma(\frac{D-2}{2})} \int_{\ell_\perp^2 > 0} d(\ell \cdot p_1) \, d(\ell \cdot p_2) \, d(\ell_\perp^2)(\ell_\perp^2)^{(D-4)/2} , \qquad (3.43)$$

where the only constraint on the integration domain comes from the fact that the radius $\ell_\perp^2$ needs to be positive (because $\ell_\perp$ is orthogonal to $p_i$ and we are using a mostly plus convention for the metric). Let us finally notice that

$$\ell_\perp^2 = \ell^2 - \ell_\|^2 = \ell^2 - \sum_{i,j=1}^{2} \ell \cdot p_i \, (\mathcal{G}_{p_1 p_2}^{-1})_{ij} \, p_j \cdot \ell = \ell^2 + \tfrac{4}{s}\ell \cdot p_1 \, \ell \cdot p_2 , \qquad (3.44)$$

and hence we can trade the final integration for that over $\ell^2$ directly with unit Jacobian. The measure thus becomes

$$\int d^D\ell = \frac{2\pi^{(D-2)/2}}{s\Gamma(\frac{D-2}{2})} \int_{\ell_\perp^2 > 0} d(\ell \cdot p_1) \, d(\ell \cdot p_2) \, d(\ell^2)\big(\ell^2 + \tfrac{4}{s}\ell \cdot p_1 \, \ell \cdot p_2\big)^{(D-4)/2} . \qquad (3.45)$$

At this stage, we can use the cut conditions to localize two out of the three integrations.

Before doing so, it is important to carefully treat the negative-energy condition entering through the $\delta^-$ constraints in (3.1). Notice that the kinematics of the left amplitude $\mathcal{A}_0^{\mathrm{L}}$ is essentially that of a single massive particle with momentum $p_1 + p_2$ scattering into two massless particles. If we flipped the signs of the energies of the massless particles, it would correspond to either to a production process of a massless particle (flipping a single sign) or a production of the vacuum (flipping two signs), neither of which is allowed kinematically. Analogous comments hold for $\mathcal{A}_0^{\mathrm{R}}$. We can thus safely replace $\delta^- \to \delta$ in (3.1), since the energy conditions are automatically imposed by the external kinematics constraints. Note that this would not be true for inelastic unitarity terms, e.g., gluing $2 \to 3$ and $3 \to 2$ genus-0 contributions to obtain the imaginary part of $2 \to 2$ genus 2 scattering amplitude.

With these points out of the way, we are left with the ordinary residue integrals imposing the on-shell conditions:

$$\ell^2 = 0, \qquad (-p_1 - p_2 - \ell)^2 = -s + 2\ell \cdot (p_1 + p_2) = 0 . \qquad (3.46)$$

We use them to localize the $\ell^2$ and $\ell \cdot p_1$ integrals, with the overall Jacobian $\frac{1}{2}$. On their support, we can also write

$$\ell \cdot p_1 = \tfrac{1}{2}(s + t_{\mathrm{L}}), \qquad \ell \cdot p_2 = -\tfrac{1}{2}t_{\mathrm{L}} , \qquad (3.47)$$

using (3.41). After the changing the remaining integration into that over $t_{\mathrm{L}}$, we have

$$\int d^D\ell \, \delta^-(\ell^2)\delta^-((-p_1 - p_2 - \ell)^2)(\cdots) = \frac{\pi^{\frac{D-2}{2}}}{2s\Gamma(\frac{D-2}{2})} \int_{-s}^{0} dt_{\mathrm{L}}\left(-\frac{t_{\mathrm{L}}(s + t_{\mathrm{L}})}{s}\right)^{(D-4)/2}(\cdots) , \qquad (3.48)$$

and the integration bounds are equivalent to $\ell_\perp^2 > 0$. Finally, to make the $s$-dependence more obvious, let us change the variables to

$$x = 1 + \frac{2t_{\mathrm{L}}}{s} \in [-1, 1], \qquad (3.49)$$

which is the cosine of the scattering angle, yielding

$$\int d^D \ell \, \delta^-(\ell^2) \, \delta^-((-p_1-p_2-\ell)^2) A_0^L\left(s, t_L = \tfrac{s}{2}(x-1)\right) A_0^R\left(s, t_R = \tfrac{s}{2}(x-1)\right)$$

$$= c_D s^{\frac{D-4}{2}} \int_{-1}^{1} dx \, (1-x^2)^{\frac{D-4}{2}} A_0(s, \tfrac{s}{2}(x-1))^2, \quad (3.50)$$

where $c_D = 2^{2-D} \pi^{\frac{D-2}{2}} / \Gamma(\frac{D-2}{2})$. In particular, for the planar annulus amplitude in $D = 10$ we obtain the following result

$$\text{Im} A_{\text{an}}^{\text{p}}\big|_{s<1}^{t=0} = \frac{\pi^2 s^5}{3 \cdot 2^{12}} \int_{-1}^{1} dx \, (1-x^2)^3 \left(\frac{\Gamma(-s)\Gamma(-\tfrac{s}{2}(x-1))}{\Gamma(1-\tfrac{s}{2}(x+1))}\right)^2, \quad (3.51)$$

where we took into account the factor $\frac{s^2}{8\pi^2}$ that appears in eq. (3.16). Let us expand this result in $\alpha'$. The $\alpha'$ dependence can easily be reinstated by dimensional analysis. The expansion around $s = 0$ starts at order $\mathcal{O}(s)$:

$$\text{Im} A_{\text{an}}^{\text{p}}\big|_{s<1}^{t=0} = \frac{\pi^2 s}{1920} \left(1 + \tfrac{2}{3}\zeta_2 s^2 + \tfrac{8}{21}\zeta_3 s^3 + \tfrac{23}{21}\zeta_4 s^4 + (\tfrac{2}{7}\zeta_2\zeta_3 + \tfrac{19}{63}\zeta_5)s^5 + (\tfrac{11}{9}\zeta_6 + \tfrac{5}{63}\zeta_3^2)s^6 + \dots\right),$$
$$(3.52)$$

where $\zeta_k$ is the $k$-th multiple zeta value (MZV), e.g., $\zeta_2 = \frac{\pi^2}{6}$. We note that the result is uniformly-transcendental, i.e., coefficients of $s^k$ are accompanied with MZVs of total weight $k$. Likewise, around $s = 1$ we have the expansion

$$\text{Im} A_{\text{an}}^{\text{p}}\big|_{s<1}^{t=0} = \frac{\pi^2}{13440(s-1)^2} \left(1 + (2\gamma_E + 560\zeta'_{-3} + 1680\zeta'_{-5} - \tfrac{11}{3})(s-1) \right. \quad (3.53)$$

$$\left. + (1.70129\dots)(s-1)^2 + (3.59874\dots)(s-1)^3 + \dots\right),$$

where $\gamma_E$ is the Euler's constant. Recall that both results are only valid in $0 < s < 1$.

### 3.3.2 General kinematics

The elastic unitarity equation with arbitrary kinematics can be computed in an entirely analogous way. We start with the general expression

$$\text{Im} A_{\text{an}}^{\text{p}}\big|_{s<1} = \frac{s^2}{8\pi^2} \int d^D \ell \, \delta^-(\ell^2) \, \delta^-((-p_1-p_2-\ell)^2) A_0(s, t_L) A_0(s, t_R), \quad (3.54)$$

where, in our conventions, the kinematic invariants are given by

$$s = -(p_1 + p_2)^2, \qquad t_L = -(p_2 + \ell)^2, \qquad t_R = -(p_3 - \ell)^2. \quad (3.55)$$

As before, we first split the loop integration into the 3-dimensional components $\ell_{\parallel}$ in the span of the external momenta $p_1$, $p_2$, $p_3$ (here we assume $D > 3$), and the $(D-3)$-dimensional orthogonal complement. The former gives

$$\int d^3 \ell_{\parallel} = |\det \mathcal{G}_{p_1 p_2 p_3}|^{-1/2} \int \prod_{i=1}^{3} d(\ell \cdot p_i), \quad (3.56)$$

with the Jacobian involving

$$|\det \mathcal{G}_{p_1 p_2 p_3}| = \tfrac{1}{4} stu. \quad (3.57)$$

In the latter case, most of the $\ell_\perp$ directions can be integrated out, except for the modulus $\ell_\perp^2$, yielding

$$\int d^{D-3}\ell_\perp = \frac{\pi^{(D-3)/2}}{\Gamma(\frac{D-3}{2})} \int_{\ell_\perp^2>0} d(\ell_\perp^2)(\ell_\perp^2)^{\frac{D-5}{2}}, \tag{3.58}$$

where

$$\ell_\perp^2 = -\frac{\det \mathcal{G}_{p_1 p_2 p_3 \ell}}{\det \mathcal{G}_{p_1 p_2 p_3}}. \tag{3.59}$$

Recall that the only restriction on the integration contour comes from requiring that $\ell_\perp^2 > 0$ (we note that $\det \mathcal{G}_{p_1 p_2 p_3} < 0$ is the definition of the physical region).

As explained above, kinematic considerations show that the energy of the cut propagators are uniquely fixed, which means we can replace $\delta^- \to \delta$ in the loop integrand without modifying the answer. On this unitarity cut, the relevant kinematic invariants become:

$$\ell^2 = 0, \qquad \ell \cdot p_1 = \frac{s+t_L}{2}, \qquad \ell \cdot p_2 = -\frac{t_L}{2}, \qquad \ell \cdot p_3 = \frac{t_R}{2}. \tag{3.60}$$

We use the two delta functions to localize $\ell^2$ and $\ell \cdot p_1$ with Jacobian $\frac{1}{2}$, followed by trading $\ell \cdot p_2$ and $\ell \cdot p_3$ for $t_L$ and $t_R$ with Jacobian $\frac{1}{4}$. This results in

$$\mathrm{Im}A_{\mathrm{an}}^{\mathrm{p}}\big|_{s<1} = \frac{\pi^{(D-7)/2}s^{\frac{3}{2}}}{32\sqrt{tu}\Gamma(\frac{D-3}{2})} \int_{\mathcal{D}} dt_L\, dt_R\, (\ell_\perp^2)^{\frac{D-5}{2}} A_0(s,t_L)A_0(s,t_R), \tag{3.61}$$

where

$$\ell_\perp^2 = \frac{-s(t^2+t_L^2+t_R^2-2tt_L-2tt_R-2t_Lt_R)+4tt_Lt_R}{4tu}, \tag{3.62}$$

and the integration domain $\mathcal{D}$ is given by the condition $\ell_\perp^2 > 0$.

To make the expressions more brief, it is convenient to remove the kinematic dependence from $\Gamma$ by using a linear change of variables to $(x, y)$ given by:

$$t_{L/R} = \frac{\sqrt{s}}{2}\left(x\sqrt{-u} \pm y\sqrt{-t} - \sqrt{s}\right), \tag{3.63}$$

with the Jacobian $\frac{1}{2}s\sqrt{tu}$. In terms of these variables we have $\ell_\perp^2 = \frac{s}{4}\left(1-x^2-y^2\right)$ and the unitarity equation becomes in D = 10,

$$\mathrm{Im}\,\mathcal{A}_{\mathrm{an}}^{\mathrm{p}}\big|_{s<1} = \frac{\pi s^5}{15 \cdot 2^8} \int_{\mathcal{D}} dx\, dy\, \left(1-x^2-y^2\right)^{\frac{5}{2}} A_0(s,t_L)A_0(s,t_R), \tag{3.64}$$

where the integration cycle $\mathcal{D}$ is the unit disk $0 \leqslant x^2 + y^2 \leqslant 1$.

Plugging in the Veneziano amplitudes (3.10) we thus find

$$\mathrm{Im}\,\mathcal{A}_{\mathrm{an}}^{\mathrm{p}}\big|_{s<1} = \frac{\pi s^5}{15 \cdot 2^8} \int_{\mathcal{D}} dx\, dy\, \left(1-x^2-y^2\right)^{\frac{5}{2}} \frac{\Gamma(-s)\Gamma(-t_L)}{\Gamma(1-s-t_L)} \frac{\Gamma(-s)\Gamma(-t_R)}{\Gamma(1-s-t_R)}. \tag{3.65}$$

In the forward limit, $t \to 0$, we find $t_L = t_R$, which allows us to integrate out $y$ and immediately arrive on the previous result (3.51).

Crucially, in the form (3.65), the imaginary part of the amplitude is expressed in terms of convergent integrals and thus can be easily evaluated for any admissible value of $s$ and $t$. Notice that all the singular behavior is already pulled out in the explicitly factor of $\Gamma(-s)^2$ giving rise to a double pole at $s = 1$.

### 3.3.3 Massive exchanges

Let us also briefly comment on how the analysis generalizes for unitarity cuts of massive string states that appear for $s > 1$. The polarization sum becomes much harder to perform because states transform in more complicated representations of the Lorentz group. Hence we have not worked out higher unitarity cuts directly, but we will derive the corresponding formulas from the worldsheet in Section 4.2.2. However, the color sum is unchanged and the loop integration changes only slightly and thus we can discuss them here. We consider the situation as in Figure 8. We write $n_1 = m_1^2$ and $n_2 = m_2^2$ where $n_i \in \mathbb{Z}_{\geqslant 0}$ in string theory.

We have now

$$t_L = -(p_2 + \ell)^2 = -2p_2 \cdot \ell + n_2 \qquad \text{and} \qquad t_R = -(p_3 - \ell)^2 = 2p_3 \cdot \ell + n_2, \tag{3.66}$$

for the momentum transfers. We can follow exactly the same steps as in the massless case. We have now

$$P_{n_1,n_2}(t_L, t_R) = \ell_\perp^2 = -\frac{\det \mathcal{G}_{p_1 p_2 p_3 \ell}}{\det \mathcal{G}_{p_1 p_2 p_3}} = -\frac{1}{4stu}\big(s^2(t_L - t_R)^2 + 2st(n_1 + n_2 - s)(t_L + t_R)$$
$$- 4st(t_L t_R + n_1 n_2) + t^2(n_1 - n_2)^2 - st^2(2n_1 + 2n_2 - s)\big). \tag{3.67}$$

The region $\ell_\perp$ is an ellipse centered around $t_L = t_R = \frac{1}{2}(n_1 + n_2 - s)$. In order to manifest this, we can perform the change of variables:

$$t_{L/R} = \frac{\sqrt{\Delta_{n_1,n_2}}}{2\sqrt{s}}\left(\sqrt{-u}\,x \pm \sqrt{-t}\,y\right) + \frac{1}{2}(n_1 + n_2 - s), \tag{3.68}$$

with

$$\Delta_{n_1,n_2} = \left[s - (\sqrt{n_1} + \sqrt{n_2})^2\right]\left[s - (\sqrt{n_1} - \sqrt{n_2})^2\right]. \tag{3.69}$$

With this substitution, we simply have $P_{n_1,n_2} = \frac{\Delta_{n_1,n_2}}{4s}(1 - x^2 - y^2)$. The general expression for the loop integration part of the unitarity cut then reads

$$\text{Im}\,\mathcal{A} = \frac{\pi^{\frac{D-3}{2}}}{4\sqrt{stu}\,\Gamma(\frac{D-3}{2})} \sum_{\substack{\text{species} \\ \text{colors} \\ \text{polarizations}}} \int_{P_{n_1,n_2}>0} dt_L\,dt_R\,P_{n_1,n_2}(t_L, t_R)^{\frac{D-5}{2}} \mathcal{A}_0^L(s, t_L)\mathcal{A}_0^R(s, t_R) \tag{3.70}$$

$$= \frac{(\pi\Delta_{n_1,n_2})^{\frac{D-3}{2}}}{(4s)^{\frac{D-2}{2}}\Gamma(\frac{D-3}{2})} \sum_{\substack{\text{species} \\ \text{colors} \\ \text{polarizations}}} \int_{\mathcal{D}} dx\,dy\,(1 - x^2 - y^2)^{\frac{D-5}{2}} \mathcal{A}_0^L(s, t_L)\mathcal{A}_0^R(s, t_R), \tag{3.71}$$

where in the second line $t_{L/R}$ are defined according to (3.68). In particular for the annulus topology we can package the polarization sum in some polynomial of the Mandelstam variables. We write

$$\text{Im}A_{\text{an}}^{\text{p}} = \frac{\pi s^2 \Gamma(-s)^2}{60\sqrt{stu}} \sum_{n_1 \geqslant n_2 \geqslant 0} \theta\big[s - (\sqrt{n_1} + \sqrt{n_2})^2\big] \int_{P_{n_1,n_2}>0} dt_L\,dt_R\,P_{n_1,n_2}(t_L, t_R)^{\frac{5}{2}}$$
$$\times Q_{n_1,n_2}(t_L, t_R)\frac{\Gamma(-t_L)\Gamma(-t_R)}{\Gamma(n_1 + n_2 + 1 - s - t_L)\Gamma(n_1 + n_2 + 1 - s - t_R)}, \tag{3.72}$$

where $Q_{n_1,n_2}(t_L, t_R)$ is a polynomial in all the Mandelstam variables. Our computation amounts to the statement $Q_{0,0}(t_L, t_R) = 1$. Here, $Q_{n_1,n_2}(t_L, t_R)$ is in general a polynomial of degree $3(n_1 + n_2)$ in the Mandelstam variables. We will compute these polynomials from the worldsheet in Section 4.2.2. A term $(n_1, n_2)$ starts to contribute at $s = (\sqrt{n_1} + \sqrt{n_2})^2$, since this is the corresponding particle production threshold.

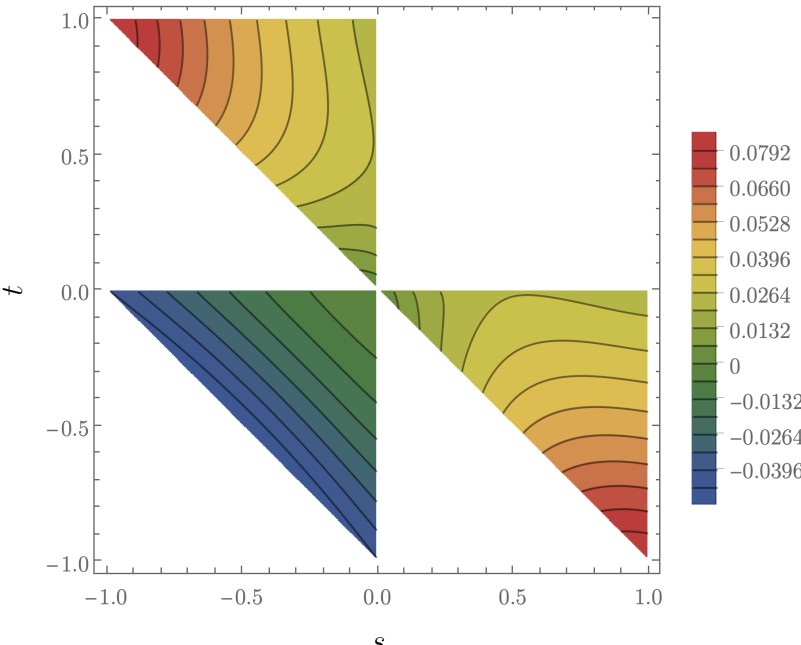

Figure 10: Numerical plot of the planar contributions (3.73) and (3.74) to the type I amplitude as a function of the Mandelstam invariants $s$ and $t$ in the three physical regions. We have set $g_s^4 t_8 = 1$ and normalized by $\frac{1}{2^9 \pi^2}(s-1)^2(t-1)^2$ in order to remove the double poles at $s = 1$ and $t = 1$.

## 3.4 All topologies and kinematic channels

The above discussion can be easily repeated for all trace structure and scattering channels, by replacing $N A_0(s, t_L) A_0(s, t_R)$ in (3.64) with the relevant factors from (3.13), (3.14), and (3.15) in the open-string case and similarly for (3.38) for closed string.

**Open string.** The total planar contribution in the $s$-channel ($s > 0$ and $t, u < 0$) equals

$$
\mathrm{Im}\,\mathcal{A}_I\big|_{s<1}^{\mathrm{tr}(t^{a_1}t^{a_2}t^{a_3}t^{a_4})} = \frac{2\pi^3 g_s^4}{15} t_8 \,\Gamma(-s)^2 s^5 \int_{\mathcal{D}} \mathrm{d}x\,\mathrm{d}y \left(1-x^2-y^2\right)^{\frac{5}{2}}
$$
$$
\times \frac{\Gamma(-t_L)}{\Gamma(1-s-t_L)} \frac{\Gamma(-t_R)}{\Gamma(1-s-t_R)} \left[N-2-\frac{2\sin(\pi s)}{\sin(\pi(s+t_R))}\right], \quad (3.73)
$$

where $t_{L/R} = \frac{\sqrt{s}}{2}\left(x\sqrt{-u} \pm y\sqrt{-t} - \sqrt{s}\right)$. The imaginary part in the $t$-channel can be obtained by relabelling $s \leftrightarrow t$. The case of the $u$-channel ($u > 0$ and $s, t < 0$) is qualitatively different and gives

$$
\mathrm{Im}\,\mathcal{A}_I\big|_{u<1}^{\mathrm{tr}(t^{a_1}t^{a_2}t^{a_3}t^{a_4})} = -\frac{2\pi^3 g_s^4 t_8 u^3}{15\Gamma(u)^2} \int_{\mathcal{D}} \mathrm{d}x'\,\mathrm{d}y' \left(1-x'^2-y'^2\right)^{\frac{5}{2}}
$$
$$
\times \Gamma(u+t_L)\Gamma(-t_L)\Gamma(u+t_R)\Gamma(-t_R), \quad (3.74)
$$

where now $t_{L/R} = \frac{\sqrt{u}}{2}\left(x'\sqrt{-s} \pm y'\sqrt{-t} - \sqrt{u}\right)$. As expected physically, the $u$-channel contribution does not have poles. In all cases, the coefficients of $N$ are those coming from the annulus topology, while those independent of $N$ come from Möbius strips in string theory. A numerical plot of this amplitude is shown in Figure 10.

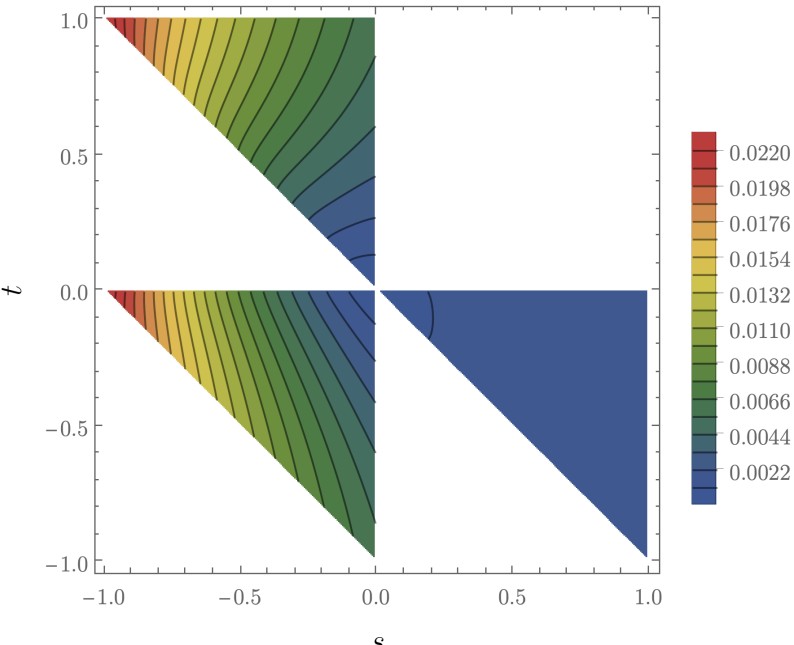

Figure 11: Numerical plot of the non-planar contributions (3.75) and (3.76) to the type I amplitude as in the $(s, t)$-plane obtained by setting $g_s^4 t_8 = 1$ and normalized by $\frac{1}{2^9 \pi^2}(s-1)^2$ in order to remove the double pole at $s = 1$.

Similarly, we can evaluate the non-planar contributions. Without loss of generality, let us focus on $\mathrm{tr}(t^{a_1} t^{a_2})\mathrm{tr}(t^{a_3} t^{a_4})$. In the $s$-channel:

$$\mathrm{Im}\,\mathcal{A}_{\mathrm{I}}\Big|_{s<1}^{\mathrm{tr}(t^{a_1} t^{a_2})\mathrm{tr}(t^{a_3} t^{a_4})} = \frac{2\pi^3 g_s^4}{15} t_8\,\Gamma(-s)^2 s^5 \int_{\mathcal{D}} \mathrm{d}x\,\mathrm{d}y\,\left(1-x^2-y^2\right)^{\frac{5}{2}}$$
$$\times \frac{\Gamma(-t_{\mathrm{L}})}{\Gamma(1-s-t_{\mathrm{L}})}\frac{\Gamma(-t_{\mathrm{R}})}{\Gamma(1-s-t_{\mathrm{R}})}\left[1-\frac{\sin(\pi t_{\mathrm{R}})}{\sin(\pi(s+t_{\mathrm{R}}))}\right], \quad (3.75)$$

while in the $u$-channel:

$$\mathrm{Im}\,\mathcal{A}_{\mathrm{I}}\Big|_{u<1}^{\mathrm{tr}(t^{a_1} t^{a_2})\mathrm{tr}(t^{a_3} t^{a_4})} = -\tfrac{1}{2}\,\mathrm{Im}\,\mathcal{A}_{\mathrm{I}}\Big|_{u<1}^{\mathrm{tr}(t^{a_1} t^{a_2} t^{a_3} t^{a_4})}. \quad (3.76)$$

The $t$-channel answer is obtained by relabelling $t \leftrightarrow u$ above. We plot the non-planar contributions in Figure 11.

**Closed string.** In the case of closed string, in the $s$-channel we have

$$\mathrm{Im}\,\mathcal{A}_{\mathrm{II}}\Big|_{s<1} = \frac{\pi^5 g_s^4}{1920} t_8 \tilde{t}_8 \frac{s^3 \Gamma(1-s)^2}{\Gamma(s)^2} \int_{\mathcal{D}} \mathrm{d}x\,\mathrm{d}y\,\left(1-x^2-y^2\right)^{\frac{5}{2}} \frac{\Gamma(-t_{\mathrm{L}})\Gamma(s+t_{\mathrm{L}})}{\Gamma(1+t_{\mathrm{L}})\Gamma(1-s-t_{\mathrm{L}})}$$
$$\times \frac{\Gamma(-t_{\mathrm{R}})\Gamma(s+t_{\mathrm{R}})}{\Gamma(1+t_{\mathrm{R}})\Gamma(1-s-t_{\mathrm{R}})}. \quad (3.77)$$

The expressions in the $t$- and $u$-channels can be obtained by relabelling. A numerical plot of this amplitude is shown in Figure 12.

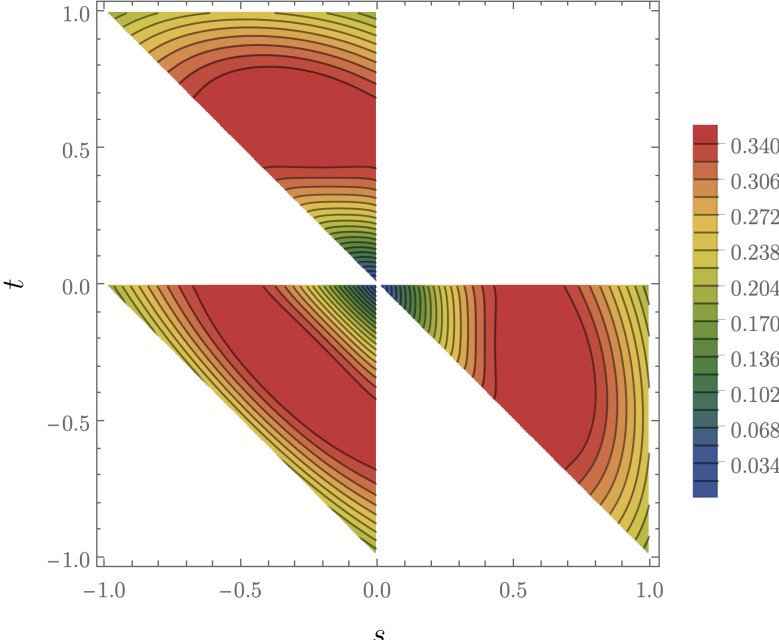

Figure 12: Numerical plot of the type II amplitude (3.77) in the $(s,t)$-plane obtained by setting $g_s^4 t_8 \tilde{t}_8 = 1$ and normalized by $\frac{2^5}{\pi^4}(s-1)^2(t-1)^2(u-1)^2$ in order to remove the double poles at $s,t,u = 1$.

# 4 Imaginary parts from the worldsheet

In this section, we will analyze the imaginary part of the amplitude further. We are essentially evaluating (2.19) and (2.20) explicitly in this section. We first discuss the open string four gluon amplitude and briefly explain the closed string calculation for a graviton scattering amplitudes in Section 4.6.

## 4.1 Decay width at $s = 1$

To warm up, we first compute a simple piece of the imaginary part of the planar amplitude $\mathcal{A}^{\mathrm{p}}$. At mass level 1, the open string has bosonic states in the representations $[2,0,0,0] = \square\square$, $[0,0,1,0] = \substack{\square \\ \square}$ of SO(9) and fermionic states in the representation $[1,0,0,1]$. These states sit in one massive supermultiplet of spacetime supersymmetry. Of course this supermultiplet is unstable against decay into the massless vector multiplet consisting of the gluon and the gluino. We will now compute the corresponding decay width.

Recall from quantum field theory, that the mass shift and decay widths can be read-off from the Dyson resummation of one-particle irreducible (1PI) diagrams. Starting with a propagator $\frac{-i}{s-m^2}$ and calling the value of the 1PI diagram $-i\Sigma(s)$, we have for a scalar field

$$\frac{-i}{s-m^2}\sum_{k=0}^{\infty}\left[-i\Sigma(s)\frac{-i}{s-m^2}\right]^k = \frac{-i}{s-m^2+\Sigma(s)}. \tag{4.1}$$

Therefore, the renormalized propagator can be identified with $\frac{-i}{s-(m^2+\delta m^2)+im\Gamma}$, where the mass shift $\delta m^2$ and decay widths $\Gamma$ at the appropriate energy scale are given by

$$\delta m^2 = -\operatorname{Re}\Sigma(s), \qquad \Gamma = \operatorname{Im}\Sigma(s)/m, \tag{4.2}$$

where $m > 0$. Note that $\operatorname{Im}\Sigma(s) \geqslant 0$ by the optical theorem.

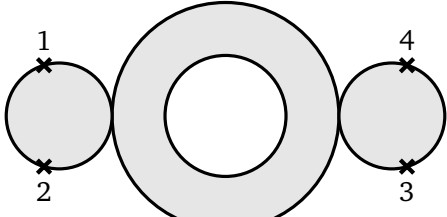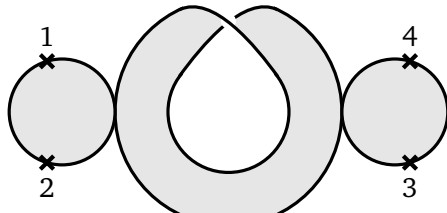

Figure 13: The double degeneration of the planar annulus and Möbius strip leading to the double pole in the amplitude at $s \in \mathbb{Z}_{\geqslant 1}$.

Thus, to one-loop approximation, the decay widths can be computed from the coefficient of the double pole of the imaginary part of the amplitude. Denoting $\mathrm{DRes}_{s=n}$ the operation of extracting this coefficient, we can incorporate the polarization and color structures accordingly and have

$$\mathop{\mathrm{DRes}}_{s=n} \mathrm{Im}\, \mathcal{A}^{\mathrm{p}} = \text{polarization and color structure} \times \sqrt{n}\, \Gamma_n \,. \tag{4.3}$$

The polarization and color structure that has to be taken out at mass level 1 is given by $t_8 \,\mathrm{tr}(t^a t^b t^c t^d)$, since these are also present at tree level. In the gluon four-point amplitude we do not have access to individual states for a given $n$. So in case that the states under consideration have degeneracy, we are really computing the sum over all decay widths. However, this will not come up in our analysis, since we are considering the first mass level. This is much simpler than computing the full imaginary part of the amplitude. The double pole comes from the double degenerations of the planar amplitude depicted in Figure 13. Only the planar amplitude can contribute to the decay width because the tree-level amplitude has also a planar color structure.

We will now work out the contribution from the planar annulus for illustration. The Möbius strip contribution is very similar. Let us choose $s$ close to 1, but not exactly equal to it. Then the integrand behaves close to the degeneration as

$$\left( \frac{\vartheta_1'(0)^2 z_{21} z_{43}}{\vartheta_1(z_{31}, \tau)\vartheta_1(z_{42}, \tau)} \right)^{-s} + \text{regular} \,, \tag{4.4}$$

where by regular we mean that the terms diverge slower than $\mathcal{O}(z_{21}^{-1} z_{43}^{-1})$ near the double degeneration. They would hence lead to a convergent integral and cannot contribute to the double pole. We now observe that we have

$$\int_{z_1} \mathrm{d}z_2 \, z_{21}^{-s} = \frac{1}{1-s} + \text{regular} \,, \tag{4.5}$$

and similarly for the $z_3$-integral. Thus we end up with just an integral over $z \equiv z_2$, as well as the $\tau$-integral. The double pole in the amplitude is hence given by the integral

$$\mathrm{Im} \mathop{\mathrm{DRes}}_{s=1} A_{\mathrm{an}}^{\mathrm{p}} = -\frac{1}{2} \int_{\circlearrowleft} \mathrm{d}\tau \int_0^1 \mathrm{d}z \left( \frac{\vartheta_1'(0,\tau)^2}{\vartheta_1(1-z,\tau)\vartheta_1(1-z,\tau)} \right)^{-1} \tag{4.6}$$

$$= -\frac{1}{8\pi^2} \int_{\circlearrowleft} \mathrm{d}\tau \int_0^1 \mathrm{d}z \, \frac{\vartheta_1(z,\tau)^2}{\eta(\tau)^6} \,, \tag{4.7}$$

where we used that $\vartheta_1'(0,\tau) = 2\pi\eta(\tau)^3$. This is a convergent integral representation of the coefficient of the double pole that we denoted by DRes. It can be further evaluated as follows.[11]

---

[11]We could also directly integrate out $z$ using $\int_0^1 \mathrm{d}z \, \vartheta_1(z,\tau)^2 = -2\vartheta_2(2\tau)$, but in view of later generalizations we will follow a different route.

Using the modular transformation of the theta and eta-function, we can change variables $\tau \rightarrow -\frac{1}{\tau}$, which leads to

$$\operatorname{Im}\operatorname*{DRes}_{s=1} A_{an}^{p} = -\frac{1}{8\pi^2} \int_{\longrightarrow} \frac{d\tau}{\tau^4} \int_0^1 dz \, e^{2\pi i z^2 \tau} \frac{\vartheta_1(z\tau,\tau)^2}{\eta(\tau)^6}. \tag{4.8}$$

Here we used the standard transformation behaviour of the theta-function under S-modular transformation. $\longrightarrow$ denotes a contour that runs horizontally in the upper half-plane.

Consider making $\operatorname{Im}(\tau)$ very large. Then most of the terms in the definition of $\vartheta_1(z\tau,\tau)$ are exponentially suppressed and do not contribute to the integral. This is the main trick that we will use throughout the paper to compute the relevant integrals. We have in fact

$$\vartheta_1(z\tau,\tau) = iq^{\frac{1}{8}-\frac{z}{2}} - iq^{\frac{1}{8}+\frac{z}{2}} - iq^{\frac{9}{8}-\frac{3z}{2}} + \dots, \tag{4.9}$$

where we recall that $q = e^{2\pi i \tau}$. Combining with the eta-function, the integrand hence behaves as

$$e^{2\pi i z^2 \tau} \frac{\vartheta_1(z\tau,\tau)^2}{\eta(\tau)^6} = q^{z^2}\left(-q^{-z} + 2 + 2q^{1-2z} - q^z - q^{2-3z}\right) + \dots. \tag{4.10}$$

All other terms are even more suppressed as $q \rightarrow 0$ and hence cannot contribute to the integral (since $0 \leqslant z \leqslant 1$). In fact, the only term here that is not suppressed, is $-q^{z^2-z}$ and thus we may replace the integrand by it. Hence

$$\operatorname{Im}\operatorname*{DRes}_{s=1} A_{an}^{p} = \frac{1}{8\pi^2} \int_{\longrightarrow} \frac{d\tau}{\tau^4} \int_0^1 dz \, q^{z(z-1)}. \tag{4.11}$$

For large $\operatorname{Im}(\tau)$, we can evaluate the integral over $z$ by extending the integration region to $(-\infty, \infty)$, which becomes a Gaussian integral. This is allowed since the integrand goes to zero for $z \in (-\infty, 0) \cup (1, \infty)$ fast enough and does not contribute to the integral.

Thus

$$\operatorname{Im}\operatorname*{DRes}_{s=1} A_{an}^{p} = \frac{1}{8\sqrt{2}\pi^2} \int_{\longrightarrow} \frac{d\tau}{(-i\tau)^{\frac{9}{2}}} e^{-\frac{\pi i \tau}{2}}. \tag{4.12}$$

Let us change variables $x = \frac{\pi i \tau}{2}$. Then the integral becomes

$$\operatorname{Im}\operatorname*{DRes}_{s=1} A_{an}^{p} = -\frac{i\pi^{\frac{3}{2}}}{128} \int_{\uparrow} dx \, (-x)^{-\frac{9}{2}} e^{-x}, \tag{4.13}$$

where $\uparrow$ denotes the rotated contour in the complex plane. We can finally deform the contour into the Hankel contour $\mathcal{H}$ that runs first from $\infty + i0^+$ to $0 + i0^+$, then surrounds zero and then runs from $0 + i0^-$ to $\infty + i0^-$. Using the Hankel representation of the Gamma-function

$$-\frac{2\pi i}{\Gamma(z)} = \int_{\mathcal{H}} dx \, (-x)^{-z} e^{-x}, \tag{4.14}$$

we get

$$\operatorname{Im}\operatorname*{DRes}_{s=1} A_{an}^{p} = \frac{\pi^{\frac{5}{2}}}{64\,\Gamma(\frac{9}{2})} = \frac{\pi^2}{420}. \tag{4.15}$$

The calculation for the Möbius strip is very similar and gives $-\frac{1}{16}$ times the result for the annulus. Hence from (4.3)

$$\Gamma_1 = g_s^2\left(1 - \frac{1}{16}\right)\frac{\pi^2}{420} = \frac{g_s^2\,\pi^2}{448}. \tag{4.16}$$

This agrees with previous results in the literature (up to our casual treatment of factors of $\alpha'$), see, e.g., [74, eq. (14)]. Similar calculations have also been performed in [27, 75–77]. However we want to emphasize that contrary to previous methods our calculation does not require any sort of regularizations: once the correct contour is used the results are manifestly convergent.

## 4.2 Planar annulus

We now work out the imaginary part of the planar annulus diagram. The method for this is very similar to the imaginary part of the double pole, but slightly more involved, since we have three $z_i$-integrals.

Let us start by mapping the integral over the circular contour $\circlearrowright$ to a horizontal contour via a modular transformation $\tau \to -\frac{1}{\tau}$. This leads to

$$
\mathrm{Im}\,A_{\mathrm{an}}^{\mathrm{p}} = -\frac{N}{64} \int_{\longrightarrow} \frac{\mathrm{d}\tau}{\tau^2} \int \mathrm{d}z_1 \, \mathrm{d}z_2 \, \mathrm{d}z_3 \, q^{sz_{41}z_{32}-tz_{21}z_{43}} \left( \frac{\vartheta_1(z_{21}\tau,\tau)\vartheta_1(z_{43}\tau,\tau)}{\vartheta_1(z_{31}\tau,\tau)\vartheta_1(z_{42}\tau,\tau)} \right)^{-s}
$$
$$
\times \left( \frac{\vartheta_1(z_{41}\tau,\tau)\vartheta_1(z_{32}\tau,\tau)}{\vartheta_1(z_{31}\tau,\tau)\vartheta_1(z_{42}\tau,\tau)} \right)^{-t}. \quad (4.17)
$$

We follow the same strategy as before. We make $\mathrm{Im}\,\tau$ very large and only have to approximate the integrand up to terms that vanish in a large $\mathrm{Im}\,\tau$ limit.

### 4.2.1 $s$-channel with $s < 1$

Let us start to work out the $s$-channel with $s < 1$. Since $0 < z_{ij} < 1$, the theta-functions have the following expansion:

$$
\vartheta_1(z\tau,\tau) = \left( iq^{\frac{1}{8}-\frac{z}{2}} - iq^{\frac{1}{8}+\frac{z}{2}} - iq^{\frac{9}{8}-\frac{3z}{2}} \right)(1 + \mathcal{O}(q)). \quad (4.18)
$$

The first term in this expansion always gives the biggest contribution. Thus, the integrand behaves for $\mathrm{Im}\,\tau$ large (i.e., $|q|$ small) like

$$
q^{\mathrm{Trop}} = q^{-s(1-z_{41})z_{32}-tz_{21}z_{43}}, \quad (4.19)
$$

up to subleading terms. Here Trop is essentially the tropicalization of the integrand and will play an important role throughout the paper. Assuming that $s$ is not too large ($s < 4$ is sufficient here), the exponent is bounded from below by $-1$, which means that the higher corrections in the expansion of the theta-functions than the one spelled out in (4.18) will not influence the result. The region Trop $< 0$ in $(z_1, z_2, z_3)$-space is pictured in Figure 14.

Next, we should be more precise in our estimate and investigate the corrections to the large $\mathrm{Im}\,\tau$ behaviour coming from the second and third term in (4.18). Consider the limit $z_{32} \to 0$, while the other $z_i$'s are kept fixed. Then the behaviour as $\mathrm{Im}\,\tau \to \infty$ of the integrand is

$$
q^{-s(1-z_{41})z_{32}-tz_{21}z_{43}}(1-q^{z_{32}})^{-t}. \quad (4.20)
$$

Hence the first correction to the exponent is

$$
-s(1-z_{41})z_{32} - tz_{21}z_{43} + z_{32}. \quad (4.21)
$$

Since $s > 0$ and $t < 0$, this correction would result in something positive as long as $s < 1$ (because $1 - z_{41} < 1$). Positive exponents of $q$ never contribute to the integral in a small $q$-limit, and thus we learn that we do not have to keep the second term in the expansion of $\vartheta_1(z_{32}\tau,\tau)$. This line of reasoning allows one to discard most of the terms in the expansion of the theta-function in (4.18). At the end, one is only left with the second terms in the expansion (4.18) for $\vartheta_1(z_{21}\tau,\tau)$ and $\vartheta_1(z_{43}\tau,\tau)$, since in those cases, the exponent (4.19) does not go to zero. Thus we have for $s < 1$

$$
\mathrm{Im}\,A_{\mathrm{an}}^{\mathrm{p}} = -\frac{N}{64} \int_{\longrightarrow} \frac{\mathrm{d}\tau}{\tau^2} \int \mathrm{d}z_1 \, \mathrm{d}z_2 \, \mathrm{d}z_3 \, q^{-s(1-z_{41})z_{32}-tz_{21}z_{43}} (1-q^{z_{21}})^{-s}(1-q^{z_{43}})^{-s}. \quad (4.22)
$$

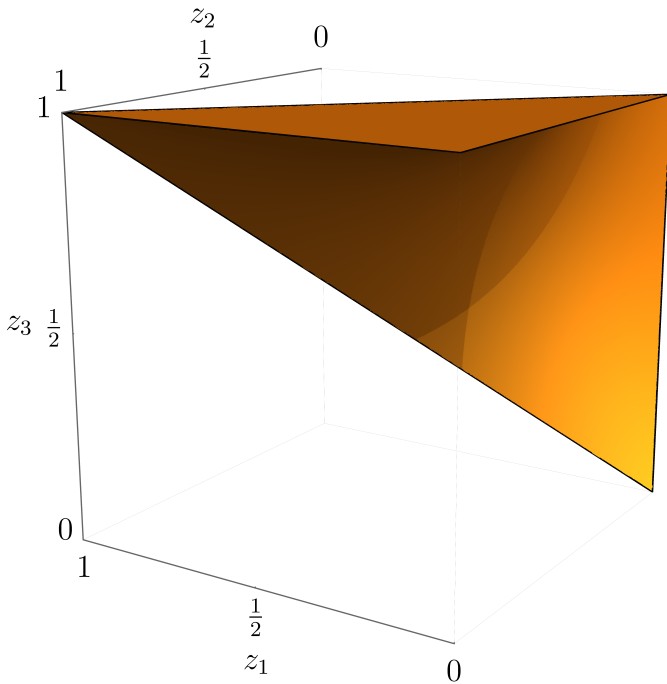

Figure 14: The region where Trop $< 0$ for the planar annulus. The figure is drawn for $s = 1$ and $t = -\frac{1}{2}$.

Let us set

$$\alpha_L = z_{21}, \qquad \alpha_R = z_{43}, \qquad t_L = -sz_{32} + tz_{43}. \tag{4.23}$$

Then we can rewrite the imaginary part as

$$\mathrm{Im}A^{\mathrm{p}}_{\mathrm{an}} = -\frac{N}{64} \int_{\longrightarrow} \frac{\mathrm{d}\tau}{\tau^2} \int_{\mathcal{R}} \mathrm{d}\alpha_L \, \mathrm{d}\alpha_R \, \mathrm{d}t_L \, \mathrm{d}t_R \, q^{t\alpha_R(\alpha_R-1)+t_L(1-\alpha_L-\alpha_R)+\frac{1}{s}(t_L-t\alpha_R)^2}$$
$$\times (1-q^{\alpha_L})^{-s}(1-q^{\alpha_R})^{-s} \sqrt{\frac{-i\tau}{2stu}} \, q^{-\frac{1}{4st(s+t)}(st_R-(s+2t)t_L+2t(s+t)\alpha_R-st)^2}. \tag{4.24}$$

We inserted an identity in the second line. The Gaussian integral over $t_R$ directly cancels the additional factors that we inserted. The integral over $(\alpha_L, \alpha_R, t_L, t_R)$ goes over some convex region $\mathcal{R}$ that we discuss momentarily. Simplifying the exponents leads to

$$\mathrm{Im}A^{\mathrm{p}}_{\mathrm{an}} = \frac{N}{64\sqrt{2stu}} \int_{\longrightarrow} \frac{\mathrm{d}\tau}{(-i\tau)^{\frac{3}{2}}} \int_{\mathcal{R}} \mathrm{d}\alpha_L \, \mathrm{d}\alpha_R \, \mathrm{d}t_L \, \mathrm{d}t_R \, q^{-t_L\alpha_L-t_R\alpha_R-P(t_L,t_R)}$$
$$\times (1-q^{\alpha_L})^{-s}(1-q^{\alpha_R})^{-s}, \tag{4.25}$$

where

$$P(t_L, t_R) = -\frac{s(t^2+t_L^2+t_R^2-2tt_L-2tt_R-2t_Lt_R)-4tt_Lt_R}{4tu}. \tag{4.26}$$

Notice the simple behaviour of the integral on $\alpha_L$ and $\alpha_R$. The definitions of $t_L$ and the shift in the Gaussian integral of $t_R$ was chosen to ensure the linear $\alpha_L$ and $\alpha_R$ dependence of the exponent. The final aspect we need to take care of is the region $\mathcal{R}$ over which we should integrate. The initial integration region $\mathcal{R}$ is described by the inequalities

$$\alpha_L \geqslant 0, \qquad \alpha_R \geqslant 0, \qquad t_L \leqslant t\alpha_R, \qquad \alpha_L + \frac{s+t}{s}\alpha_R - \frac{1}{s}t_L \leqslant 1. \tag{4.27}$$

We will now change the integration region to $\tilde{\mathcal{R}}$, which is described by the inequalities

$$\alpha_{\mathrm{L}} \geqslant 0, \qquad \alpha_{\mathrm{R}} \geqslant 0, \qquad t_{\mathrm{L}} \leqslant 0, \qquad t_{\mathrm{R}} \leqslant 0. \tag{4.28}$$

We claim that the integral is unchanged when changing the integration region like this. For this, we need to check that the exponent $-t_{\mathrm{L}}\alpha_{\mathrm{L}} - t_{\mathrm{R}}\alpha_{\mathrm{R}} - P(t_{\mathrm{L}}, t_{\mathrm{R}})$ is positive on the difference $(\mathcal{R} \cap \tilde{\mathcal{R}}^c) \cup (\mathcal{R}^c \cap \tilde{\mathcal{R}})$ of the two regions. This is easily checked in `Mathematica` using the `Reduce` command, or directly by hand. We can use the integral formula

$$\int_0^\infty \mathrm{d}\alpha \, q^{-t\alpha}(1-q^\alpha)^{-s} = \frac{i}{2\pi\tau} \int_0^1 \mathrm{d}x \, (1-x)^{-s} x^{-1-t} = \frac{i}{2\pi\tau} \frac{\Gamma(1-s)\Gamma(-t)}{\Gamma(1-s-t)}, \tag{4.29}$$

where we substituted $\alpha = \frac{1}{2\pi i \tau} \log x$. We obtain from (4.25) after changing $\mathcal{R} \to \tilde{\mathcal{R}}$ and integrating out $(\alpha_{\mathrm{L}}, \alpha_{\mathrm{R}})$

$$\mathrm{Im}\,A_{\mathrm{an}}^{\mathrm{p}} = \frac{N}{256\pi^2 \sqrt{2stu}} \int_{\longrightarrow} \frac{\mathrm{d}\tau}{(-i\tau)^{\frac{7}{2}}} \int_{-\infty}^0 \mathrm{d}t_{\mathrm{L}} \int_{-\infty}^0 \mathrm{d}t_{\mathrm{R}} \, q^{-P(t_{\mathrm{L}},t_{\mathrm{R}})}$$
$$\times \frac{\Gamma(1-s)\Gamma(-t_{\mathrm{L}})}{\Gamma(1-s-t_{\mathrm{L}})} \frac{\Gamma(1-s)\Gamma(-t_{\mathrm{R}})}{\Gamma(1-s-t_{\mathrm{R}})}. \tag{4.30}$$

We are almost done. We restrict the integration region in $(t_{\mathrm{L}}, t_{\mathrm{R}})$ to the region with $P(t_{\mathrm{L}}, t_{\mathrm{R}}) > 0$, since otherwise the $\tau$-integral gives zero. For $P(t_{\mathrm{L}}, t_{\mathrm{R}})$, we notice the Hankel representation of the Gamma-function as in the computation of the decay in Section 4.1. We have

$$\int_{\longrightarrow} \frac{\mathrm{d}\tau}{(-i\tau)^{\frac{7}{2}}} \, q^{-P(t_{\mathrm{L}},t_{\mathrm{R}})} = \frac{64\sqrt{2}\,\pi^3}{15} P(t_{\mathrm{L}}, t_{\mathrm{R}})^{\frac{5}{2}}. \tag{4.31}$$

Putting everything together gives

$$\mathrm{Im}\,A_{\mathrm{an}}^{\mathrm{p}}\Big|_{s<1} = \frac{N\pi}{60\sqrt{stu}} \int_{P>0} \mathrm{d}t_{\mathrm{L}} \, \mathrm{d}t_{\mathrm{R}} \, P(t_{\mathrm{L}}, t_{\mathrm{R}})^{\frac{5}{2}} \frac{\Gamma(1-s)\Gamma(-t_{\mathrm{L}})}{\Gamma(1-s-t_{\mathrm{L}})} \frac{\Gamma(1-s)\Gamma(-t_{\mathrm{R}})}{\Gamma(1-s-t_{\mathrm{R}})}. \tag{4.32}$$

From this, one can easily check that one gets back the correct double residue at $s = 1$, what we computed in (4.15). From the discussion in Section 3, we see that $t_{\mathrm{L}}$ and $t_{\mathrm{R}}$ have the interpretation of the momentum transfers in the unitarity cut. Their appearance from the worldsheet is somewhat obscure to us. One of them appeared as a Schwinger parameter, whereas the other momentum transfer appeared as an auxiliary variable. We also see that $P(t_{\mathrm{L}}, t_{\mathrm{R}})$ is precisely $(\ell_\perp)^2$, where $\ell_\perp$ is the component of the loop momentum that is orthogonal to all external momenta, see Section 3.3.2.

**Forward limit.** Let us mention that the analysis simplifies significantly in the forward limit (i.e., for $t = 0$). In this case it is unnecessary to introduce $t_{\mathrm{R}}$ and we have

$$\mathrm{Im}\,A_{\mathrm{an}}^{\mathrm{p}}\Big|_{s<1} = -\frac{N}{64s} \int_{\longrightarrow} \frac{\mathrm{d}\tau}{\tau^2} \int_{\mathcal{R}} \mathrm{d}\alpha_{\mathrm{L}} \, \mathrm{d}\alpha_{\mathrm{R}} \, \mathrm{d}t_{\mathrm{L}} \, q^{t_{\mathrm{L}}(1-\alpha_{\mathrm{L}}-\alpha_{\mathrm{R}})+\frac{1}{s}t_{\mathrm{L}}^2}(1-q^{\alpha_{\mathrm{L}}})^{-s}(1-q^{\alpha_{\mathrm{R}}})^{-s}$$
$$= \frac{N}{256\pi^2 s} \int_{\longrightarrow} \frac{\mathrm{d}\tau}{(-i\tau)^4} \int_{\mathcal{R}} \mathrm{d}\alpha_{\mathrm{L}} \, \mathrm{d}\alpha_{\mathrm{R}} \, \mathrm{d}t_{\mathrm{L}} \, q^{t_{\mathrm{L}}+\frac{1}{s}t_{\mathrm{L}}^2}\left(\frac{\Gamma(1-s)\Gamma(-t_{\mathrm{L}})}{\Gamma(1-s-t_{\mathrm{L}})}\right)^2$$
$$= \frac{N\pi^2}{96s^4} \int_{-s}^0 \mathrm{d}t_{\mathrm{L}} \, (-t_{\mathrm{L}})^3 (s+t_{\mathrm{L}})^3 \left(\frac{\Gamma(1-s)\Gamma(-t_{\mathrm{L}})}{\Gamma(1-s-t_{\mathrm{L}})}\right)^2. \tag{4.33}$$

Alternatively, one can also see this by noticing that the ellipse defined by $P(t_{\mathrm{L}}, t_{\mathrm{R}}) > 0$ degenerates to a diagonal line in this case that forces $t_{\mathrm{L}} = t_{\mathrm{R}}$. This is discussed more systematically in Section 6.2.

### 4.2.2 Larger values of $s$

Let us now extend the analysis to larger values of $s$. In this case, more terms in the $q$-expansion of $\vartheta_1$ have to be kept, but otherwise the steps are exactly the same. For any choice of $s$, there is a finite number of terms in the $q$-expansion that can contribute to the imaginary part. As we explained in Section 3.3.3, the expected structure is given by eq. (3.72).

For the first few cases, we find

$$Q_{0,0} = 1, \tag{4.34a}$$

$$Q_{1,0} = 2\left(-2st_Lt_R - s^2t_L + st_L - s^2t_R + st_R + s^2t - 2st + t\right), \tag{4.34b}$$

$$\begin{aligned}
Q_{2,0} = {}&2s^4t_Lt_R + 4s^3t_Lt_R^2 + 4s^3t_L^2t_R - 4s^3tt_Lt_R - 12s^3t_Lt_R + 4s^2t_L^2t_R^2 - 10s^2t_Lt_R^2 \\
&- 10s^2t_L^2t_R + 12s^2tt_Lt_R + 18s^2t_Lt_R - 2st_L^2t_R^2 + 4st_Lt_R^2 + 4st_L^2t_R - 12stt_Lt_R \\
&- 6st_Lt_R + 4tt_Lt_R + s^4t_L^2 - 2s^4tt_L - s^4t_L - 4s^3t_L^2 + 10s^3tt_L + 4s^3t_L + 5s^2t_L^2 \\
&- 18s^2tt_L - 5s^2t_L - 2st_L^2 + 14stt_L + 2st_L - 4tt_L + s^4t_R^2 - 2s^4tt_R - s^4t_R \\
&- 4s^3t_R^2 + 10s^3tt_R + 4s^3t_R + 5s^2t_R^2 - 18s^2tt_R - 5s^2t_R - 2st_R^2 + 14stt_R \\
&+ 2st_R - 4tt_R + s^4t^2 + s^4t - 6s^3t^2 - 6s^3t + 13s^2t^2 + 13s^2t - 12st^2 - 12st \\
&+ 4t^2 + 4t.
\end{aligned} \tag{4.34c}$$

Higher values of $Q_{n_1,n_2}(t_L, t_R)$ are tabulated in an ancillary file `Q.txt` attached to the submission.

## 4.3 Möbius strip

We next compute the contribution to the imaginary part of the planar amplitude from the Möbius strip. For $\tilde{\tau} \in i + \mathbb{R}$, let

$$\tau = \frac{\tilde{\tau} - 1}{2\tilde{\tau} - 1}, \tag{4.35}$$

and the same functional form holds for the inverse relation. Then $\tau$ lies on the circle that touches the real line at $\tau = \frac{1}{2}$, see Figure 1. We thus want to change variables from $\tau$ to $\tilde{\tau}$, since it maps the circular contour $\circlearrowright$ to a horizontal contour $\longrightarrow$ as before. For this we can use the transformation behaviour of $\vartheta_1$,

$$\vartheta_1\left(\frac{z}{c\tau + d}, \frac{a\tau + b}{c\tau + d}\right) = \varepsilon\sqrt{c\tau + d}\, e^{\frac{\pi i c z^2}{c\tau + d}}\, \vartheta_1(z; \tau). \tag{4.36}$$

Here, $\varepsilon \equiv \varepsilon(a, b, c, d)$ is an eighth root of unity that is in general complicated to spell out in closed form, but cancels out of our calculation. We hence compute

$$\vartheta_1(z, \tau) = \varepsilon(2\tau - 1)^{-\frac{1}{2}}\, e^{-\frac{2\pi i z^2}{2\tau - 1}}\, \vartheta_1\left(\frac{z}{2\tau - 1}, \frac{\tau - 1}{2\tau - 1}\right) \tag{4.37}$$

$$= \varepsilon(2\tilde{\tau} - 1)^{\frac{1}{2}}\, e^{2\pi i (2\tilde{\tau} - 1)z^2}\, \vartheta_1\left(-z(2\tilde{\tau} - 1), \tilde{\tau}\right). \tag{4.38}$$

We do not have to know its precise form since it cancels out of the integrand. We hence have[12]

$$\begin{aligned}
\text{Im}\, A_{\text{Möb}} = \frac{1}{2}\int_{\longrightarrow} \frac{d\tilde{\tau}}{(2\tilde{\tau} - 1)^2} \int_{0 < z_1 < z_2 < z_3 < 1} & dz_1\, dz_2\, dz_3\, e^{4\pi i s(2\tilde{\tau} - 1)z_{41}z_{32} - 4\pi i t(2\tilde{\tau} - 1)z_{43}z_{21}} \\
&\times \left(\frac{\vartheta_1(z_{21}(2\tilde{\tau} - 1), \tilde{\tau})\vartheta_1(z_{43}(2\tilde{\tau} - 1), \tilde{\tau})}{\vartheta_1(z_{31}(2\tilde{\tau} - 1), \tilde{\tau})\vartheta_1(z_{42}(2\tilde{\tau} - 1), \tilde{\tau})}\right)^{-s} \\
&\times \left(\frac{\vartheta_1(z_{32}(2\tilde{\tau} - 1), \tilde{\tau})\vartheta_1(z_{41}(2\tilde{\tau} - 1), \tilde{\tau})}{\vartheta_1(z_{31}(2\tilde{\tau} - 1), \tilde{\tau})\vartheta_1(z_{42}(2\tilde{\tau} - 1), \tilde{\tau})}\right)^{-t}.
\end{aligned} \tag{4.39}$$

---

[12]There is an extra minus sign in this formula, since we turned around the direction of the contour.

To obtain slightly nicer formulas, we will also rename $(2\tilde{\tau} - 1) \to 2\tau$ (which differs of course from the original $\tau$). Here, $\tau$ runs also along a horizontal contour, which we take to be $\tau \in iL + \mathbb{R}$ for some $L \in \mathbb{R}_{>0}$. Calling also $q = e^{2\pi i\tau}$, we have

$$
\operatorname{Im} A_{\text{Möb}} = \frac{1}{8} \int_{\longrightarrow} \frac{d\tau}{\tau^2} \int_{0 < z_1 < z_2 < z_3 < 1} dz_1 \, dz_2 \, dz_3 \; q^{4sz_{41}z_{32} - 4tz_{43}z_{21}}
$$
$$
\times \left( \frac{\vartheta_1\left(2z_{21}\tau, \tau + \frac{1}{2}\right) \vartheta_1\left(2z_{43}\tau, \tau + \frac{1}{2}\right)}{\vartheta_1\left(2z_{31}\tau, \tau + \frac{1}{2}\right) \vartheta_1\left(2z_{42}\tau, \tau + \frac{1}{2}\right)} \right)^{-s}
$$
$$
\times \left( \frac{\vartheta_1\left(2z_{41}\tau, \tau + \frac{1}{2}\right) \vartheta_1\left(2z_{32}\tau, \tau + \frac{1}{2}\right)}{\vartheta_1\left(2z_{31}\tau, \tau + \frac{1}{2}\right) \vartheta_1\left(2z_{42}\tau, \tau + \frac{1}{2}\right)} \right)^{-t} . \tag{4.40}
$$

We can now make $L$ very large as for the annulus. This has again the effect of simplifying the involved $\vartheta_1$-functions, since only a finite number of terms in their $q$-expansion can contribute to the integral. In fact, since $0 < z_{ij} < 1$, we have

$$
\vartheta_1\left(2z\tau, \tau + \frac{1}{2}\right) = \varepsilon \left( q^{\frac{1}{8} + z} - q^{\frac{1}{8} - z} - q^{\frac{9}{8} - 3z} + q^{\frac{25}{8} - 5z} \right)(1 + \mathcal{O}(q)) , \tag{4.41}
$$

where $\varepsilon$ is a 16th order root of unity that cancels out of the expression.

### 4.3.1 $s$-channel with $s < 1$

Assuming again that $s < 1$, we can stop the expansion at this point, since the exponent of the prefactor $q^{4sz_{41}z_{32} - 4tz_{43}z_{21}}$ is always bigger than $-1$ and any positive $q$-exponent can be scaled away by making $\operatorname{Im}(\tau)$ very large. The integrand hence grows for $\operatorname{Im}(\tau) \to \infty$ as $q^{\text{Trop}}$, where

$$
\begin{aligned}
\text{Trop} = {} & 4sz_{41}z_{32} - 4tz_{43}z_{21} + s\left(\max(z_{21}, 3z_{21} - 1) + \max(z_{43}, 3z_{43} - 1)\right) \\
& + t\left(\max(z_{32}, 3z_{32} - 1) + \max(z_{41}, 3z_{41} - 1)\right) \\
& + u\left(\max(z_{31}, 3z_{31} - 1) + \max(z_{42}, 3z_{42} - 1)\right).
\end{aligned} \tag{4.42}
$$

The additional terms come from the second and third term in the expansion (4.41). By the now familiar argument, only regions with $\text{Trop} < 0$ can contribute to the integral. This region is actually disconnected in the present case. A plot is shown in Figure 15. Thus it is natural to subdivide the integration regions in $z_i$ into four different pieces. We always assume that $z_4 = 1$. Then we can split the integration region $0 < z_1 < z_2 < z_3 < 1$ in to the following subregions, so that each subregion only contains one of the components where $\text{Trop} < 0$. We set

$$
\Gamma^{(1)} = \{0 < z_{21}, z_{31}, z_{41}, z_{32}, z_{42}, z_{43} < \tfrac{1}{2}\}, \tag{4.43a}
$$
$$
\Gamma^{(2)} = \{0 < z_{21}, z_{43} < \tfrac{1}{2}, \tfrac{1}{2} < z_{31}, z_{41}, z_{32}, z_{42} < 1\}, \tag{4.43b}
$$
$$
\Gamma^{(3)} = \{0 < z_{32}, z_{42}, z_{43} < \tfrac{1}{2}, \tfrac{1}{2} < z_{31}, z_{41} < 1\}, \tag{4.43c}
$$
$$
\Gamma^{(4)} = \{0 < z_{21}, z_{31}, z_{32} < \tfrac{1}{2}, \tfrac{1}{2} < z_{41}, z_{42} < 1\}. \tag{4.43d}
$$

These four different regions correspond to the four different ways the Möbius strip can be cut. They are depicted in Figure 16.

In the different regions various different terms may be dropped in the expansions of the theta-functions. It is also important to know which boundaries of the regions $\Gamma^{(i)}$ are part of the region $\text{Trop} < 0$, since this tells us where we need to keep terms in the theta-functions that

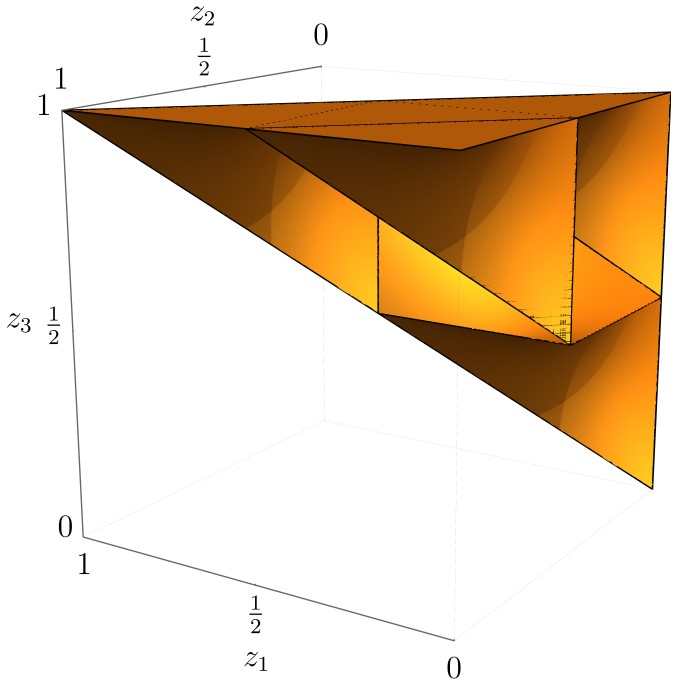

Figure 15: The four regions in the $(z_1, z_2, z_3)$-space where Trop $< 0$. The picture is drawn for $s = 1$ and $t = -\frac{1}{2}$.

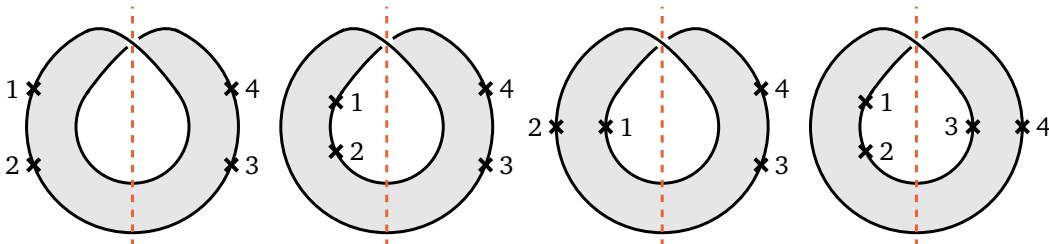

Figure 16: The four different cuttings of the Möbius strip amplitude in the $s$-channel. They correspond to the four regions $\Gamma^{(1)}, \dots, \Gamma^{(4)}$ (in this order).

become relevant near the boundary. We have

$$\partial\Gamma^{(1)} \cap \partial\{\text{Trop} < 0\} = \{z_1 = z_2\} \cup \{z_3 = z_4 = 1\}, \tag{4.44a}$$

$$\partial\Gamma^{(2)} \cap \partial\{\text{Trop} < 0\} = \{z_1 = z_2\} \cup \{z_3 = z_4 = 1\}, \tag{4.44b}$$

$$\partial\Gamma^{(3)} \cap \partial\{\text{Trop} < 0\} = \{z_3 = z_4 = 1\}, \tag{4.44c}$$

$$\partial\Gamma^{(4)} \cap \partial\{\text{Trop} < 0\} = \{z_1 = z_2\}. \tag{4.44d}$$

Let us denote the contributions to the imaginary parts of the amplitude by $\text{Im}\,A^{(i)}$. Then

$$\text{Im}\,A_{\text{Möb}}^{(1)} = \frac{1}{8}\int_{\longrightarrow}\frac{d\tau}{\tau^2}\int_{\Gamma^{(1)}} dz_1\,dz_2\,dz_3\; q^{-4s(z_1-\frac{1}{2})z_{32}-4tz_{43}z_{21}}(1-q^{2z_{21}})^{-s}(1-q^{2z_{43}})^{-s}, \tag{4.45a}$$

$$\text{Im}\,A_{\text{Möb}}^{(2)} = \frac{1}{8}\int_{\longrightarrow}\frac{d\tau}{\tau^2}\int_{\Gamma^{(2)}} dz_1\,dz_2\,dz_3\; q^{-4sz_1(z_{32}-\frac{1}{2})-4tz_{43}z_{21}}(1-q^{2z_{21}})^{-s}(1-q^{2z_{43}})^{-s}, \tag{4.45b}$$

$$\text{Im}\,A_{\text{Möb}}^{(3)} = \frac{1}{8}\int_{\longrightarrow}\frac{d\tau}{\tau^2}\int_{\Gamma^{(3)}} dz_1\,dz_2\,dz_3\; q^{-4sz_1(z_{32}-\frac{1}{2})+2s(\frac{1}{2}-z_2)-4tz_{43}(z_{21}-\frac{1}{2})}$$
$$\times(1+q^{1-2z_{21}})^{-s}(1-q^{2z_{43}})^{-s}, \tag{4.45c}$$

$$\text{Im}\,A_{\text{Möb}}^{(4)} = \frac{1}{8}\int_{\longrightarrow}\frac{d\tau}{\tau^2}\int_{\Gamma^{(4)}} dz_1\,dz_2\,dz_3\; q^{-4sz_1z_{32}+2s(z_3-\frac{1}{2})-4t(z_{43}-\frac{1}{2})z_{21}}$$
$$\times(1-q^{z_{21}})^{-s}(1+q^{1-2z_{43}})^{-s}. \tag{4.45d}$$

We can simplify this as follows. First we notice that $\text{Im}\,A_{\text{Möb}}^{(1)} = \text{Im}\,A_{\text{Möb}}^{(2)}$, which follows by shifting $z_1 \to z_1 + \frac{1}{2}$ and $z_2 \to z_2 + \frac{1}{2}$. We also notice that $\text{Im}\,A^{(1)}$ is the same as $-\frac{1}{N} \times$ the imaginary part of the planar annulus amplitude (4.22). We finally observe that $\text{Im}\,A_{\text{Möb}}^{(3)} = \text{Im}\,A_{\text{Möb}}^{(4)}$, which follows from the change of variables

$$z_1^{(3)} = z_3^{(4)} - \frac{1}{2}, \quad z_2^{(3)} = z_4^{(4)} - \frac{1}{2}, \quad z_3^{(3)} = z_1^{(4)} + \frac{1}{2}, \quad z_4^{(3)} = z_2^{(4)} + \frac{1}{2}. \tag{4.46}$$

that maps $\Gamma^{(3)}$ to $\Gamma^{(4)}$ (after shifting $z_4^{(4)}$ to $z_4^{(4)} = 1$). Thus all that remains is to compute $\text{Im}\,A_{\text{Möb}}^{(3)}$. We change variables to

$$z_{21} = \frac{1}{2}(1-\alpha_{\text{L}}), \qquad z_{43} = \frac{\alpha_{\text{R}}}{2}, \qquad t_{\text{L}} = -2sz_1 + 2tz_{43}. \tag{4.47}$$

We also integrate in the factor

$$1 = s\sqrt{\frac{-i\tau}{2stu}}\int_{-\infty}^{\infty} dt_{\text{R}}\, q^{-\frac{1}{4st(s+t)}(st_{\text{R}}-(s+2t)t_{\text{L}}+2t(s+t)\alpha_{\text{R}}-st)^2} \tag{4.48}$$

as before. Then the imaginary part of the amplitude becomes

$$\text{Im}\,A_{\text{Möb}}^{(3)} = -\frac{1}{64\sqrt{2stu}}\int\frac{d\tau}{(-i\tau)^{\frac{3}{2}}}\int_{\mathcal{R}} dt_{\text{L}}\,dt_{\text{R}}\,d\alpha_{\text{L}}\,d\alpha_{\text{R}}\; q^{\alpha_{\text{L}}(s+t_{\text{L}})-\alpha_{\text{R}}t_{\text{R}}-P(t_{\text{L}},t_{\text{R}})}$$
$$\times(1+q^{\alpha_{\text{L}}})^{-s}(1-q^{\alpha_{\text{R}}})^{-s}, \tag{4.49}$$

with the same polynomial $P(t_{\text{L}}, t_{\text{R}})$ as in the planar annulus amplitude, see eq. (4.26).

We next want to change the integration region from the initial integration region to a new region $\tilde{\mathcal{R}}$ where

$$\alpha_{\text{R}} \geqslant 0, \qquad -s \leqslant t_{\text{L}} \leqslant 0, \qquad -s \leqslant t_{\text{R}} \leqslant 0. \tag{4.50}$$

Using the `Reduce` command in `Mathematica` one checks again that the leading exponent is positive in the $q \to 0$ limit on the difference on the two sets. Hence this change of integration region does not affect the result.

We next use the integral identity

$$\int_{-\infty}^{\infty} \mathrm{d}\alpha_{\mathrm{L}}\, q^{\alpha_{\mathrm{L}}(s+t_{\mathrm{L}})}(1+q^{\alpha_{\mathrm{L}}})^{-s} = \frac{i\,\Gamma(s+t_{\mathrm{L}})\Gamma(-t_{\mathrm{L}})}{2\pi\tau\,\Gamma(s)}, \tag{4.51}$$

together with (4.29) to obtain

$$\mathrm{Im}A_{\mathrm{M\ddot{o}b}}^{(3)} = -\frac{1}{256\pi^2\sqrt{2stu}}\int \frac{\mathrm{d}\tau}{(-i\tau)^{\frac{7}{2}}}\int_{-s}^{0}\mathrm{d}t_{\mathrm{L}}\int_{-s}^{0}\mathrm{d}t_{\mathrm{R}}\,q^{-P(t_{\mathrm{L}},t_{\mathrm{R}})}$$
$$\times \frac{\Gamma(s+t_{\mathrm{L}})\Gamma(-t_{\mathrm{L}})\Gamma(1-s)\Gamma(-t_{\mathrm{R}})}{\Gamma(s)\Gamma(1-s-t_{\mathrm{R}})} \tag{4.52}$$

$$= -\frac{\pi}{60\sqrt{stu}}\int \mathrm{d}t_{\mathrm{L}}\,\mathrm{d}t_{\mathrm{R}}\,P(t_{\mathrm{L}},t_{\mathrm{R}})^{\frac{5}{2}}\frac{\Gamma(s+t_{\mathrm{L}})\Gamma(-t_{\mathrm{L}})\Gamma(1-s)\Gamma(-t_{\mathrm{R}})}{\Gamma(s)\Gamma(1-s-t_{\mathrm{R}})}, \tag{4.53}$$

where we used (4.31) and it is understood that the integral runs over the region where $P(t_{\mathrm{L}}, t_{\mathrm{R}})$ is positive. We hence recognize this contribution as a gluing of a $u$-channel disk amplitude and an $s$-channel disk amplitude, as discussed in Section 3.1. For reference, we hence find the following contribution to the imaginary part from the Möbius strip in the $s$-channel after summing over the four contributions:

$$\mathrm{Im}A_{\mathrm{M\ddot{o}b}} = -\frac{\pi\,\Gamma(1-s)^2}{30\sqrt{stu}}\int \mathrm{d}t_{\mathrm{L}}\,\mathrm{d}t_{\mathrm{R}}\,P(t_{\mathrm{L}},t_{\mathrm{R}})^{\frac{5}{2}}\frac{\Gamma(-t_{\mathrm{L}})\Gamma(-t_{\mathrm{R}})}{\Gamma(1-s-t_{\mathrm{L}})\Gamma(1-s-t_{\mathrm{R}})}$$
$$\times \left(1 + \frac{\sin(\pi s)}{\sin(\pi(s+t_{\mathrm{L}}))}\right). \tag{4.54}$$

### 4.3.2  $u$-channel with $u < 1$

For the Möbius strip, there is also an imaginary part in the $u$-channel where $s < 0$, $t < 0$ and $u > 0$. The exponent Trop given by (4.42) is now negative in a single connected region that does not touch the boundary of the integration region $0 < z_1 < z_2 < z_3 < 1$. A picture of the relevant region is given in Figure 17. In fact, the region of negativity is fully contained in the set

$$\Gamma = \{0 < z_{21}, z_{32}, z_{43} < \tfrac{1}{2}, \tfrac{1}{2} < z_{41} < 1, 0 < z_1 < z_2 < z_3 < 1\}. \tag{4.55}$$

This tells us again which terms in the expansion of the theta-functions we should keep. We get

$$\mathrm{Im}A_{\mathrm{M\ddot{o}b}} = \frac{1}{8}\int_{\longrightarrow} \frac{\mathrm{d}\tau}{\tau^2}\int_{\Gamma}\mathrm{d}z_1\,\mathrm{d}z_2\,\mathrm{d}z_3\,q^{4s(z_{41}-\frac{1}{2})z_{32}-4tz_{43}z_{21}+2t(z_{41}-\frac{1}{2})}$$
$$\times (1+q^{1-2z_{31}})^{-u}(1+q^{1-2z_{42}})^{-u}. \tag{4.56}$$

We can again change variables as before. The relevant change of variables is

$$z_1 = \frac{\alpha_{\mathrm{L}}}{2} - \frac{s\alpha_{\mathrm{R}}}{2u} - \frac{t_{\mathrm{L}}}{2u}, \quad z_2 = \frac{\alpha_{\mathrm{R}}+1}{2}, \quad z_3 = \frac{1}{2} - \frac{s\alpha_{\mathrm{R}}}{2u} - \frac{t_{\mathrm{L}}}{2u}. \tag{4.57}$$

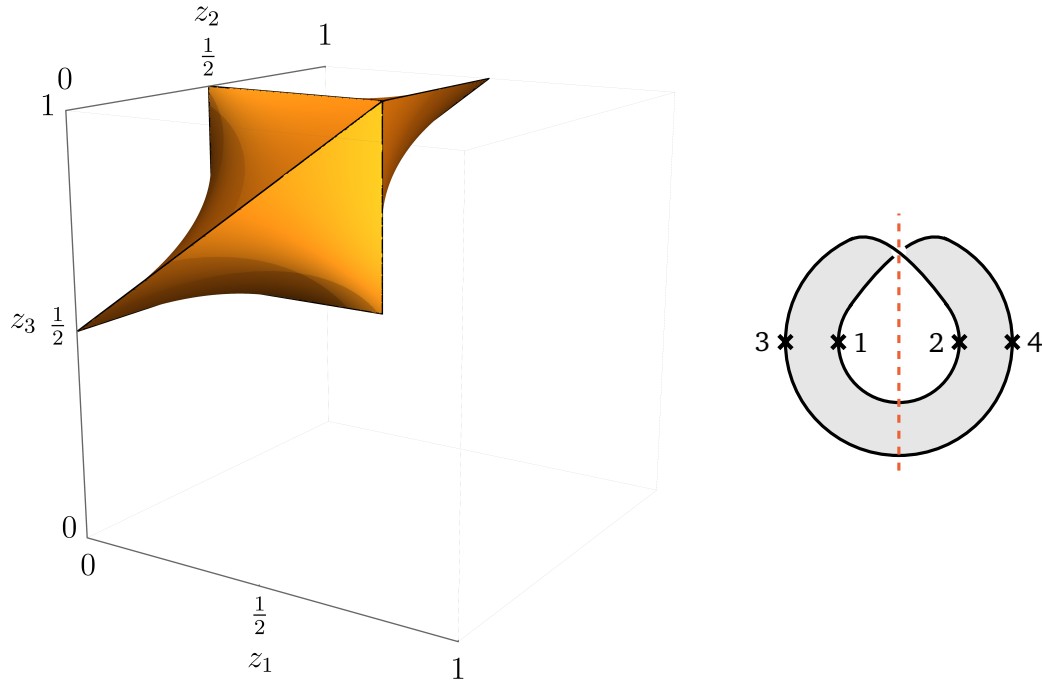

Figure 17: Left: the region in the $(z_1, z_2, z_3)$-space where the exponent Trop $< 0$. The picture is drawn for $s = -\frac{1}{2}$ and $t = -\frac{1}{2}$. Right: the corresponding unique unitarity cut in the $u$-channel.

We integrate in the term $q^{-\frac{1}{4stu}(ut_R - (u+2t)t_L + 2t(t+u)\alpha_R - tu)^2}$ to bring the integral into the following form

$$\text{Im}A_{\text{Möb}}\big|_{u<1} = -\frac{1}{64\sqrt{2stu}} \int_{\longrightarrow} \frac{d\tau}{(-i\tau)^{\frac{3}{2}}} \int_{-u}^{0} dt_L \int_{-u}^{0} dt_R \int_{-\infty}^{\infty} d\alpha_L \int_{-\infty}^{\infty} d\alpha_R$$
$$\times q^{-\alpha_L t_L - \alpha_R t_R - \tilde{P}(t_L, t_R)}(1 + q^{\alpha_L})^{-u}(1 + q^{\alpha_R})^{-u} \qquad (4.58)$$

$$= -\frac{1}{256\pi^2\sqrt{2stu}} \int_{\longrightarrow} \frac{d\tau}{(-i\tau)^{\frac{3}{2}}} \int_{-u}^{0} dt_L \int_{-u}^{0} dt_R\, q^{-\tilde{P}(t_L, t_R)}$$
$$\times \frac{\Gamma(t_L + u)\Gamma(-t_L)\Gamma(t_R + u)\Gamma(-t_R)}{\Gamma(u)^2} \qquad (4.59)$$

$$= -\frac{\pi}{60\sqrt{stu}\,\Gamma(u)^2} \int_{\tilde{P}>0} dt_L\, dt_R\, \tilde{P}(t_L, t_R)^{\frac{5}{2}}$$
$$\times \Gamma(t_L + u)\Gamma(-t_L)\Gamma(t_R + u)\Gamma(-t_R). \qquad (4.60)$$

We changed the integration region as before an used eqs. (4.51) and (4.31). Here,

$$\tilde{P}(t_L, t_R) = -\frac{u(t^2 + t_L^2 + t_R^2 - 2tt_L - 2tt_R - 2t_L t_R) - 4tt_L t_R}{4st}. \qquad (4.61)$$

This is the same as (4.26), except that $s \leftrightarrow u$ have been exchanged. Since the planar annulus topology does not contribute in the $u$-channel, (4.60) is the total contribution to the imaginary part of the planar amplitude, in agreement with (3.74).

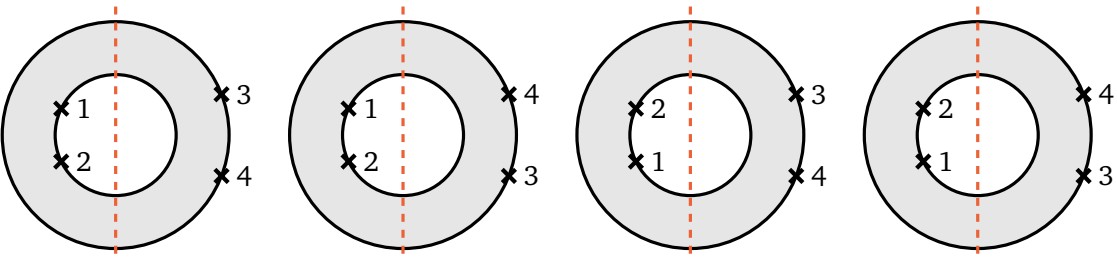

Figure 18: The four cuts of the non-planar annulus diagram in the $s$-channel corresponding to the four regions $\Gamma^{(1)}, \ldots, \Gamma^{(4)}$ in this order.

## 4.4 Non-planar annulus

Proceeding as before, after mapping the contour over the circle $\circlearrowright$ to the horizontal line, the imaginary part of the non-planar amplitude is given by

$$
\text{Im} A_{\text{an}}^{\text{n-p}} = -\frac{1}{64} \int_{\longrightarrow} \frac{d\tau}{\tau^2} \int dz_1\, dz_2\, dz_3\, q^{sz_{41}z_{32}-tz_{21}z_{43}} \left( -\frac{\vartheta_1(z_{21}\tau,\tau)\vartheta_1(z_{43}\tau,\tau)}{\vartheta_2(z_{31}\tau,\tau)\vartheta_2(z_{42}\tau,\tau)} \right)^{-s}
$$
$$
\times \left( \frac{\vartheta_2(z_{41}\tau,\tau)\vartheta_2(z_{32}\tau,\tau)}{\vartheta_2(z_{31}\tau,\tau)\vartheta_2(z_{42}\tau,\tau)} \right)^{-t}. \quad (4.62)
$$

Recall that the integration domain imposes $z_2 - 1 \leqslant z_1 \leqslant z_2$, $0 \leqslant z_2 \leqslant 1$ and $0 \leqslant z_3 \leqslant z_4 = 1$. In addition to the expansion of the $\vartheta_1(z\tau,\tau)$ from (4.18), we also need

$$
\vartheta_2(z\tau,\tau) = \left( q^{\frac{1}{8}-\frac{z}{2}} + q^{\frac{1}{8}+\frac{z}{2}} + q^{\frac{9}{8}-\frac{3z}{2}} + q^{\frac{25}{8}-\frac{5z}{2}} + q^{\frac{9}{8}+\frac{3z}{2}} \right)(1+\mathcal{O}(q)), \quad (4.63)
$$

where only the four terms displayed above can possibly contribute in the $|q| \to 0$ limit since $-1 \leqslant z \leqslant 2$. To be more precise, $z_{21}, z_{43}, z_{42} \in [0,1]$, while $z_{41} \in [0,2]$, $z_{32} \in [-1,1]$ and $z_{31} \in [-1,2]$.

Let us now determine more precisely which regions of the $z_i$-integrals can contribute. The integrand goes as $q^{\text{Trop}}$ with

$$
\text{Trop} = sz_{41}z_{32} - tz_{43}z_{21} + \tfrac{1}{2}s\left(z_{21}+z_{43}\right) + \tfrac{1}{2}t\left( \max(z_{41}, 3z_{41}-1) + |z_{32}| \right)
$$
$$
+ \tfrac{1}{2}u\left( \max(-z_{31}, z_{31}, 3z_{31}-1) + z_{42} \right). \quad (4.64)
$$

Solving for Trop < 0 reveals several disconnected regions where the integrand can diverge.

### 4.4.1 $s$-channel with $s < 1$

We start with the $s$-channel scattering, i.e., $s > 0$ and $t, u < 0$ and work below the first massive threshold, $s < 1$. There are four disconnected regions solving Trop < 0 which correspond to the four different ways that the annulus can be cut in the $s$-channel, see Figure 18. As we already mentioned, the non-planar annulus diagram should be divided by 2 in the end, since it is invariant under orientation reversal. This is manifest in Figure 18, since the first and last as well as the second and third cutting are actually equivalent. They are contained within the chambers

$$
\Gamma^{(1)} = \{0 < z_{41} < 1, \ -1 < z_{32} < 0, \ -1 < z_{31} < 0\}, \quad (4.65a)
$$
$$
\Gamma^{(2)} = \{0 < z_{41} < 1, \ 0 < z_{32} < 1, \ 0 < z_{31} < 1\}, \quad (4.65b)
$$
$$
\Gamma^{(3)} = \{1 < z_{41} < 2, \ -1 < z_{32} < 0, \ 0 < z_{31} < 1\}, \quad (4.65c)
$$
$$
\Gamma^{(4)} = \{1 < z_{41} < 2, \ 0 < z_{32} < 1, \ 1 < z_{31} < 2\}. \quad (4.65d)
$$

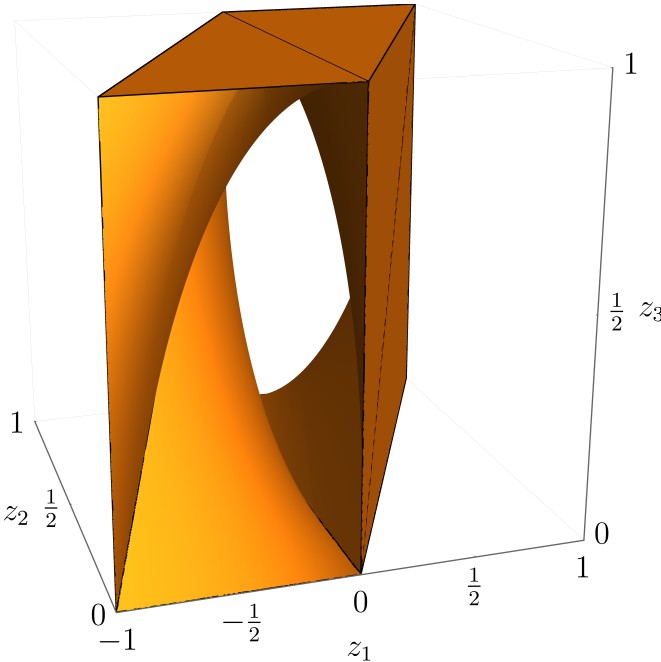

Figure 19: The four regions with Trop $< 0$ for the $s$-channel of the non-planar annulus amplitude.

See Figure 19 for a picture. As is evident from the figure, one can nicely stitch together the four regions by translating $\Gamma^{(1)}$ by the vector $(0,0,1)$, $\Gamma^{(3)}$ by the vector $(1,0,1)$ and $\Gamma^{(4)}$ by the vector $(1,0,0)$ in $(z_1, z_2, z_3)$. Hence we then only get a single integration region $\Gamma$ that takes the form

$$\Gamma = \{-1 < z_{21}, z_{43} < 1, \ 0 < z_{31}, z_{41}, z_{32}, z_{42} < 1\}. \tag{4.66}$$

Dropping again the terms in the expansion of the theta function that do not contribute to the imaginary part yields

$$\mathrm{Im}\, A_{\mathrm{an}}^{\mathrm{n\text{-}p}}\big|_{s<1} = -\frac{1}{64} \int_{\longrightarrow} \frac{\mathrm{d}\tau}{\tau^2} \int_{\Gamma} \mathrm{d}z_1 \, \mathrm{d}z_2 \, \mathrm{d}z_3 \, q^{s(z_{41}-1)z_{32}-tz_{21}z_{43}} |1-q^{z_{21}}|^{-s} |1-q^{z_{43}}|^{-s}. \tag{4.67}$$

Here we use $|\cdot|$ to mean

$$|1-q^{z_{21}}| \equiv \begin{cases} 1-q^{z_{21}}, & z_{21} \geqslant 0, \\ -1+q^{z_{21}}, & z_{21} \leqslant 0, \end{cases} \tag{4.68}$$

and similarly for $|1-q^{z_{43}}|$.

We can recognize that the above expression is identical to (4.22) already encountered in the planar case, except that the integration region is extended to also include negative values of $z_{21}$ and $z_{43}$. Here after the same chain of manipulations, the contribution equals the analog of (4.25), which reads

$$\mathrm{Im}\, A_{\mathrm{an}}^{\mathrm{n\text{-}p}}\big|_{s<1} = \frac{1}{64\sqrt{2stu}} \int_{\longrightarrow} \frac{\mathrm{d}\tau}{(-i\tau)^{\frac{3}{2}}} \int_{-\infty}^{\infty} \mathrm{d}\alpha_{\mathrm{L}} \, \mathrm{d}\alpha_{\mathrm{R}} \int \mathrm{d}t_{\mathrm{L}} \, \mathrm{d}t_{\mathrm{R}} \, q^{-t_{\mathrm{L}}\alpha_{\mathrm{L}}-t_{\mathrm{R}}\alpha_{\mathrm{R}}-P(t_{\mathrm{L}},t_{\mathrm{R}})}$$
$$\times |1-q^{\alpha_{\mathrm{L}}}|^{-s} |1-q^{\alpha_{\mathrm{R}}}|^{-s}. \tag{4.69}$$

Performing the integral over $\alpha_L$, $\alpha_R$ and $\tau$ gives

$$\operatorname{Im} A_{\text{an}}^{\text{n-p}}\Big|_{s<1} = \frac{\pi\Gamma(1-s)^2}{60\sqrt{stu}} \int_{P>0} dt_L \, dt_R \, P(t_L, t_R)^{\frac{5}{2}} \frac{\Gamma(-t_L)}{\Gamma(1-s-t_L)} \frac{\Gamma(-t_R)}{\Gamma(1-s-t_R)}$$
$$\times \left(1 - \frac{\sin(\pi t_L)}{\sin(\pi(s+t_L))}\right)\left(1 - \frac{\sin(\pi t_R)}{\sin(\pi(s+t_R))}\right). \qquad (4.70)$$

This can be further simplified as follows. The polynomial $P(t_L, t_R)$ is invariant under $(t_L, t_R) \to (-s - t_L, -s - t_R)$. Thus

$$\int_{P>0} dt_L \, dt_R \, P(t_L, t_R)^{\frac{5}{2}} \frac{\Gamma(s + t_L)\Gamma(s + t_R)}{\Gamma(1 + t_L)\Gamma(1 + t_R)}$$
$$= \int_{P>0} dt_L \, dt_R \, P(t_L, t_R)^{\frac{5}{2}} \frac{\Gamma(-t_L)\Gamma(-t_R)}{\Gamma(1 - s - t_L)\Gamma(1 - s - t_R)}, \qquad (4.71)$$

and hence

$$\operatorname{Im} A_{\text{an}}^{\text{n-p}}\Big|_{s<1} = \frac{\pi\Gamma(1-s)^2}{30\sqrt{stu}} \int_{P>0} dt_L \, dt_R \, P(t_L, t_R)^{\frac{5}{2}} \frac{\Gamma(-t_L)}{\Gamma(1-s-t_L)} \frac{\Gamma(-t_R)}{\Gamma(1-s-t_R)}$$
$$\times \left(1 - \frac{\sin(\pi t_L)}{\sin(\pi(s+t_L))}\right). \qquad (4.72)$$

### 4.4.2 $u$-channel with $u < 1$

We now move on the $u$-channel kinematics with $u > 0$ and $s, t < 0$. The solutions to Trop $< 0$ are two disconnected regions given by

$$\Gamma^{(1)} = \{0 < z_{41} < 1, \, -1 < z_{32} < 0, \, -1 < z_{31} < 1\}, \qquad (4.73a)$$
$$\Gamma^{(2)} = \{1 < z_{41} < 2, \, 0 < z_{32} < 1, \, 0 < z_{31} < 2\}. \qquad (4.73b)$$

A picture is given in Figure 20. The two regions can again be joined together to the region $\Gamma$ by translating $\Gamma^{(2)}$ by the vector $(1, 1, 0)$ in $(z_1, z_2, z_3)$. We obtain the following formula for $\operatorname{Im} A_{\text{an}}^{\text{n-p}}$ in the $u$-channel:

$$\operatorname{Im} A_{\text{an}}^{\text{n-p}}\Big|_{u<1} = -\frac{1}{64} \int_{\longrightarrow} \frac{d\tau}{\tau^2} \int_{\Gamma} dz_1 \, dz_2 \, dz_3 \, q^{u(1-z_{41})z_{32} - tz_{31}z_{42}} (1 + q^{z_{31}})^{-u} (1 + q^{z_{42}})^{-u}. \qquad (4.74)$$

Up to a simple change of variables and a minus sign, this is the same as (4.56). Thus we can immediately conclude

$$\operatorname{Im} A_{\text{an}}^{\text{n-p}}\Big|_{u<1} = -\operatorname{Im} A_{\text{Möb}}\Big|_{u<1}. \qquad (4.75)$$

It corresponds to the unique unitarity cut displayed in Figure 20.

## 4.5 Summary of the open string

For convenience, let us summarize the full imaginary part of the imaginary part of the worldsheet amplitude. Since we have to sum over color orderings, the final amplitude is crossing symmetric and we can assume to be in the $s$-channel. We work out the result for $s < 1$, but in principle one can repeat the exercise of Section 4.2.2 to extend this to any desired range. We have to sum over different color orderings. In both the planar and non-planar cases, there are three different color orderings. Since the string is unoriented, the reverse color ordering is equivalent and does not need to be counted separately. Thus in the planar case, the color structures are $\operatorname{tr}(t^{a_1} t^{a_2} t^{a_3} t^{a_4})$, $\operatorname{tr}(t^{a_1} t^{a_2} t^{a_4} t^{a_3})$ and $\operatorname{tr}(t^{a_1} t^{a_3} t^{a_2} t^{a_4})$. The second ordering is obtained from the

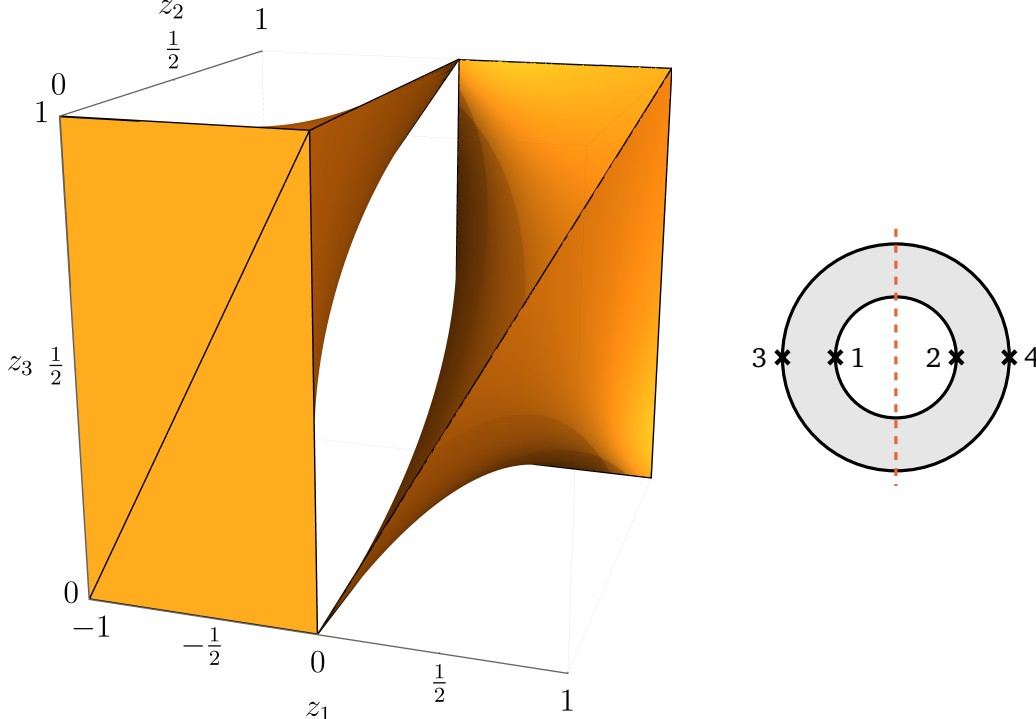

Figure 20: Left: the two regions with Trop $< 0$ for the $u$-channel of the non-planar annulus amplitude. Right: the unique unitarity cut in the $u$-channel of the non-planar annulus. It corresponds to the union of the two regions that fit together after a suitable translation.

first by exchanging $t$ and $u$. The polynomial $P(t_L, t_R)$ has the property that exchanging $t$ with $u$ can be compensated by $t_L \to -s - t_L$. The final color structure corresponds to the exchange of $s$ and $u$ and thus we should use the $u$-channel formula in this case. In the non-planar case the three color structures are $\text{tr}(t^{a_1} t^{a_2}) \text{tr}(t^{a_3} t^{a_4})$, $\text{tr}(t^{a_1} t^{a_3}) \text{tr}(t^{a_2} t^{a_4})$ and $\text{tr}(t^{a_1} t^{a_4}) \text{tr}(t^{a_2} t^{a_3})$. The second is obtained from the first by interchanging $s$ and $u$, while the third is obtained from the first by interchanging $s$ and $t$. For the non-planar annulus, both the $t$ and $u$-channel behave equally. Thus we should use the $u$-channel formula (4.75) for the second and third color structure, but interchange the meaning of $u$ and $s$ or $u$ and $t$, respectively. For the non-planar amplitudes, we also need to include a symmetry factor of $\frac{1}{2}$. This is because orientation reversal of the worldsheet maps the diagram to itself, whereas the color order is reversed in the planar case.

We thus have from eqs. (4.32), (4.54), (4.60), (4.72) and (4.75)

$$
\text{Im } \mathcal{A}_I \Big|_{s<1} = \frac{128\pi^3 g_s^4 t_8 \Gamma(1-s)^2}{15\sqrt{stu}} \int_{P(t_L, t_R)>0} dt_L \, dt_R \, \frac{P(t_L, t_R)^{\frac{5}{2}} \Gamma(-t_L)\Gamma(-t_R)}{\Gamma(1-s-t_L)\Gamma(1-s-t_R)}
$$

$$
\times \Bigg[ \text{tr}(t^{a_1} t^{a_2} t^{a_3} t^{a_4})\left(N - 2 - \frac{2\sin(\pi s)}{\sin(\pi(s+t_L))}\right) - \text{tr}(t^{a_1} t^{a_2} t^{a_4} t^{a_3}) \frac{(N-2)\sin(\pi t_L) + 2\sin(\pi s)}{\sin(\pi(s+t_L))}
$$

$$
- \text{tr}(t^{a_1} t^{a_3} t^{a_2} t^{a_4}) \frac{\sin^2(\pi s)}{\sin(\pi(s+t_L))\sin(\pi(s+t_R))} + \text{tr}(t^{a_1} t^{a_2})\text{tr}(t^{a_3} t^{a_4})\left(1 - \frac{\sin(\pi t_L)}{\sin(\pi(s+t_L))}\right)
$$

$$
+ \frac{(\text{tr}(t^{a_1} t^{a_3})\text{tr}(t^{a_2} t^{a_4}) + \text{tr}(t^{a_1} t^{a_4})\text{tr}(t^{a_2} t^{a_3}))\sin^2(\pi s)}{2\sin(\pi(s+t_L))\sin(\pi(s+t_R))} \Bigg]. \tag{4.76}
$$

This precisely agrees with the sum of (3.13), (3.14) and (3.15) with the explicit results from

unitary cuts given in eqs. (3.73), (3.74), (3.75) and (3.76). Of course we should set $N = 32$ in the string, but we left it free for ease of comparison.

## 4.6 Closed string

Finally, we repeat the same calculation for the type II closed string. As explained earlier, we should evaluate the integral (2.20). Recall that we write $z_i = x_i + \tau y_i$ and $\tau = \tau_x + \tau_y$ with $\tau_x = \text{Re}\,\tau$ and $\tau_y = i\,\text{Im}\,\tau$. The variables will then be complexified again to account for the $i\varepsilon$ prescription.

We call the original moduli space $\mathcal{M}_{1,4} \subset \mathcal{M}_{1,4}^{\mathbb{C}}$ the real slice of the compactification. It corresponds to $\tau_y \in i\mathbb{R}$ and all other variables $x_i$, $y_i$ and $\tau_x$ real. However, the only variable that needs to be genuinely complex is $\tau_y \in \mathbb{H}$. Below, $|X|^2$ will be a shortcut for the holomorphic function in the integration variables $x_i$, $y_i$, $\tau_x$ and $\tau_y$ that coincides with $|X|^2$ on the real slice where $x_i$, $y_i$, $\tau_x \in \mathbb{R}$ and $\tau_y \in i\mathbb{R}$. The strategy is very similar to the open string. We push the imaginary part $\text{Im}\,\tau_y$ to very large values which simplifies the Green's functions

$$\exp(G(z_{ij}, \tau)) = |\vartheta_1(z_{ij}, \tau)|^2 e^{-2\pi \frac{(\text{Im}\,z_{ij})^2}{\text{Im}\,\tau}}\,, \tag{4.77}$$

that enter (2.15a). We have (with $z_{ij} = x_{ij} + \tau y_{ij}$)

$$\exp(G(z_{ij}, \tau)) = |\vartheta_1(z_{ij}, \tau)|^2\, e^{2\pi i \tau_y y_{ij}^2} \tag{4.78}$$
$$= \vartheta_1(x_{ij} + (\tau_x + \tau_y)y_{ij}, \tau_x + \tau_y)$$
$$\times \vartheta_1(-x_{ij} + (-\tau_x + \tau_y)y_{ij}, -\tau_x + \tau_y)\, e^{2\pi i \tau_y y_{ij}^2}\,, \tag{4.79}$$

where the second line spells out the meaning of $|\vartheta_1(z_{ij}, \tau)|^2$ away from the real slice. The leading exponential behaviour as $\text{Im}\,\tau_y \to \infty$ is hence (assuming that $x_{ij} \in [-1, 1]$ and $y_{ij} \in [-1, 1]$)

$$\exp(G(z_{ij}, \tau)) \sim e^{\frac{\pi i}{2}\tau_y - 2\pi i \tau_y |y_{ij}|(1 - |y_{ij}|)}\,. \tag{4.80}$$

According to the general recipe explained in Section 2.3, we define $q = e^{2\pi i \tau_y}$ (which differs from the naive definition as $e^{2\pi i \tau}$). Then the integrand behaves to leading order in a large $\text{Im}\,\tau_y$ limit as $q^{\text{Trop}}$ with

$$\text{Trop} = \sum_{i<j} s_{ij}|y_{ij}|(1 - |y_{ij}|)\,. \tag{4.81}$$

We assume without loss of generality that we are in the $s$-channel. Then only regions in the $y_i$-integration with $\text{Trop} < 0$ can contribute, since the integrand otherwise decays if we take $\text{Im}\,\tau_y \to \infty$. There are two identical such regions that touch along the diagonal, as depicted in Figure 21, where we chose the gauge $y_4 = 0$. Since the bottom and top of the cube are identified, this should actually be thought of as one connected region. To make this manifest, we will allow $y_3$ to be negative. Then the region of positivity is characterized by the condition

$$-\min(1 - y_1, 1 - y_2) < y_3 < \min(y_1, y_2)\,. \tag{4.82}$$

This tells us that

$$y_{41} \leqslant 0\,, \qquad y_{42} \leqslant 0\,, \qquad y_{32} \leqslant 0\,, \qquad y_{31} \leqslant 0 \tag{4.83}$$

everywhere on this region. Both $y_{12}$ and $y_{34}$ have indefinite sign and hence we have to keep two terms in the expansion of the theta-function. This yields the formula

$$\text{Im}\,A_{\text{II}} = \frac{1}{2}\int_{\longrightarrow} \frac{d\tau_y}{\tau_y^2}\int_0^1 d\tau_x \int_{\mathcal{R}} \prod_{i=1}^3 dx_i\, dy_i\, q^{-2sy_1(y_{32}+1)-2ty_{21}(1-y_3)}$$
$$\times \left|1 - e^{2\pi i z_{21}}\right|^{-2s}\left|1 - e^{2\pi i z_{43}}\right|^{-2s}\,, \tag{4.84}$$

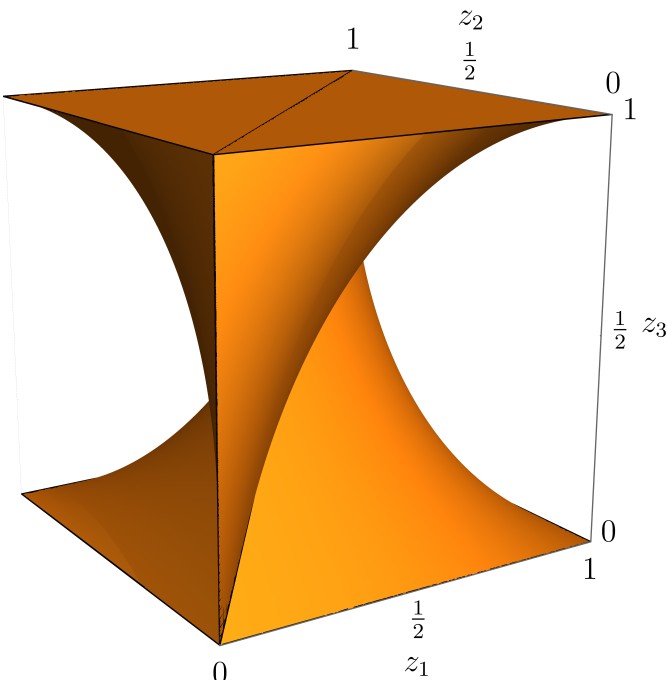

Figure 21: The two regions in the $(y_1, y_2, y_3)$-space where the exponent Trop can be negative. The picture is drawn for $s = 1$ and $t = -\frac{1}{2}$. They should be viewed as one region, since $y_3$ is identified periodically.

where the region $\mathcal{R}$ is characterized by (4.83). We remark that this is very similar to the expression for the annulus, see eq. (4.22). It only depends on $x_{21}$ and $x_{43}$, but not on $x_2$ and $x_1$ separately. Thus we may set $x_1 = x_3 = 0$, say, and only integrate over $x_2$ and $x_4$. We also observe that by definition

$$e^{2\pi i z_{21}} = e^{2\pi i(x_{21} + (\tau_x + \tau_y)y_{21})} = e^{2\pi i((x_{21} + \tau_x y_{21}) + \tau_y y_{21})}, \tag{4.85}$$

and similarly for $e^{2\pi i z_{21}}$. Since $\tau_x$ is real we may hence define $x_\mathrm{L} = x_2 + \tau_x y_{21}$, without affecting the measure. Similarly $x_\mathrm{R} = x_4 + \tau_x y_{43}$. After this, the integrand depends no longer on $\tau_x$ and we may integrate it out. Thus we now have

$$\mathrm{Im}A_\mathrm{II} = -\frac{1}{2} \int_{\longrightarrow} \frac{\mathrm{d}\tau_y}{\tau_y^2} \int_0^1 \mathrm{d}x_\mathrm{L}\,\mathrm{d}x_\mathrm{R} \int_{\mathcal{R}} \prod_{i=1}^3 \mathrm{d}y_i\, q^{-2sy_1(y_{32}+1)-2ty_{21}(1-y_3)}$$
$$\times \left|1 - e^{2\pi i(x_\mathrm{L} + \tau_y y_{21})}\right|^{-2s} \left|1 - e^{2\pi i(x_\mathrm{R} + \tau_y y_{43})}\right|^{-2s}. \tag{4.86}$$

The next steps are very similar to the open string case. We change variables in the $y_i$-integration according to

$$y_1 = \frac{t}{s}y_\mathrm{R} - \frac{t_\mathrm{L}}{s}, \quad y_2 = y_\mathrm{L} + \frac{t}{s}y_\mathrm{R} - \frac{t_\mathrm{L}}{s}, \quad y_3 = -y_\mathrm{R}. \tag{4.87}$$

We also integrate in the factor

$$1 = \sqrt{\frac{si\tau}{t(s+t)}} \int_{-\infty}^{\infty} \mathrm{d}t_\mathrm{R}\, q^{-\frac{1}{2st(s+t)}(st_\mathrm{R} - (s+2t)t_\mathrm{L} + 2t(s+t)y_\mathrm{R} - st)^2}, \tag{4.88}$$

which leads to

$$\text{Im}A_{\text{II}} = \frac{1}{2\sqrt{stu}} \int_{\longrightarrow} \frac{d\tau_y}{(-i\tau_y)^{\frac{3}{2}}} \int_0^1 dx_{\text{L}} dx_{\text{R}} \int_{\mathcal{R}} dt_{\text{L}} dt_{\text{R}} dy_{\text{L}} dy_{\text{R}} \, q^{-2t_{\text{L}}y_{\text{L}}-2t_{\text{R}}y_{\text{R}}-2P(t_{\text{L}},t_{\text{R}})}$$

$$\times \left|1 - e^{2\pi i(x_{\text{L}}+\tau_y y_{\text{L}})}\right|^{-2s} \left|1 - e^{2\pi i(x_{\text{R}}+\tau_y y_{\text{R}})}\right|^{-2s}, \quad (4.89)$$

where $P(t_{\text{L}}, t_{\text{R}})$ is the same polynomial that appeared for the open string and which is given by (4.26). We want to change the integration region $\mathcal{R}$ to $\tilde{\mathcal{R}}$, that is characterized by the inequalities

$$-s \leqslant t_{\text{L}} \leqslant 0, \qquad -s \leqslant t_{\text{R}} \leqslant 0. \quad (4.90)$$

We again need to check that the exponent $-2t_{\text{L}}y_{\text{L}} - 2t_{\text{R}}y_{\text{R}} - 2P(t_{\text{L}}, t_{\text{R}})$ is nowhere negative on the difference of the two regions, which is readily shown. We have the analogue identity to (4.29), which reads

$$\int_{-\infty}^{\infty} dy_{\text{R}} \int_0^1 dx_{\text{R}} \, e^{-4\pi i \tau_y t_{\text{L}} y_{\text{R}}} \left|1 - e^{2\pi i(x_{\text{R}}+\tau_y y_{\text{R}})}\right|^{-2s} \quad (4.91)$$

$$= \frac{i}{\tau_y} \int_{0 < \text{Re}\, z_{\text{R}} < 1} d^2 z_{\text{R}} \, \left|e^{2\pi i z_{\text{R}}}\right|^{-2t_{\text{R}}} \left|1 - e^{2\pi i z_{\text{R}}}\right|^{-2s} \quad (4.92)$$

$$= \frac{i}{(2\pi)^2 \tau_y} \int d^2 \zeta_{\text{R}} \, |\zeta_{\text{R}}|^{-2t_{\text{R}}-2} |1 - \zeta_{\text{R}}|^{-2s} \quad (4.93)$$

$$= -\frac{is^2}{4\pi \tau_y} \frac{\Gamma(-t_{\text{R}})\Gamma(-s)\Gamma(-u_{\text{R}})}{\Gamma(1+s)\Gamma(1+t_{\text{R}})\Gamma(1+u_{\text{R}})}, \quad (4.94)$$

where we set $u_{\text{R}} = -s - t_{\text{R}}$ to make crossing symmetry manifest. From the first to the second line we defined $z_{\text{R}} = x_{\text{R}} + \tau_y y_{\text{R}}$, which leads to an integration region in the complex plane. Thus we finally obtain

$$\text{Im}A_{\text{II}} = \frac{s^4}{32\pi^2\sqrt{stu}} \int_{\longrightarrow} \frac{d\tau_y}{(-i\tau_y)^{\frac{9}{2}}} \int_{-s}^0 dt_{\text{L}} \int_{-s}^0 dt_{\text{R}} \, q^{-2P(t_{\text{L}},t_{\text{R}})}$$

$$\times \frac{\Gamma(-t_{\text{L}})\Gamma(-s)\Gamma(-u_{\text{L}})}{\Gamma(1+s)\Gamma(1+t_{\text{L}})\Gamma(1+u_{\text{L}})} \frac{\Gamma(-t_{\text{R}})\Gamma(-s)\Gamma(-u_{\text{R}})}{\Gamma(1+s)\Gamma(1+t_{\text{R}})\Gamma(1+u_{\text{R}})} \quad (4.95)$$

$$= \frac{16\pi s^4}{15\sqrt{stu}} \int dt_{\text{L}} dt_{\text{R}} \, P(t_{\text{L}}, t_{\text{R}})^{\frac{5}{2}} \frac{\Gamma(-t_{\text{L}})\Gamma(-s)\Gamma(-u_{\text{L}})}{\Gamma(1+s)\Gamma(1+t_{\text{L}})\Gamma(1+u_{\text{L}})}$$

$$\times \frac{\Gamma(-t_{\text{R}})\Gamma(-s)\Gamma(-u_{\text{R}})}{\Gamma(1+s)\Gamma(1+t_{\text{R}})\Gamma(1+u_{\text{R}})}. \quad (4.96)$$

This agrees with the result from unitarity (3.77) when we remember that $\mathcal{A}_{\text{II}} = 2^{-5}\pi^4 t_8 \tilde{t}_8 A_{\text{II}}$.

## 5 Stringy Landau singularities

We now want to explore somewhat more systematically all possible singularities of the amplitude in the complex $(s, t)$-plane, even outside of the physically allowed kinematics. This will recover all the Landau singularities, also known as normal and anomalous thresholds, that one expects the amplitude to have from field-theory considerations.

## 5.1 Landau equation in field theory

Before discussing this topic in string theory, let us recall how to analyze Landau singularities [29–31] in quantum field theory, see, e.g., [24,49] for recent reviews. Here, we focus on the one-loop case, where all Landau singularities can be solved analytically.

To make the analogy as close as possible, we would like to represent the box Feynman integral

$$\mathcal{I}_{\vec{n}}(s,t) = \int \frac{\mathrm{d}^{\mathrm{D}}\ell}{i\pi^{\mathrm{D}/2}} \frac{1}{[\ell^2 + n_1][(\ell+p_1)^2 + n_2][(\ell+p_1+p_2)^2 + n_3][(\ell-p_4)^2 + n_4]} \qquad (5.1)$$

in terms of Schwinger parameters $\alpha_i$ for $i = 1, 2, 3, 4$. Here, $n_i$ are the squared masses of the intermediate propagators, in the conventions of Figure 22. Recall that all external momenta are incoming. Following a standard procedure, Schwinger parameters are introduced by representing each propagator by

$$\frac{1}{q_j^2 + n_j} = i \int_0^\infty \mathrm{d}\alpha_j \, e^{-i(q_j^2 + n_j)\alpha_j} \,. \qquad (5.2)$$

The Feynman $i\varepsilon$ is in principle needed for convergence of this integral, but we will suppress it since for the purposes of identifying positions of possible singularities, it is not important (but can be always restored by replacing $n_j \to n_j - i\varepsilon$). After using (5.2) on every propagator, Wick rotation, integrating out the Gaussian integral in the loop momenta, and rescaling each $\alpha_i \to 2\pi\tau\alpha_i$ while keeping $\sum_{i=1}^4 \alpha_i = 1$, one arrives at

$$\mathcal{I}_{\vec{n}}(s,t) = \frac{1}{(2\pi i)^{\frac{\mathrm{D}-8}{2}}} \int_0^\infty \frac{\mathrm{d}\tau}{\tau^{\frac{\mathrm{D}-6}{2}}} \int_0^\infty \prod_{i=1}^4 \mathrm{d}\alpha_i \, e^{2\pi i\tau \mathcal{V}_{\vec{n}}(\vec{\alpha})} \delta(\textstyle\sum_{i=1}^4 \alpha_i - 1), \qquad (5.3)$$

where

$$\mathcal{V}_{\vec{n}}(\vec{\alpha}) = \frac{s\alpha_1\alpha_3 + t\alpha_2\alpha_4}{\sum_{i=1}^4 \alpha_i} - \sum_{i=1}^4 n_i\alpha_i \,. \qquad (5.4)$$

Note that at this stage we could just integrate out $\tau$, giving rise to a more familiar Schwinger (or Feynman) parametrization of the box integral, but we purposely kept it in the above form to make the connection to string theory more transparent. Inclusion of a numerator in (5.1) would only result in an additional polynomial in the integrand of (5.3) and shifting the powers of $\tau$, but does not affect $\mathcal{V}_{\vec{n}}$. In the worldline formalism, $\mathcal{V}_{\vec{n}}$ is physically interpreted as the on-shell action obtained after localizing the path integral, and the $\alpha_i$'s can be interpreted as the moduli of a given graph topology.

Landau analysis simply asks when the above integral can lead to branch cuts. A divergence can only happen when $\mathcal{V}_{\vec{n}} = 0$, since then the $\tau$ integral has a singularity (either at 0 or $\infty$), and moreover one cannot deform the $\alpha_i$-contour to avoid this singularity. This happens for specific values of $\alpha_i = \alpha_i^*$ for which

$$\alpha_i^* \, \partial_{\alpha_i} \mathcal{V}_{\vec{n}}(\vec{\alpha}^*) = 0, \qquad (5.5)$$

for $i = 1, 2, 3, 4$. The branch $\partial_{\alpha_i} \mathcal{V}_{\vec{n}}$ corresponds to the contour being pinched by roots of $\mathcal{V}_{\vec{n}}$ (pinch singularity), while $\alpha_i^* = 0$ comes from the fact that the endpoint of the contour needs to stay fixed (endpoint singularity). Of course, both can happen at the same time. The conditions (5.5) are known as the Landau equations. One also distinguishes between *leading* and *subleading* Landau singularities, where all or only a subset of the $\partial_{\alpha_i} \mathcal{V}_{\vec{n}}$ conditions are imposed respectively. Another special case are *second-type* singularities, which happen when

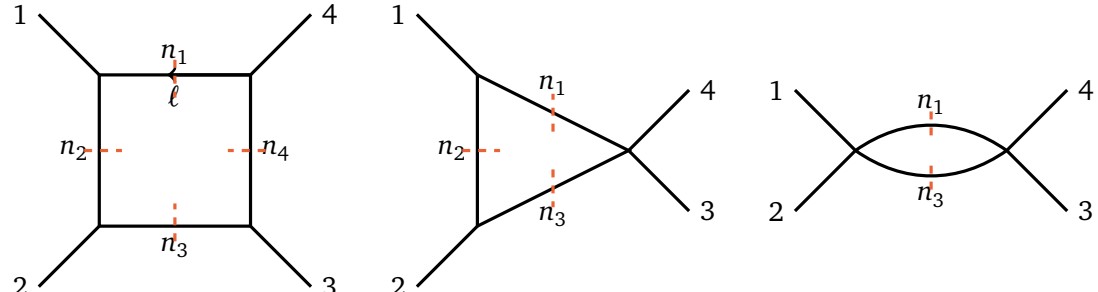

Figure 22: Field theory diagrams leading to Landau singularities in the string amplitude. The numbers $n_i \in \mathbb{Z}_{\geqslant 0}$ label the mass square of the corresponding particle in the field theory. Only the normal thresholds (right) are present in physical regions, and only the normal and box anomalous thresholds (left) appear on the physical sheet.

$\sum_{i=1}^{4} \alpha_i^* = 0$. Because of homogeneity, the above conditions automatically imply vanishing of $\mathcal{V}_{\vec{n}}$ because

$$\mathcal{V}_{\vec{n}}(\vec{\alpha}^*) = \sum_{i=1}^{4} \alpha_i^* \, \partial_{\alpha_i}, \mathcal{V}_{\vec{n}}(\vec{\alpha}^*) = 0 \, . \tag{5.6}$$

Because of homogeneity, only 3 out of 4 Schwinger parameters are fixed by (5.5). To be more precise, one should treat $(\alpha_1 : \alpha_2 : \alpha_3 : \alpha_4) \in \mathbb{CP}^3$ as being defined only projectively and $\sum_{i=1}^{4} \alpha_i = 1$ is a particular fixing of the projective invariance. At any rate, this leaves us with one net constraint imposed by (5.5), which puts one condition on the *external* kinematics $(s, t)$.

Another interpretation of Landau equations (5.5) is as determining classical saddle points of the action $\mathcal{V}_{\vec{n}}$ (either in the bulk of the $\alpha_i$-integration or on its boundaries). In other words, Landau singularities appear for special kinematic points where the diagram can be realized as a *classical* scattering process. This is in perfect agreement with the loop-momentum space formulation of Landau equations, in which one requires that all the internal momenta are on their mass shell, $q_i^2 = -n_i$, and that the sum of space-time displacements $\alpha_i q_i$ around the loop sums to zero, $\sum_{i=1}^{4} \alpha_i q_i = 0$. Since in order to get to (5.3), we had to localize the loop momentum on a Gaussian saddle, we can actually reconstruct its value on the solution of Landau equations:

$$\ell^* = -\frac{p_1(\alpha_2^* + \alpha_3^* + \alpha_4^*) + p_2(\alpha_3^* + \alpha_4^*) + p_3\alpha_4^*}{\sum_{i=1}^{4} \alpha_i^*} \, . \tag{5.7}$$

For example, from here one sees that second-type singularities correspond to potential divergences at infinite loop momentum.

The above physical interpretation automatically tells us that for a $2 \to 2$ scattering of massless states, only normal thresholds (exchange of two intermediate particles) can lie within a physical region, since any other configuration would require at least one kinematically disallowed vertex.

At first glance, it might appear as if we only need to be concerned with solutions for which $\alpha_i^* \geqslant 0$, since this defines the integration contour. However, it might also happen that the contour needs to be first deformed in the complex directions to avoid a potential divergence before encountering a pinch singularity somewhere in the complex planes. Hence, to classify all potential singularities, one needs to allow for complex $\alpha$'s. At one loop, it is known that singularities on the physical sheet (connected to physical kinematics without crossing branch cuts) only occur when $\alpha_i \geqslant 0$ [78], and hence we will pay attention to those in particular.

**Box anomalous threshold.** The equations (5.5) are actually very simple to solve. It is convenient to first organize the kinematics into the $4 \times 4$ symmetric matrix

$$\mathbf{Y}_{\vec{n}} = - \begin{bmatrix} 2n_1 & n_1+n_2 & n_1+n_3-s & n_1+n_4 \\ n_1+n_2 & 2n_2 & n_2+n_3 & n_2+n_4-t \\ n_1+n_3-s & n_2+n_3 & 2n_3 & n_3+n_4 \\ n_1+n_4 & n_2+n_4-t & n_3+n_4 & 2n_4 \end{bmatrix}, \tag{5.8}$$

such that

$$\mathcal{V}_{\vec{n}}(\vec{\alpha}) = \frac{\frac{1}{2}\vec{\alpha}^{\intercal}\mathbf{Y}_{\vec{n}}\vec{\alpha}}{\sum_{i=1}^4 \alpha_i}. \tag{5.9}$$

Assuming that $\sum_{i=1}^4 \alpha_i^* \neq 0$, the leading Landau equations can be therefore restated as the matrix equation $\mathbf{Y}_{\vec{n}}\vec{\alpha}^* = 0$, which admits a solution if only if

$$\det \mathbf{Y}_{\vec{n}} = 0. \tag{5.10}$$

This are the positions of the box anomalous thresholds with a particular assignment of the internal masses $\vec{n}$. Let us initially assume that all $n_i$ are strictly positive, since massless cases warrant a more careful discussion due to IR divergences.

Explicitly, the box anomalous threshold is located at

$$\det \mathbf{Y}_{\vec{n}} = s^2 t^2 - 2(n_2+n_4)s^2 t - 2(n_1+n_3)st^2 + (n_2-n_4)^2 s^2 + (n_1-n_3)^2 t^2 \tag{5.11}$$
$$+ 2[n_2(n_3-2n_4) + n_3 n_4 + n_1(n_2-2n_3+n_4)]st = 0.$$

The Schwinger parameters on this solution are the null vector of $\mathbf{Y}_{\vec{n}}$ on the support of (5.10), explicitly:

$$\vec{\alpha}^* = \begin{bmatrix} st(n_2+n_3) - [s(n_2-n_4) + t(n_1-n_3)](n_2-n_3) \\ t(n_1-n_3)^2 + s(s(t-n_4) + n_2(n_3-2n_4-s) + n_3(n_4-2t) + n_1(n_2+n_4-2(n_3+t))) \\ st(n_1+n_2) + [s(n_2-n_4) - t(n_1-n_3)](n_1-n_2) \\ 2s[sn_2 - (n_1-n_2)(n_2-n_3)] \end{bmatrix}. \tag{5.12}$$

Recall that $\vec{\alpha}^*$ only make sense up to an overall rescaling. Note that (5.10) generically has multiple branches intersecting the real kinematic space $(s, t) \in \mathbb{R}^2$, but not all of them have positive $\alpha$'s. One can check that for any choice of the masses $\vec{n}$, the only solutions with $\alpha_i^* > 0$, i.e., those on the physical sheet, have to lie in the quadrant $s, t \geqslant 0$, corresponding to unphysical kinematics (recall that physically-allowed kinematics must satisfy $stu > 0$), see Figure 23 for an example.

**Triangle anomalous threshold.** Similarly, we can determine positions of the triangle subleading singularities by not varying one Schwinger parameter and instead setting it to zero. Without loss of generality, let us set $\alpha_4^* = 0$, see Figure 22. Repeating previous steps gives

$$\det \mathbf{Y}_{[4]}^{[4]} = 2s[sn_2 - (n_1-n_2)(n_2-n_3)] = 0, \tag{5.13}$$

where the notation means we remove the column and row 4 before computing the determinant. The corresponding $\alpha$'s are given by

$$\vec{\alpha}^* = \begin{bmatrix} -2sn_2 + (n_1-n_2)(n_2-n_3) \\ s(n_1+n_2) - (n_1-n_2)(n_1-n_3) \\ (n_1-n_2)^2 \\ 0 \end{bmatrix}, \tag{5.14}$$

up to projective invariance. It is not difficult to convince oneself that there are no solutions with positive $\alpha$'s.

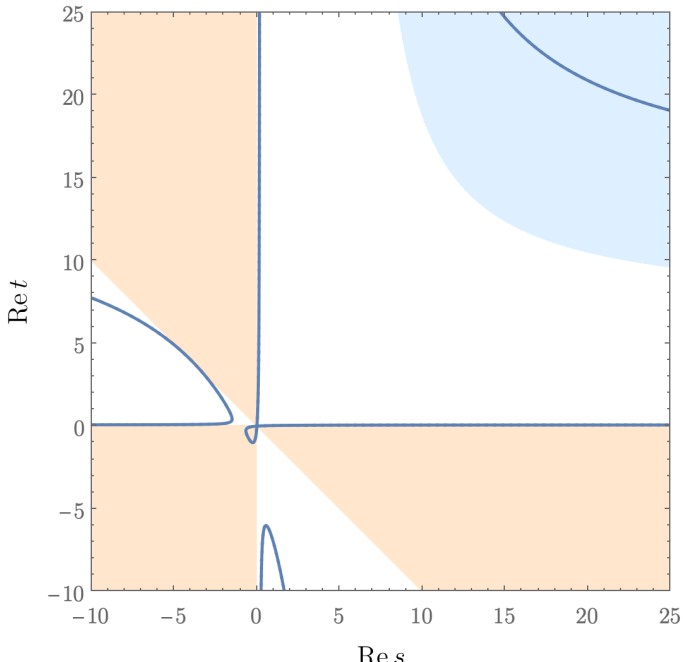

Figure 23: Four branches of the box Landau singularity (5.10) with $\vec{n} = (1, 3, 2, 4)$ in blue. Physical regions are shown in orange, while the region carved out by imposing positivity of $\vec{\alpha}^*$ from (5.12) is contained in $s, t \geqslant 0$ and plotted in blue. This illustrates that only one branch of the box anomalous threshold lies on the physical sheet, though not in any physical region.

**Normal and pseudo-normal thresholds.** Finally, we are have singularities corresponding to pairs of Schwinger parameters set to zero, e.g., $\alpha_2^* = \alpha_4^* = 0$. This gives

$$\det \mathbf{Y}_{[24]}^{[24]} = -\left(s - (\sqrt{n_1} + \sqrt{n_3})^2\right)\left(s - (\sqrt{n_1} - \sqrt{n_3})^2\right) = 0. \tag{5.15}$$

The first and second branch give the normal and pseudo-normal thresholds respectively. The corresponding $\alpha$'s are

$$\vec{\alpha}^* = \begin{bmatrix} s - n_1 - n_3 \\ 0 \\ 2n_1 \\ 0 \end{bmatrix}, \tag{5.16}$$

from which we see that only the normal thresholds are on the physical sheet.

**IR and Second-type singularities.** We can now return back to the cases with massless particles. Every time there are at least two adjacent vanishing masses,

$$n_i = n_{i+1} = 0, \tag{5.17}$$

the Feynman integral (5.3) has a soft/collinear divergence in $D \leqslant 4$. At the level of Landau singularities, they correspond are codimension-0 in the kinematic space, i.e., they are present regardless of the value of the external kinematics (likewise, one encounters codimension-2 singularities at $s = t = 0$, where $\alpha$'s lie on manifolds of saddles). Since here we focus on $D = 10$, we do not need to delve further into the subtle discussion of IR divergences.

Last but not least, there are second-type singularities corresponding the particular solutions where $\sum_{i=1}^4 \alpha_i^* = 0$. One can show that all of them are contained in the set $stu = 0$.

## 5.2 Planar annulus

Let us return to the imaginary part of the planar annulus four-point function given by (4.17). It gives a definition of the imaginary part of the amplitude for *all* real values of $s$ and $t$, regardless of whether they lie in the kinematically allowed regime. We will assume that the imaginary part that is defined in this way corresponds to the imaginary part of the physical sheet.

For complex kinematics, the integral (4.17) does not converge, since it is either exponentially growing or decaying for large values of $\mathrm{Re}\,\tau$. However, (4.17) can be easily analytically continued, which rotates the $\tau$-contour in the complex plane depending on the phase of Trop. We will in the following focus on real kinematics, which already captures all possible (anomalous) thresholds.

Since we can make $q$ in the equation arbitrarily small, it is again a good idea to consider the $q$-expansion of the integrand. Here, care has to be taken since $z_{ij}$ can possibly become small in the integration region and hence terms such as $q^{z_{ij}}$ in the expansion are not well-behaved. Thus, we should first start by studying again the leading exponent as $q \to 0$, i.e., the tropicalization of the integrand,

$$\mathrm{Trop} = -s(1 - z_{41})z_{32} - t z_{21} z_{43}\,. \tag{5.18}$$

In the following it will be useful to introduce the analogues of the field-theory Schwinger parameters. Let us set

$$\alpha_1 = 1 - z_{41}\,, \qquad \alpha_2 = z_{21}\,, \qquad \alpha_3 = z_{32}\,, \qquad \alpha_4 = z_{43}\,. \tag{5.19}$$

They are of course not independent and satisfy $\sum_{i=1}^{4} \alpha_i = 1$. In these variables

$$\mathrm{Trop} = -\frac{s\alpha_1\alpha_3 + t\alpha_2\alpha_4}{\sum_{i=1}^{4} \alpha_i}\,. \tag{5.20}$$

In general, the amplitude will have a singularity when a new term in the $q$-expansion has a non-zero region in the Schwinger parameter space where the exponent is negative. This will first happen at a particular point in $\vec{\alpha} = (\alpha_1, \alpha_2, \alpha_3, \alpha_4)$, which we call $\vec{\alpha}^*$. For a term $T$ in the $q$-expansion, we have an associated tropicalization of the integrand $\mathrm{Trop}_T$ for $q \to 0$. Using the constraint $\sum_{i=1}^{4} \alpha_i = 1$, we can always homogenize $\mathrm{Trop}_T$ and assume that it is a homogeneous rational function in the $\alpha$'s, just like (5.4).

For a region forming in the interior of the integration region, i.e., for $\alpha_i^* \geqslant 0$, we have

$$\partial_{\alpha_i} \mathrm{Trop}_T(\vec{\alpha}^*) = 0\,. \tag{5.21}$$

Using the assumed homogeneity, this also implies that

$$\mathrm{Trop}_T(\vec{\alpha}^*) = \sum_{i=1}^{4} \alpha_i^* \, \partial_{\alpha_i} \mathrm{Trop}_T(\vec{\alpha}^*) = 0\,. \tag{5.22}$$

If $\vec{\alpha}^*$ lies instead on the boundary of the integration region, we only need to require that the derivatives along the boundary vanish. Following QFT terminology, we call these equations the leading and subleading Landau equations, respectively. The bottom line is, one recovers exactly the same conditions for the position of possible discontinuities of the amplitude as in Section 5.1.

**Leading Landau singularities.** Let us first consider the case where the singularity originates from a point $\vec{\alpha}^*$ in the interior. In this case, all the $\alpha$'s are bounded from below by the distance of the critical point to the boundary of the integration region. We may thus simply $q$-expand

the theta-functions. We are interested what exponents of $q$ appear in this expansion. It is easy to see that the exponents appearing in the expansion of the theta function are

$$\left(-i q^{\frac{\alpha_1}{2}-\frac{1}{8}} \vartheta_1(\alpha_1, \tau)\right)^{-s} = \sum_{n_+, n_- \in \mathbb{Z}_{\geqslant 0}} a(n_+, n_-, s) q^{n_+ \alpha_1 + n_-(1-\alpha_1)}, \qquad (5.23)$$

where $a(n_+, n_-, s)$ is a polynomial in $s$. It can then be seen that we can parametrize the exponents that appear in a expansion of the integrand as

$$\mathrm{Trop}_{n_1, n_2, n_3, n_4} = -\frac{s\alpha_1\alpha_3 + t\alpha_2\alpha_4}{\sum_{i=1}^4 \alpha_i} + \sum_{i=1}^4 n_i \alpha_i. \qquad (5.24)$$

At this stage, the problem is equivalent to that studied in Section 5.1 for any choice of the integers $n_i \in \mathbb{Z}_{\geqslant 0}$. In particular, we find that the planar amplitude has an infinite number of anomalous thresholds lying on the physical sheet, but for unphysical kinematics $s, t \geqslant 0$. This makes the analytic structure somewhat complicated, see Figure 24.

Let us discuss further the branch of (5.11) lying on the physical sheet. Let us notice that the coefficient in (5.11) of $t^2$ is

$$s^2 - 2(n_1 + n_3)s + (n_1 - n_3)^2 = \left[s - (\sqrt{n_1} + \sqrt{n_3})^2\right]\left[s - (\sqrt{n_1} - \sqrt{n_3})^2\right], \qquad (5.25)$$

and similarly for the coefficient of $s^2$ with $(s, n_1, n_3) \to (t, n_2, n_4)$. The two zeros correspond to the normal and pseudo-normal threshold of the bubble diagram and will be discussed below. We observe that for large $t$, we can drop the terms of order $t$ and 1 and only have to keep the $t^2$ term in (5.11). This tells us that the Landau singularity on the physical sheet asymptotes $s = (\sqrt{n_1} + \sqrt{n_3})^2$ for large $t$ and $t = (\sqrt{n_2} + \sqrt{n_4})^2$ for large $s$.

In particular, the Landau singularity is contained in the region

$$(\sqrt{n_1} + \sqrt{n_3})^2 \leqslant s < \infty, \qquad (\sqrt{n_2} + \sqrt{n_4})^2 \leqslant t < \infty. \qquad (5.26)$$

This means that even though there are infinitely many Landau singularities labelled by $n_1$, $n_2$, $n_3$ and $n_4$, they can never accumulate at a finite point. There are accumulation points at infinity, e.g., since there is a $\mathbb{Z}^2$-worth of curves asymptoting to $s = (\sqrt{n_1} + \sqrt{n_3})^2$. For example all Landau singularities on the physical sheet in the region $0 \leqslant s, t < 16$ are plotted in Figure 24.

**Subleading Landau singularities.** One can easily repeat the same analysis for subleading singularities: the triangle anomalous thresholds, as well as normal and pseudo-normal thresholds. Just as in QFT, only the normal thresholds at

$$s = (\sqrt{n_1} + \sqrt{n_3})^2, \qquad t = (\sqrt{n_2} + \sqrt{n_4})^2 \qquad (5.27)$$

appear on the physical sheet and in the physical regions, see Figure 24.

## 5.3 Other diagrams and closed string

Let us very briefly comment on the other cases that we have not discussed here. Their analysis is very similar. In general, one considers the different terms $\mathrm{Trop}_T$ in the general expansion (2.13). They are functions of the other moduli, in this case the positions $z_i$ on the worldsheet (as well as $\mathrm{Re}\,\tau$ for the closed string). For the closed string, $\mathrm{Re}\,\tau$ as well as $\mathrm{Re}\,z_i$ are irrelevant for the analysis because they do not influence the real part of $\mathrm{Trop}_T$. They can in fact always be removed, similar to what we described in Section 4.6. Thus $\mathrm{Trop}_T$ is always a function of three other moduli, which can be identified with the Schwinger parameters $\alpha_i$ subject to the constraint $\sum_{i=1}^4 \alpha_i = 1$. One then homogenizes $\mathrm{Trop}_T$ in $\alpha_i$ and solves the Landau equations (5.21), as well as the subleading Landau equations. The results are in line with the field-theory expectations, i.e., one finds a family of normal and anomalous thresholds in all allowed channels.

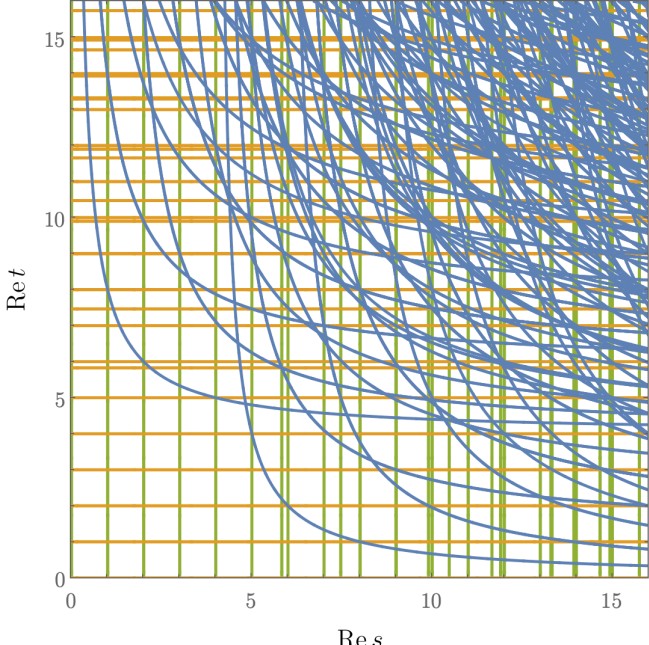

Figure 24: All Landau singularities of the planar amplitude in the region $0 \leqslant s, t < 16$: normal thresholds in the $s$- and $t$-channel (green and orange), as well as box anomalous thresholds (blue).

# 6 Physical properties of the imaginary parts

In this section we study properties of the imaginary parts of amplitudes derived in Section 4. For concreteness, we mostly focus on the planar annulus contribution. If need be, analogous analysis can be repeated for other topologies with almost identical steps.

## 6.1 Convergent integral representation

To summarize the results of Section 4, we found that the imaginary part of the planar annulus amplitude admits the following integral representation:

$$\mathrm{Im}A_{\mathrm{an}}^{\mathrm{p}} = \frac{\pi N}{60} \frac{\Gamma(1-s)^2}{\sqrt{stu}} \sum_{n_1 \geqslant n_2 \geqslant 0} \theta\big[s - (\sqrt{n_1}+\sqrt{n_2})^2\big] \int_{P_{n_1,n_2}>0} dt_{\mathrm{L}} \, dt_{\mathrm{R}} \, P_{n_1,n_2}(t_{\mathrm{L}}, t_{\mathrm{R}})^{\frac{5}{2}}$$
$$\times Q_{n_1,n_2}(t_{\mathrm{L}}, t_{\mathrm{R}}) \frac{\Gamma(-t_{\mathrm{L}})\Gamma(-t_{\mathrm{R}})}{\Gamma(n_1+n_2+1-s-t_{\mathrm{L}})\Gamma(n_1+n_2+1-s-t_{\mathrm{R}})}, \quad (6.1)$$

where $N = 32$ and $P_{n_1,n_2}$ is the following ratio of Gram determinants evaluated on the cut:

$$P_{n_1,n_2}(t_{\mathrm{L}}, t_{\mathrm{R}}) = -\frac{\det \mathcal{G}_{p_1 p_2 p_3 \ell}}{\det \mathcal{G}_{p_1 p_2 p_3}}\bigg|_{\ell^2=-n_1, (p_1+p_2+\ell)^2=-n_2} \quad (6.2)$$

$$= -\frac{1}{4stu} \det \begin{bmatrix} 0 & s & u & n_2-s-t_{\mathrm{L}} \\ s & 0 & t & t_{\mathrm{L}}-n_1 \\ u & t & 0 & n_1-t_{\mathrm{R}} \\ n_2-s-t_{\mathrm{L}} & t_{\mathrm{L}}-n_1 & n_1-t_{\mathrm{R}} & 2n_1 \end{bmatrix}. \quad (6.3)$$

One can show that the region of integration, defined by $P_{n_1,n_2} > 0$ is an ellipse contained in the region $-s+n_1+n_2 \leqslant t_{\mathrm{L,R}} \leqslant 0$ above the threshold $s > (\sqrt{n_1}+\sqrt{n_2})^2$ for a given term. Above,

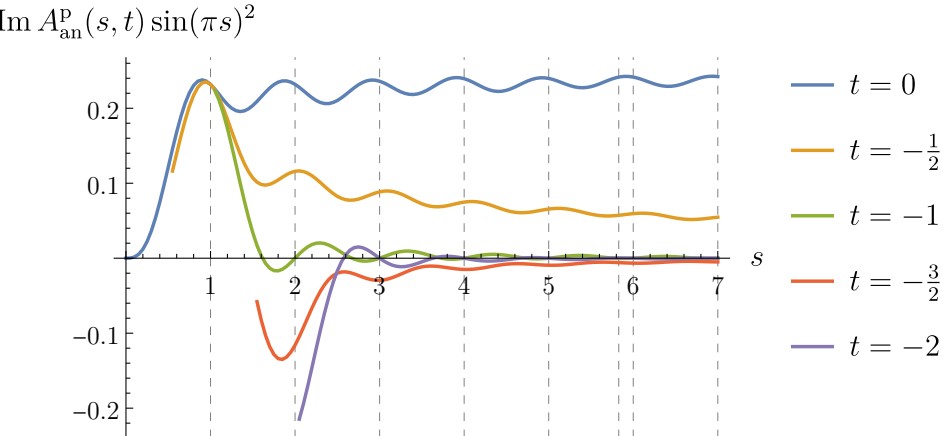

Figure 25: Plot of $\mathrm{Im}\,A_{\mathrm{an}}^{\mathrm{p}}(s,t)$ for few momentum transfers $t$ in the $s$-channel, $s > -t$. We multiply the function by $\sin(\pi s)^2$ in order to remove double poles at every positive integer $s$. Dashed vertical lines indicate the values of $s$ at which a new threshold opens up.

$Q_{n_1,n_2}$ are polynomials in all the kinematic variables, which encode the intricate patterns of the string interactions allowed at a given mass level. The first couple of them are

$$Q_{0,0}(t_L, t_R) = 1\,, \tag{6.4a}$$

$$Q_{1,0}(t_L, t_R) = 2(-2st_L t_R - s^2 t_L + st_L - s^2 t_R + st_R + s^2 t - 2st + t)\,. \tag{6.4b}$$

The next polynomials for $(\sqrt{n_1} + \sqrt{n_2})^2 < 7$ can be found in the ancillary file `Q.txt`.

Recall that it is often convenient to change the integration variables to $(x, y)$ given by

$$t_{L/R} = \frac{\sqrt{\Delta_{n_1,n_2}}}{2\sqrt{s}}\left(\sqrt{-u}\,x \pm \sqrt{-t}\,y\right) + \frac{1}{2}(n_1 + n_2 - s)\,, \tag{6.5}$$

where

$$\Delta_{n_1,n_2} = \left[s - (\sqrt{n_1} + \sqrt{n_2})^2\right]\left[s - (\sqrt{n_1} - \sqrt{n_2})^2\right]\,. \tag{6.6}$$

With this substitution, we simply have $P_{n_1,n_2} = \frac{\Delta_{n_1,n_2}}{4s}(1 - x^2 - y^2)$, and the measure of each integral in (6.1) becomes

$$\frac{\pi N}{60}\frac{\Gamma(1-s)^2}{\sqrt{stu}}\int_{P_{n_1,n_2} > 0} \mathrm{d}t_L\,\mathrm{d}t_R\,P_{n_1,n_2}(t_L, t_R)^{\frac{5}{2}}(\cdots)$$

$$= \frac{\pi N}{3840}\frac{\Gamma(1-s)^2}{s^4}\Delta_{n_1,n_2}^{\frac{7}{2}}\int_{0 < x^2 + y^2 < 1} \mathrm{d}x\,\mathrm{d}y\,(1 - x^2 - y^2)^{\frac{5}{2}}(\cdots)\,, \tag{6.7}$$

with the rest of the integrand evaluated on the support of (6.5).

The advantage of the above form is that, in contrast with the moduli space integrals, (6.1) is manifestly convergent. This is because the only potential singularity of the integrand can be a simple pole coming from the Gamma functions $\Gamma(-t_{L/R})$. Since $\{t_L = 0\}$ and $\{t_R = 0\}$ touches the ellipse $P(t_L, t_R) \geqslant 0$ only at the boundary, the integral still converges. In particular, it means that (6.1) can be evaluated numerically without the need for additional analytic continuation. A plot of this function in the $s$-channel kinematics was already given in Figure 3. To make this result a bit more readable, we plot a few fixed-$t$ slices in Figure 25 and fixed-angle in Figure 26.

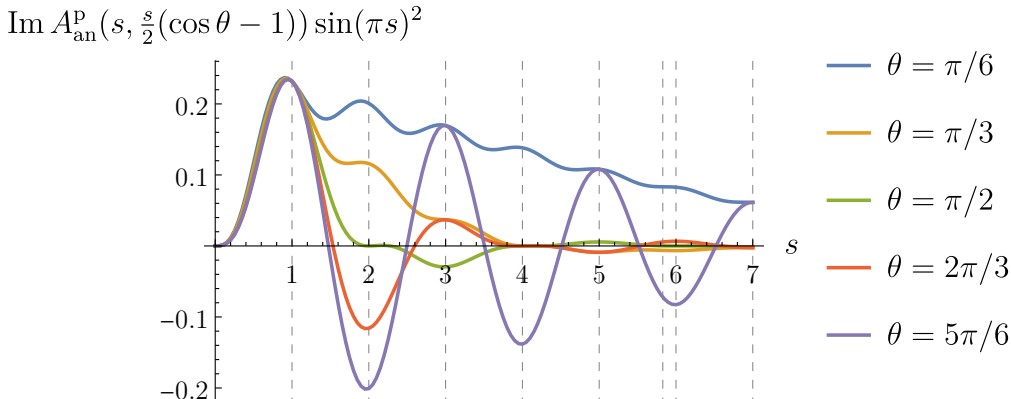

Figure 26: Plot of $\mathrm{Im}A_{\mathrm{an}}^{\mathrm{p}}(s,t)$ for few valued of scattering angles $\theta$ related to Mandelstam invariants by $\cos\theta = 1 + 2t/s$, multiplied by $\sin(\pi s)^2$.

## 6.2 Forward limit and total cross-section

In the forward limit, $t \to 0$, the variables (3.68) both degenerate to

$$t_{\mathrm{L}} = t_{\mathrm{R}} = \frac{1}{2}\left(\sqrt{\Delta_{n_1,n_2}}\, x + n_1 + n_2 - s\right). \tag{6.8}$$

In other words, the phase space becomes one-dimensional and is parametrized by a single scattering angle between $p_2$ and $\ell$, equal to that between $-\ell$ and $p_3$. In particular, it means the integral loses the $y$-dependence, which can be now easily integrated out. The measure in (3.71) becomes

$$\int_{0<x^2+y^2<1} \mathrm{d}x\,\mathrm{d}y\,(1-x^2-y^2)^{\frac{D-5}{2}}(\cdots) \overset{t\to 0}{=} \sqrt{\pi}\frac{\Gamma(\frac{D-3}{2})}{\Gamma(\frac{D-2}{2})}\int_{-1}^{1}(1-x^2)^{\frac{D-4}{2}}(\cdots), \tag{6.9}$$

where the prefactor on the right-hand side equals $\frac{5}{16}\pi$ in D = 10. Alternatively, in terms of the original integration variables $(t_{\mathrm{L}}, t_{\mathrm{R}})$, we have

$$P_{n_1,n_2}(t_{\mathrm{L}},t_{\mathrm{R}})^{\frac{D-5}{2}} \overset{t\to 0}{=} 2\sqrt{\pi}\sqrt{-t}\frac{\Gamma(\frac{D-3}{2})}{\Gamma(\frac{D-2}{2})}P_{n_1,n_2}(t_{\mathrm{L}},t_{\mathrm{L}})^{\frac{D-4}{2}}\delta(t_{\mathrm{L}}-t_{\mathrm{R}}). \tag{6.10}$$

Note that $\sqrt{-t}$ cancels out the square root in the prefactor, so the amplitude remains finite in the forward limit.

In addition, we also need to analyze the behavior of the polarization-dependent prefactor $t_8$, defined in (3.9). Recall that we can realize the forward limit by going to the frame

$$(p_3,\epsilon_3,t^{a_3}) = (-p_2,\overline{\epsilon}_2,t^{a_2}), \qquad (p_4,\epsilon_4,t^{a_4}) = (-p_1,\overline{\epsilon}_1,t^{a_1}), \tag{6.11}$$

where the additional minus signs and complex conjugation arises because of our conventions in which every particle is incoming. In this limit, we encounter contractions of the type $(F_i\overline{F_i})^{\mu\nu} = p_i^\mu p_i^\nu \epsilon_i \cdot \overline{\epsilon}_i$, and almost all terms vanish or cancel out, except for

$$\begin{aligned} t_8(1234) &\overset{t\to 0}{=} \mathrm{tr}_v(F_1 F_2 \overline{F_2 F_1}) + \mathrm{tr}_v(F_1 \overline{F_2} F_2 \overline{F_1}) \\ &= \tfrac{1}{2}s^2 \epsilon_1 \cdot \overline{\epsilon}_1\, \epsilon_2 \cdot \overline{\epsilon}_2\,. \end{aligned} \tag{6.12}$$

The significance of the forward limit is that, appropriately normalized, it computes the one-loop total cross-section for scattering of two gluons as

$$\sigma_{gg}^{\mathrm{tot}} \propto \frac{1}{s}\mathrm{Im}\,\mathcal{A}_{\mathrm{I}}\big|_{t=0} \propto g_s^4 s\,\mathrm{Im}A_{\mathrm{I}}\big|_{t=0}, \tag{6.13}$$

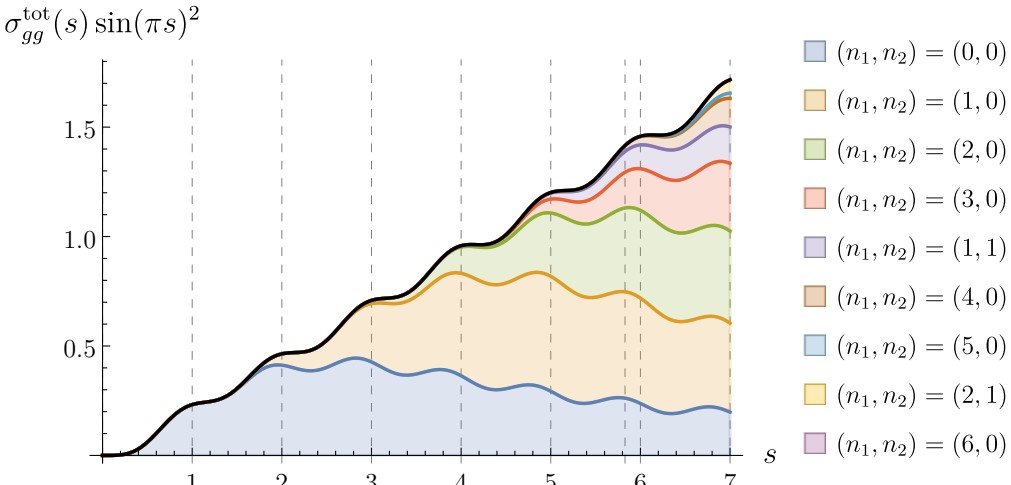

Figure 27: Total cross section of the planar annulus amplitude, proportional to $s \operatorname{Im} A_{\text{an}}^{\text{p}}$, normalized by $\sin(\pi s)^2$ in order to remove double poles (black). We set the polarization and color dependence to 1. Contributions from the individual thresholds are denoted with colors.

where in the second transition we used (6.12). Previous discussions of string-theory cross sections include [79–81]. Here, we focus on the contribution coming from the planar annulus. We plot this quantity, normalized by $\sin(\pi s)^2$ in order to remove double poles, in Figure 27. Note that it contain an additional factor of $s$ compared to Figures 25 and 26.

An interesting feature is that each threshold opens up very slowly. We explain this in Section 6.6. At the energies plotted in Figure 27, the cross-section appears to be roughly linear in $s$. Nevertheless, we caution the reader to not extrapolate this behavior to high energies, since this would contradict the standard lore of the Regge behavior. We elaborate on this point in Section 7.

## 6.3 Decay widths

As cross check of our analysis, we computed the double residues of the imaginary parts of the amplitude directly as in Section 4.1. We can use the formal identity

$$\operatorname*{Res}_{s=n} z^{-s} = \frac{\pi}{(n-1)!} \delta^{(n-1)}(z), \tag{6.14}$$

to localize the $z_1$ and $z_3$ integral. We already used it implicitly for $n=1$ in (4.5). Following similar steps gives

$$\operatorname*{DRes}_{s=n} \operatorname{Im} A_{\text{an}}^{\text{p}} = -\frac{\pi^2}{2((n-1)!)^2} \int_{\mathcal{G}} d\tau \, dz_1 \, dz_2 \, dz_3 \, \delta^{(n-1)}(z_{21}) \delta^{(n-1)}(z_{43}) \vartheta_1(z_{32}, \tau)^{-t}$$

$$\times \left( \frac{z_{21} z_{43}}{\vartheta_1(z_{21}, \tau) \vartheta_1(z_{43}, \tau)} \right)^n \vartheta_1(z_{41}, \tau)^{-t} \vartheta_1(z_{31}, \tau)^{n+t} \vartheta_1(z_{42}, \tau)^{n+t} \tag{6.15}$$

$$= -\frac{\pi^2}{2((n-1)!)^2} \int_{\mathcal{G}} d\tau \, dz_2 \, \partial_{z_1}^{n-1} \partial_{z_3}^{n-1} \Big|_{\substack{z_1=z_2 \\ z_3=1}} \left[ \left( \frac{z_{21} z_{43}}{\vartheta_1(z_{21}, \tau) \vartheta_1(z_{43}, \tau)} \right)^n \right.$$

$$\left. \times \vartheta_1(z_{31}, \tau)^{n+t} \vartheta_1(z_{42}, \tau)^{n+t} \vartheta_1(z_{32}, \tau)^{-t} \vartheta_1(z_{41}, \tau)^{-t} \right]. \tag{6.16}$$

The result is in general a polynomial of order $n-1$ in $t$. The $t$-dependence captures the spin dependence of the intermediate states. We could decompose the double residue in terms of

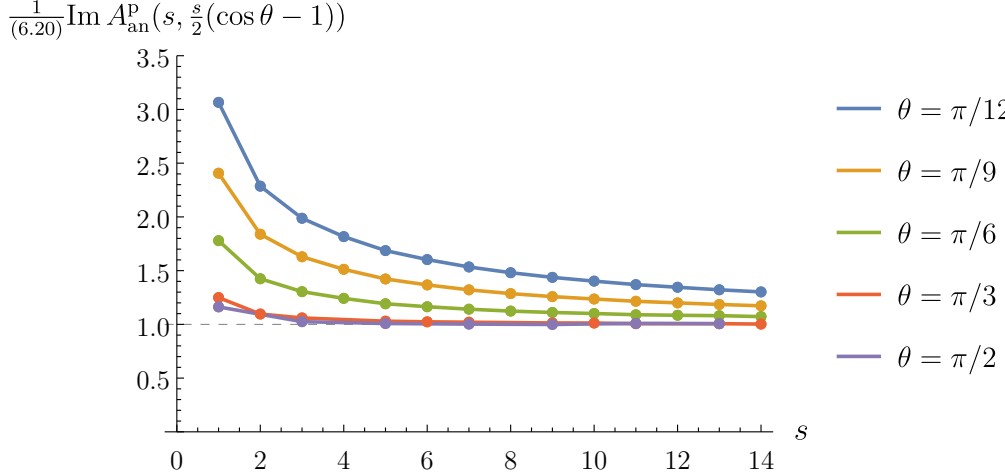

Figure 28: Ratio between $\mathrm{Im}A_{\mathrm{an}}^{\mathrm{p}}(s, \frac{s}{2}(\cos\theta - 1))$ at fixed-angles $\theta$ extracted from decay widths at integer $s = n$, and the analytic prediction from the right-hand side of (6.20). The ratio approaching 1 at higher and higher energies indicates exponential suppression of the imaginary part as $\sim e^{-S_{\mathrm{tree}}}$.

Gegenbauer polynomials, which would lead to the decay widths of the states with fixed spin. For example, the coefficient of the maximal exponent $t^{n-1}$ represents states on the leading Regge trajectory and we have in particular the following simple formula

$$\left.\mathrm{DRes}_{s=n} \mathrm{Im}A_{\mathrm{an}}^{\mathrm{p}}\right|_{t^{n-1}} = -\frac{\pi^2}{2((n-1)!)^2} \int_{\Omega} \mathrm{d}\tau \int_0^1 \mathrm{d}z \left(\frac{\vartheta_1(z,\tau)}{2\pi\eta(\tau)^3}\right)^{2n} \left(-\partial_z^2 \log\vartheta_1(z,\tau)\right)^{n-1}. \quad (6.17)$$

Except for the different choice of contour, this formula appeared before in [82]. It is difficult to evaluate this integral for arbitrary $n$ in closed form.

We have computed the double residues up to $n = 14$. They are tabulated in Appendix A and ancillary file `decaywidths.txt`. We observe that they are numerically very close to the expression

$$\mathrm{DRes}_{s=n} \mathrm{Im}A_{\mathrm{an}}^{\mathrm{p}} \sim \frac{1}{4\pi^2} \frac{\Gamma(t+n)}{\Gamma(t+1)\Gamma(n)}. \quad (6.18)$$

In the cases $n = 1, 2, 3$, the $t$-dependence of this formula is exact (though the prefactors are different from $\frac{1}{4\pi^2}$), but for higher values one finds small deviations. For example

$$\mathrm{DRes}_{s=14} \mathrm{Im}A_{\mathrm{an}}^{\mathrm{p}} \sim 3.98795 \cdot 10^{-12}(t + 1.00010)(t + 2.00070)(t + 3.00185)(t + 4.00077)$$
$$\times (t + 4.99986)(t + 5.99963)(t + 7)(t + 8.00037)(t + 9.00014)$$
$$\times (t + 9.99923)(t + 10.99815)(t + 11.99930)(t + 12.99990). \quad (6.19)$$

## 6.4 High-energy fixed-angle limit

We can use the decay widths as a proxy for studying the behavior of the imaginary parts in various limits. The advantage of doing so is that the computation of decay widths is much simpler than the full $\mathrm{Im}A_{\mathrm{an}}^{\mathrm{p}}$, but it already captures some of its features.

In particular, let us use this method to read-off the behavior in the fixed-angle high-energy limit in the $s$-channel, $s \to \infty$, $t \to -\infty$ with the ratio $s/t$ fixed. Since we expect the amplitude to be exponentially-suppressed, the asymptotic behavior should be reached even for reasonably low energies (in contrast with the Regge limit). As a matter of fact, we can already use the

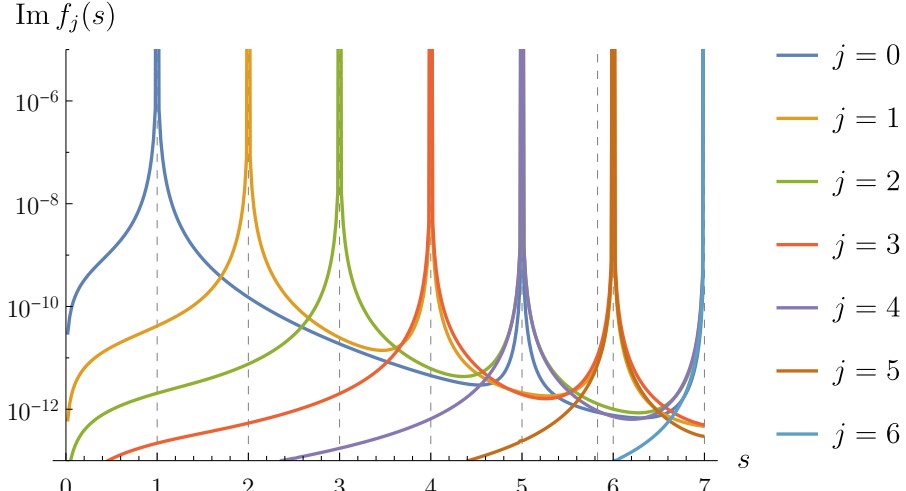

Figure 29: Imaginary parts of the partial wave coefficients $f_j(s)$ for first few spins $j$. We observe that spins up to $j$ always dominate the imaginary part up to $s \lesssim j+1$. A given $f_j(s)$ has a double pole peak at a positive integer $s = n$ only if $n+j$ is odd, with the exception of $(n, j) = (3, 0)$.

empirical formula (6.18) to find at leading orders

$$\text{Im}A_{\text{an}}^{\text{p}}(s, t) \sim -\sqrt{\frac{s}{8\pi t u}} \frac{\sin(\pi t)}{\sin(\pi s)^2} e^{-S_{\text{tree}}} \quad \text{as} \quad s, -t \to \infty, \tag{6.20}$$

where $S_{\text{tree}}$ is the tree-level on-shell action

$$S_{\text{tree}} = s \log(s) + t \log(-t) + u \log(-u). \tag{6.21}$$

The argument in the exponent matches the form predicted by Gross and Manes [32]. We can verify this behavior numerically by plotting the ratio between the imaginary part of the amplitude and the approximation (6.20) at a few scattering angles in Figure 28. Up to $n = 14$, we find that the ratio approaches 1, thus verifying the expected behavior (6.20).

## 6.5 Low-spin dominance

The imaginary part of the partial wave coefficients $f_j(s)$ for exchange of spin $j$ can be now easily extracted. In the conventions of [83] we have

$$\text{Im} f_j(s) = c_{\text{D}}'' \int_{-1}^{1} dz \, (1-z^2)^{\frac{D-4}{2}} G_j^{(\text{D})}(z) \, \text{Im}A_{\text{an}}^{\text{p}}(s, t(z)), \tag{6.22}$$

where $z = 1 + 2t/s$ is the cosine of the scattering angle, $c_{10}'' = \frac{1}{786432\pi^4}$ is a normalization constant, and $G_j^{(\text{D})}(z) = {}_2F_1(-j, j+D-3, \frac{D-2}{2}; \frac{1-z}{2})$ are the Gegenbauer polynomials in D dimensions.

Values of $\text{Im} f_j$ for $j \leqslant 6$ as a function of the energy $s$ are plotted in Figure 29. Note that this time we do not normalize by $\sin(\pi s)^2$ in order to see which spins dominate more clearly. Recall that unitarity implies that $\text{Im} f_j(s) \geqslant 0$. We observe an interesting pattern we will refer to as *low-spin dominance*.

From Figure 29 we can read-off that the scalar (spin $j = 0$) contributing dominates over contributions from all the other spins (note the logarithmic scale) all the way up to $s \lesssim 1$, where it has a double pole. Similarly, going to higher and higher energies, only low spin contributions give an appreciable contribution to $\text{Im} f_j(s)$. More concretely, keeping spins up to $j$ seems to be

enough until $s \lesssim j+1$ in the range of energies plotted in the above figure. A version of low-spin dominance was previously observed in field theory and tree-level string scattering, where only spin-0 dominates across all energies [35, 36], see also [37].

The pattern of divergences can be easily explained. First of all, simple poles are absent because they would give rise to negative contributions to $\text{Im} f_j(s)$, thus violating unitarity. Since we already know the imaginary part of the amplitude has at most double-poles and (6.22) cannot introduce new singularities, $\text{Im} f_j(s)$ simply inherits the double poles at $s = n$ visible in Figure 29.

The double poles are absent whenever $n+j$ is even. This follow simply from the symmetry under relabelling $t \to -n-t$ and simultaneously $z_3 \leftrightarrow z_4$ (before fixing $z_4$) in (6.15). Under this exchange, (6.15) changes by a factor of $(-1)^{n+1}$. Further, Gegenbauer polynomials under the same flip, translating to $z \to -z$, satisfy $G_j^{(D)}(z) = (-1)^j G_j^{(D)}(-z)$. Hence the integrand of (6.22) has parity $(-1)^{n+j+1}$ in $z$, meaning that all the coefficients of double poles at $s = n$ vanish if $n+j$ is even. In addition, we find that the double pole at $(n, j) = (3, 0)$ also vanishes in the space-time dimension $D = 10$, which can be verified by the direct computation given in (A.3). Because of the approximate behavior of double residues from (6.18), the presence of double poles at a given spin follows the same pattern as the tree-level partial wave coefficients $B_{n,j}^D$ encountered in [2].

It is expected that a similar pattern will persist at higher genera and their resummation will dampen the peaks in Figure 29 into Breit–Wigner distributions. It is presently not clear if this dampening will preserve the low-spin dominance, since a self-consistent computation would require a resummation of all-genus amplitudes.

## 6.6 Behavior near thresholds

Let us now study the rate at which a contribution from each new threshold at $s = (\sqrt{n_1} + \sqrt{n_2})^2$ opens up. This rate is related to the explicit prefactor $\Delta_{n_1,n_2}^{\frac{7}{2}}$ already pulled out of the integrand in the representation (6.7).

We first note that if $n_1 = 0$ and/or $n_2 = 0$, the corresponding $\Delta_{n_1,n_2}$ contains two powers of the threshold-expansion variable, e.g.,

$$\Delta_{n_1,0} = (s - n_1)^2 . \tag{6.23}$$

Otherwise, when both $n_1$ and $n_2$ are non-zero, we have only one:

$$\Delta_{n_1,n_2} = 4\sqrt{n_1 n_2}\left[s - (\sqrt{n_1} + \sqrt{n_2})^2\right] . \tag{6.24}$$

Moreover, the explicit prefactor $\Gamma(1-s)^2$ in (6.1) contains double poles at every positive integer $s$, which modifies the discussion when either $n_1 = 0$ or $n_2 = 0$, but not both. Finally, in the form (3.71) also contains an additional factor of $s^{\frac{2-D}{2}} = s^{-4}$, which is important near the $s = 0$ threshold. The bottom line is that we will have to consider these cases separately.

In addition to the prefactors, we also need to know the behavior of the integrand near the thresholds, which is inherited through the $s$-dependence of $t_{L/R}$ as in (3.68). Near the threshold, these momentum transfers are given by

$$t_L = t_R = -\sqrt{n_1 n_2} . \tag{6.25}$$

Since the tree-level amplitudes entering (6.1) are evaluated in their physical regimes, $-s \leqslant t_{L,R} \leqslant 0$, the integrand remains bounded close to almost all thresholds, except those with $n_1 = 0$ and/or $n_2 = 0$, where the integrand develops a simple or double pole because of the Gamma functions in (6.1). Near those places, one also needs to analyse the polynomials $Q_{n_1,n_2}$ which can have additional zeros.

From the explicit results for $Q_{n_1,n_2}$, we found that within their domain of validity, they only have zeros when $n_1 > 0$ and $n_2 = 0$ (recall we set $n_1 \geqslant n_1$), precisely at the value of $s = n_1$ where this threshold opens up. For example,

$$Q_{1,0}(t_L, t_R)\big|_{s=1} = -4t_L t_R, \tag{6.26a}$$

$$Q_{2,0}(t_L, t_R)\big|_{s=2} = 4t_L t_R(3t_L t_R - t - 1), \tag{6.26b}$$

$$Q_{3,0}(t_L, t_R)\big|_{s=3} = -8t_L t_R\left(-6t t_L t_R + 5t_L^2 t_R^2 - 9t_L t_R + t_L^2 + t_R^2 + t^2 + 3t + 2\right), \tag{6.26c}$$

but the polynomials $Q_{1,1}$ and $Q_{2,1}$ is finite:

$$Q_{1,1}(-1,-1)\big|_{s=4} = 4(4 + 3t)(8 + 3t), \tag{6.27}$$

as well as

$$Q_{2,1}(-\sqrt{2},-\sqrt{2})\big|_{s=(\sqrt{2}+1)^2} = 16\big((86 + 60\sqrt{2})t^3 + (747 + 528\sqrt{2})t^2$$
$$+ (1983 + 1402\sqrt{2})t + 2(775 + 548\sqrt{2})\big). \tag{6.28}$$

In particular, all the polynomials $Q_{n_1,n_2}$ remain finite at any higher threshold. We will assume that this pattern extends to higher $(n_1, n_2)$.

Let us first spell out the "generic" case when neither $n_1$ nor $n_2$ are zero and $(\sqrt{n_1} + \sqrt{n_2})^2$ is not an integer, since it can be stated in the simplest way. In this case, the integrand of each contribution is independent of $x$ and $y$, which means we can simply integrate them out using

$$\int_{0<x^2+y^2<1} dx\, dy\, (1-x^2-y^2)^{\frac{D-5}{2}} = \frac{2\pi}{D-3}, \tag{6.29}$$

and put $t_L$ and $t_R$ to their threshold value (6.25), giving

$$\text{Im}A_{\text{an}}^{\text{p}}\big|_{s=(\sqrt{n_1}+\sqrt{n_2})^2} = \frac{\pi^2 N(n_1 n_2)^{\frac{7}{4}}\Gamma(1-(\sqrt{n_1}+\sqrt{n_2})^2)^2\Gamma(\sqrt{n_1 n_2})^2}{105(\sqrt{n_1}+\sqrt{n_2})^8\Gamma(1-\sqrt{n_1 n_2})^2}$$
$$\times Q_{n_1,n_2}(-\sqrt{n_1 n_2}, -\sqrt{n_1 n_2})\big[s - (\sqrt{n_1}+\sqrt{n_2})^2\big]^{\frac{7}{2}} + \text{reg} + \dots, \tag{6.30}$$

where $Q_{n_1,n_2}$ is evaluated at the threshold and we only display the leading contribution from the new threshold, which exists on top of the regular terms coming from lower thresholds.

In the case when neither of $n_1$ and $n_2$ are zero, but $(\sqrt{n_1}+\sqrt{n_2})^2$ is an integer (implying that also $\sqrt{n_1 n_2}$ is an integer), one needs to take more care with the fact that the terms $\Gamma(1-s)^2$ and $\Gamma(n_1+n_2+1-s-t_{L/R})$ give additional poles and zeros (the factors $\Gamma(-t_{L/R})$ and $Q_{n_1,n_2}$ remain finite). The first gives an additional double pole enhancement compared to the previous case. To analyze the effect of Gamma functions in the denominators, we need to expand $t_{L/R}$ to subleading order:

$$t_{L/R} = -\sqrt{n_1 n_2} + c_{n_1,n_2}^{L/R}(x, y)\big[s - (\sqrt{n_1}+\sqrt{n_2})^2\big]^{\frac{1}{2}} + \dots, \tag{6.31}$$

where

$$c_{n_1,n_2}^{L/R}(x, y) = \frac{2(n_1 n_2)^{\frac{1}{4}}}{\sqrt{n_1}+\sqrt{n_2}}\left(\sqrt{(\sqrt{n_1}+\sqrt{n_2})^2 + t}\, x \pm \sqrt{-t}\, y\right). \tag{6.32}$$

This tells us that each $\Gamma(n_1+n_2+1-s-t_{L/R})$ gives a square root suppression of the threshold behavior. The overall leading behavior is therefore enhanced by a single power of the expansion parameter compared to the previous case and we have

$$\text{Im}A_{\text{an}}^{\text{p}}\big|_{s=(\sqrt{n_1}+\sqrt{n_2})^2} \propto \big[s - (\sqrt{n_1}+\sqrt{n_2})^2\big]^{\frac{5}{2}} + \text{reg} + \dots. \tag{6.33}$$

Next we consider the case when $n_2 = 0$ but $n_1 \neq 0$. Recall from (6.23) that there is an overall $\Delta_{n_1,0}^{\frac{7}{2}} = (s-n_1)^7$ already pulled out of the integral and the factor of $\Gamma(1-s)^2$ gives an additional double pole. To analyze the integrand, we also need to expand $t_{L/R}$ to the next order:

$$t_{L/R} = \frac{s-n_1}{2\sqrt{n_1}}\left(\sqrt{t+n_1}x \pm \sqrt{-t}y - \sqrt{n_1}\right) + \dots,\tag{6.34}$$

which in particular means that $t_{L/R} \to 0$ as we approach the threshold. The Gamma functions $\Gamma(-t_{L/R})$ therefore give a simple pole each, which is cancelled by a simple zero of the polynomials $Q_{n_1,0}$. The factors $\Gamma(n_1+n_2+1-s-t_{L/R})$ approach one and hence do not influence the power counting. The total leading behavior is thus

$$\mathrm{Im}A_{\mathrm{an}}^{\mathrm{p}}\big|_{s=n_1} \propto (s-n_1)^5 + \mathrm{reg} + \dots.\tag{6.35}$$

Finally, we are left with the $(n_1, n_2) = (0,0)$ term, which is a bit special because in approaching $s \to 0^+$ one needs to also approach $t \to 0^-$ to remain in the physical region, so the answer depends on how we approach the threshold. For example, fixing the forward limit $t = 0$ first, we find that the imaginary part goes as a single power of $s$ (as worked out in (3.52)). Alternatively, at any non-zero $t$, the leading behavior is $s^2$. The $\alpha' \to 0$ limit (i.e., both $s$ and $t$ taken to be small, with their ratio fixed) will be given in (6.51).

## 6.7 Low-energy expansion

In order to evaluate integrals of the type (6.1) in practice in the low-energy expansion, it is useful to make a further change of variables:

$$(x,y) = \sqrt{1-r}\left(\frac{z+z^{-1}}{2}, \frac{z-z^{-1}}{2i}\right),\tag{6.36}$$

with the Jacobian $\frac{1}{2iz}$. The integration over the unit disk here is translated to that over its radius $\sqrt{1-r}$ and $z = e^{i\theta}$ related to the angular coordinate $\theta$. Hence, after the change of variables, the integration runs over $r \in [0,1]$ and unit circle in $z$. We also have

$$(1-x^2-y^2)^{\frac{D-5}{2}} = r^{\frac{D-5}{2}}.\tag{6.37}$$

In general, starting from the expression (3.71) we thus have

$$\mathrm{Im}\,\mathcal{A} = \frac{(\pi\Delta_{n_1,n_2})^{\frac{D-3}{2}}}{2i(4s)^{\frac{D-2}{2}}\Gamma(\frac{D-3}{2})}\sum_{\substack{\text{species}\\\text{colors}\\\text{polarizations}}}\int_0^1 \mathrm{d}r\, r^{\frac{D-5}{2}}\oint_{|z|=1}\frac{\mathrm{d}z}{z}\,\mathcal{A}_0^{\mathrm{L}}(s,t_{\mathrm{L}})\mathcal{A}_0^{\mathrm{R}}(s,t_{\mathrm{R}}).\tag{6.38}$$

At this stage, we can perform the $z$ integral by picking up all the residues enclosed by the unit circle.

Since we are interested in the low-energy expansion, only the massless cuts $(n_1, n_2) = (0,0)$ can contribute and we specialize to this case now. In terms of the original variables $(t_{\mathrm{L}}, t_{\mathrm{R}})$, we have

$$t_{L/R} = \frac{\sqrt{s}\sqrt{1-r}}{4}\left[\sqrt{-u}(z+z^{-1}) \mp i\sqrt{-t}(z-z^{-1})\right] - \frac{s}{2},\tag{6.39}$$

and $\Delta_{0,0} = s^2$, so the prefactor in (6.38) is $-\frac{i}{960}\pi^3 s^3$. In the case at hand, the amplitudes $\mathcal{A}^{\mathrm{L/R}}$ are Veneziano amplitudes, which admit simple low-energy expansion. For instance, we have (see, e.g., [84]):

$$\frac{\Gamma(1-s)\Gamma(-t_{\mathrm{L}})}{\Gamma(1-s-t_{\mathrm{L}})} = -\frac{1}{t_{\mathrm{L}}}\exp\left[\sum_{k=2}^{\infty}\frac{\zeta_k}{k}\left(s^k + t_{\mathrm{L}}^k - (s+t_{\mathrm{L}})^k\right)\right].\tag{6.40}$$

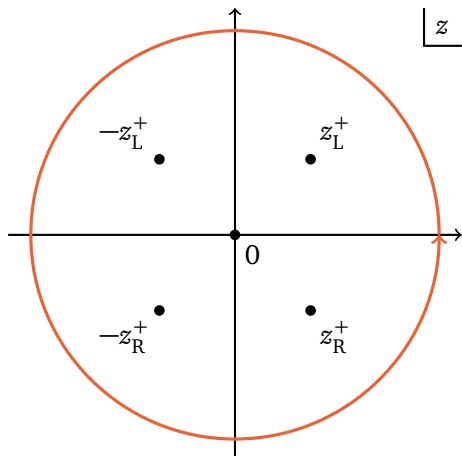

Figure 30: The five singularities contributing to the evaluation of the phase-space integral at $z = (z_L^+, z_R^+, -z_L^+, -z_R^+, 0)$, responsible respectively for: massless $t_L$-, $t_R$-, $u_L$-, and $u_R$-channel exchanges, as well as the contact terms. The red contour is at $|z| = 1$. If the $\mathcal{A}_0^{L/R}$ had massive poles or branch cuts, they would similarly show up within the unit disk.

The strategy will therefore be to expand the integrand in $\alpha'$ and evaluate each term explicitly. Since at each order, the integrand will be a rational function of the kinematic invariants, $t_L$, $t_R$, $u_L$, and $u_R$, it suffices to know how to take residues around the values of $z$ corresponding to those poles and arond $z = 0$.

The roots of $t_L = 0$ and $t_R = 0$ are given by, respectively

$$z_L^\pm = \frac{\sqrt{s}}{\sqrt{-u} - i\sqrt{-t}}\left(\frac{1 - \sqrt{r}}{1 + \sqrt{r}}\right)^{\pm\frac{1}{2}}, \qquad z_R^\pm = \frac{\sqrt{s}}{\sqrt{-u} + i\sqrt{-t}}\left(\frac{1 - \sqrt{r}}{1 + \sqrt{r}}\right)^{\pm\frac{1}{2}}, \qquad (6.41)$$

which satisfy

$$|z_L^\pm| = |z_R^\pm| = \left(\frac{1 - \sqrt{r}}{1 + \sqrt{r}}\right)^{\pm\frac{1}{2}}. \qquad (6.42)$$

This means only $z_{L,R}^+$ are enclosed by the contour given that $0 \leqslant r \leqslant 1$. Similarly, for the $u_L = 0$ and $u_R = 0$ singularities give rise to poles at $z = -z_{L,R}^\pm$ and only $-z_{L/R}^+$ lie within the unit circle. Finally, residues around $z = 0$ can be picked up when the integrand has contact (rational) terms in the kinematics. We illustrate it in Figure 30.

For example, in the low-energy expansion of the imaginary part of the planar annulus amplitude, we only encounter linear combinations of the integrals

$$\int_0^1 dr\, r^{\frac{5}{2}} \oint_{|z|=1} \frac{dz}{z}\, t_L^a t_R^b \bigg|_{(6.39)}, \qquad (6.43)$$

where $a$ and $b$ are integers bigger or equal to $-1$. We first focus on the contributions with $a, b \geqslant 0$, i.e., contact interactions (or cuts of bubble diagrams). It is easy to show that they can only lead to polynomial terms in $s$ and $t$. To see this, let us call $t_L = \beta z + \overline{\beta}/z + \gamma$ and $t_R = \overline{\beta}z + \beta/z + \gamma$ in (6.39) as a shorthand. The integral is invariant under simultaneous replacements $(\beta, z) \to (-\beta, -z)$ as well as $(\beta, z) \to (\overline{\beta}, 1/z)$. This in particular means that the result of the $z$-residue has to be a polynomial generated by the invariants

$$|\beta|^2 = \tfrac{1}{16}(1 - r)s^2, \qquad \beta^2 + \overline{\beta}^2 = \tfrac{1}{8}(1 - r)s(s + 2t). \qquad (6.44)$$

We therefore end up with a polynomial in $s$, $t$ and integrals of the form $\int_0^1 dr\, r^{n/2}$ for $n \geqslant 5$ which are finite. Similar analysis can be repeated for cases with either $a = -1$ or $b = -1$, i.e., cuts of triangles, where the result of (6.43) is still rational in $s$ and $t$, but can have a simple pole in $s$ (which cancels with the prefactor in (6.1)). The case $(a, b) = (-1, -1)$, corresponding to cuts of boxes, is qualitatively different because it can produce logarithms. We study it more closely in the following section.

### 6.7.1 Field-theory example

Let us illustrate the above manipulations on the case of the scalar massless box Feynman integral in D dimensions. Here, we are tasked with computing

$$\text{Im}\,\mathcal{I}_{\text{box}}(s, t) = \frac{s^{\frac{D-4}{2}}}{2\pi} \int_{0 < x^2 + y^2 < 1} \frac{dx\, dy\, (1 - x^2 - y^2)^{\frac{D-5}{2}}}{t_{\text{L}} t_{\text{R}}}, \tag{6.45}$$

where the overall normalization is chosen for later convenience. After changing to $(r, z)$ we have

$$\text{Im}\,\mathcal{I}_{\text{box}}(s, t) = \frac{1}{2} s^{\frac{D-4}{2}} \int_0^1 dr\, r^{\frac{D-5}{2}} \frac{1}{2\pi i} \oint_{|z|=1} \frac{dz}{z\, t_{\text{L}} t_{\text{R}}}. \tag{6.46}$$

The integral over $z$ only picks up residues at $z = z_{\text{L}}^+$ and $z_{\text{R}}^+$ (the integrand is regular at $z = 0$ because of the absence of contact terms) and gives

$$\text{Im}\,\mathcal{I}_{\text{box}}(s, t) = -2 s^{\frac{D-6}{2}} \int_0^1 \frac{dr\, r^{\frac{D-6}{2}}}{t + ru}. \tag{6.47}$$

Recall we work in the $s$-channel with $-s < t < 0$, where $r \to 1$ corresponded to vanishing of the Gram determinant (where the loop momentum $\ell$ becomes collinear with at least one of the external momenta) and $r \to 0$ is the region dominating at the threshold $s = 0$.

The analytic properties of the imaginary part of the box diagram are obvious in this representation: as $s \to 0$, the integral develops a logarithmic singularity coming from the region $r \to 1$, while in the limit $t \to 0$, the integrand behaves as $\sim r^{\frac{D-8}{2}}$, giving a divergence in $D \leqslant 6$. Moreover, even at finite $(s, t)$ there is a soft-collinear divergence in $D \leqslant 4$ coming from the region $r \to 0$. Finally, branch cuts of the integral appear when the root of the denominator at $r = -t/u$ approaches the integration contour (we already discussed the endpoint singularities). This actually gives a straightforward way of evaluating the double discontinuity, where we only need to study the difference between the integration contour evaluated at $t \pm i\varepsilon$ as $\varepsilon \to 0^+$, giving

$$\text{Disc}_t \text{Disc}_s \mathcal{I}_{\text{box}}(s, t) = i s^{\frac{D-6}{2}} \oint_{r = -t/u} \frac{dr\, r^{\frac{D-6}{2}}}{t + ru} = -\frac{2\pi}{u} \left( -\frac{st}{u} \right)^{\frac{D-6}{2}}, \tag{6.48}$$

in the region $s, t > 0$. Similar logic can be applied to any other diagram in field theory.

At any rate, the expression (6.47) can be easily integrated and gives

$$\text{Im}\,\mathcal{I}_{\text{box}}(s, t) = -\frac{4 s^{\frac{D-6}{2}}}{t\,(D-4)}\, {}_2F_1(1, \tfrac{D-4}{2}, \tfrac{D-2}{2}, -\tfrac{u}{t}), \tag{6.49}$$

in agreement with the standard results, see, e.g., [85, App. B.4]. For us, the most interesting case is D = 10, where we have

$$\text{Im}\,\mathcal{I}_{\text{box}}(s, t)\big|_{\text{D}=10} = -\frac{s^2}{u^3} \left( 2t^2 \log\left( -\tfrac{s}{t} \right) - (s + 3t)u \right). \tag{6.50}$$

This result will reappear multiple times in the leading $\alpha'$-expansion of string amplitudes and will be called simply $\text{Im}\,\mathcal{I}_{\text{box}}(s, t)$.

### 6.7.2 Open string

Let us now apply the above technique to the low-energy expansion of open string amplitudes. We only spell out the contribution to $\mathrm{tr}(t^{a_1}t^{a_2}t^{a_3}t^{a_4})$ since the remaining contributions can be obtained without any difficulty, but would be lengthy to display. In the conventions of (4.76) we have

$$
\begin{aligned}
\mathrm{Im}\,\mathcal{A}_{\mathrm{I}} = {}& \pi^2 g_s^4 t_8 \mathrm{tr}(t^{a_1}t^{a_2}t^{a_3}t^{a_4})\Bigg[ \frac{\alpha'\,\mathrm{Im}[(N{-}4)\mathcal{I}_{\mathrm{box}}(s,t)-2\mathcal{I}_{\mathrm{box}}(s,u)]}{120} \\
&+ \frac{\zeta_2}{180}\alpha'^3(N{-}3)s^3 + \frac{\zeta_3}{1260}\alpha'^4 s^3\big((4N{-}22)s + (N{-}2)t\big) \\
&+ \frac{\zeta_2^2}{50400}\alpha'^5 s^3\big(2(92N{-}219)s^2 + (15{-}8N)st + (4N{-}9)t^2\big) \\
&+ \frac{\zeta_5}{15120}\alpha'^6 s^3\big((38N{-}208)s^3 + 6(2N{-}5)s^2 t + 3(N{-}4)st^2 + (N{-}2)t^3\big) \\
&+ \frac{\zeta_2\zeta_3}{5040}\alpha'^6 s^4\big(12(N{-}3)s^2 + t((N{-}2)u + t) + st\big) \\
&+ \frac{\zeta_3^2}{30240}\alpha'^7 s^4\big(4(5N{-}28)s^3 + 2(N{+}1)s^2 t - 3(N{-}4)st^2 - (N{-}2)t^3\big) \\
&+ \frac{\zeta_2^3}{5292000}\alpha'^7 s^3\big(70(176N{-}383)s^4 + 25(9{-}11N)s^3 t + 3(119N{-}347)s^2 t^2 \\
&\qquad\qquad + 4(17{-}9N)st^3 + 2(16N{-}33)t^4\big) \\
&+ \frac{\zeta_7}{831600}\alpha'^8 s^3\big(20(83N{-}452)s^5 + 5(129N{-}368)s^4 t + 2(148N{-}593)s^3 t^2 \\
&\qquad\qquad + (137N{-}362)s^2 t^3 + 4(9N{-}29)st^4 + 10(N{-}2)t^5\big) \\
&+ \frac{\zeta_2^2\zeta_3}{756000}\alpha'^8 s^4\big(60(20N{-}47)s^4 + 5(66{-}23N)s^3 t + 6(33{-}8N)s^2 t^2 \\
&\qquad\qquad - (N{-}6)st^3 + 2(9{-}4N)t^4\big) \\
&+ \frac{\zeta_2\zeta_5}{37800}(N{-}3)\alpha'^8 s^4\big(70s^4 - 5s^3 t - 6s^2 t^2 - 2st^3 - t^4\big) + \mathcal{O}(\alpha'^9)\Bigg] \\
&+ \text{non-planar contributions}\,.
\end{aligned}
\tag{6.51}
$$

We checked that the planar annulus contribution (coefficient of $N\mathrm{tr}(t^{a_1}t^{a_2}t^{a_3}t^{a_4})$) agrees with the $\alpha'$-expansion given in [34, Section 7.1].

### 6.7.3 Closed string

For closed-string type II amplitude in the conventions of (4.95), the $\alpha'$-expansion is given by

$$
\begin{aligned}
\mathrm{Im}\,\mathcal{A}_{\mathrm{II}} = {}& \frac{\pi^5 g_s^4 t_8 \tilde{t}_8}{8}\Bigg[ \frac{\alpha'\,\mathrm{Im}[\mathcal{I}_{\mathrm{box}}(s,t)+\mathcal{I}_{\mathrm{box}}(s,u)]}{60} + \frac{\zeta_3}{45}\alpha'^4 s^4 + \frac{\zeta_5}{1260}\alpha'^6 s^4(22s^2 - tu) \\
&+ \frac{\zeta_3^2}{1260}\alpha'^7 s^5(12s^2 + tu) + \frac{\zeta_7}{18900}\alpha'^8 s^4\big(260s^4 - 25s^2 tu + 2t^2 u^2\big) \\
&+ \frac{\zeta_3\zeta_5}{4725}\alpha'^9 s^5\big(70s^4 + 5s^2 tu - t^2 u^2\big) + \mathcal{O}(\alpha'^{10})\Bigg],
\end{aligned}
\tag{6.52}
$$

which agrees with the previous computations for the low-energy expansion of the non-analytic terms of superstring amplitudes [33, Theorem 4.1] spelled out in [34, Section 7.1], see also [65, 66, 86] for prior work on $\alpha'$-expansion.

# 7 Conclusion

In this paper we found a representation of the imaginary part of string one-loop amplitudes that is exact in $\alpha'$. We derived both from unitarity cuts and directly from the worldsheet. We used this representation to investigate various physical properties of genus-one string amplitudes. There are a few interesting points to mention.

**Regge limit.** We found a tension with the usual claim of Regge growth in the amplitude (see, e.g., [87]) that predicts that $A_{\text{an}}^{\text{p}} \sim s^{-1+t} \log(s)$, whereas the computed imaginary part in the domain $0 < s \lesssim 14$ is roughly $s^t$, see eq. (6.18) for a good empirical approximation (recall that additional powers of $s$ come after reinstating $t_8 \sim s^2$). This indicates reading off the Regge asymptotics from the numerics for $s \lesssim 14$ is far too naive.

While we could not reach energies high enough to directly verify the Regge limit numerically, one reason indicating that the behavior at $s \lesssim 14$ has not reached its asymptotics is as follows. We can look at the contributions to $\text{Im} A_{\text{an}}^{\text{p}}$ from each individual threshold $(n_1, n_2)$. Since we know the expressions for those quantities for any $s$, we can compute that (setting $t = 0$ for simplicity) they asymptote to, for the values we have computed,

$$\text{Im} A_{\text{an},0,0}^{\text{p}} \sim \frac{\pi^2}{128} s^{-1} \log^{-4}(s), \tag{7.1a}$$

$$\text{Im} A_{\text{an},n_1,0}^{\text{p}} \sim \frac{\pi^2}{16 n_1} s^{-1} \log^{-5}(s), \quad 1 \leqslant n_1 \leqslant 6, \tag{7.1b}$$

$$\text{Im} A_{\text{an},1,1}^{\text{p}} \sim \frac{\pi^2}{64} s^{-1} \log^{-5}(s), \tag{7.1c}$$

$$\text{Im} A_{\text{an},2,1}^{\text{p}} \sim \frac{\pi^2}{32} s^{-1} \log^{-5}(s). \tag{7.1d}$$

We hence conjecture that in general $\text{Im} A_{\text{an},n_1,n_2}^{\text{p}} \sim \text{const.} \times s^{-1} \log^{-5}(s)$ for some constant depending on $n_1$ and $n_2$ (for all $(n_1, n_2) \neq (0,0)$). The corrections to the above behavior are suppressed by further powers of $\log^{-1}(s)$, meaning that one would have to go to extremely large values of $s$ for each approximation to become accurate. Hence each of the individual terms, and conceivably their sum, is consistent with the Regge behaviour. This is also indicated from the fact that the empirical estimate (6.18) becomes less and less accurate with larger $n$, cf. Appendix A, and hence it should not be used to directly extrapolate to the Regge limit. We plan to return to the question of Regge asymptotics using more direct saddle-point methods on the moduli space in the future.

**Low-spin dominance.** We also observed a small tension with the lore of low-spin dominance [35, 36]. Expanding the imaginary part of the amplitude into its partial-wave coefficients shows that the amplitude (stripped of the polarization prefactor $t_8$) is dominated by scalar exchange for $s < 1$, but the partial waves of spins $\lesssim j-1$ are of comparable size for $s \sim j$. It would be interesting to further study this effect in string theory at higher genera.

**Possible generalizations.** Our discussion of the various types of string theories and their diagrams was not exhaustive. We could have considered heterotic strings, the Klein bottle diagram, etc. We expect that the corresponding calculations are relatively straightforward to carry out. One can also consider the string on a compactified background, such as $\mathbb{R}^{1,3} \times CY_3$. While the massless tree-level amplitude in four dimensions is blind to the compactification, the one-loop amplitude is sensitive to it. In particular, the particle spectrum in four dimensions

detects the geometry of the internal manifold and correspondingly the cutting rules will include the additional particles from the compactification.

There are many possible generalizations and directions to explore for the future. First of all, it would of course be interesting to understand these computations more systematically at higher genus. As we mentioned in Section 3, we can always compute the imaginary part of an amplitude by a special contour winding around the non-separating divisor in the complexification $\mathcal{M}^{\mathbb{C}}_{g,n}$ (or $\mathfrak{M}_{g,n_{\mathrm{NS}},n_{\mathrm{NR}}} \times \mathfrak{M}_{g,n_{\mathrm{NS}},n_{\mathrm{NR}}}$) and there should be a more abstract argument that shows that it reproduces the expected holomorphic cutting rules.

**Holomorphic cuts.** Just as in field theory, in order to prove unitarity using contour deformation arguments, it is the easiest to turn to holomorphic (as opposed to Cutkosky) cutting rules [49]. To understand this statement, recall that the $S$-matrix can be decomposed as $S = \mathbb{1} + iT$, so unitarity $SS^{\dagger} = \mathbb{1}$ implies $\mathrm{Im}\, T = \frac{1}{2} T T^{\dagger}$ for identical external states. Crucially, the right-hand side of this expression involves the dagger operator and hence is not purely holomorphic. This incarnation of unitarity is difficult to see from the moduli-space perspective, where we deal with purely holomorphic contour deformations. To fix this, one instead solves iteratively for $T^{\dagger}$ in terms of $T$, leaving us with the holomorphic version of unitarity:

$$\mathrm{Im}\, T = -\tfrac{1}{2} \sum_{c=1}^{\infty} (-iT)^{c+1}. \tag{7.2}$$

The price we need to pay is that we have to include more and more cuts at higher and higher genus. Recall that each pairwise contraction of $T$-matrix elements instructs us to sum over all the intermediate on-shell states and integrate over their phase space. In particular, this means that for a given term to contribute at all, it needs to consist of physically-allowed subprocesses only.

For example, in the case of the genus-one four-point amplitude of massless external states we have, schematically,

$$\mathrm{Im}\, \mathcal{A}_1^{12 \to 34} = \tfrac{1}{2} \sum_{56} \mathcal{A}_0^{12 \to 56} \mathcal{A}_0^{56 \to 34}, \tag{7.3}$$

where $\mathcal{A}_g^{\mathrm{in} \to \mathrm{out}}$ denotes the genus-$g$ amplitude and the sum-integral goes over all species, polarizations, colors, degeneracies, etc. of the intermediate states 5 and 6 (we ignored one-particle exchanges that give rise to delta-function terms only). Note that the lack of conjugation of $\mathcal{A}_g^{56 \to 34}$ (in comparison with Cutkosky) is perfectly consistent with the aforementioned fact that no complex-conjugation is needed because no further propagators can go on-shell after the first cut. The simplest situation in which at least two simultaneous holomorphic cuts are needed occurs for the genus-one six-point amplitude of massless states, where

$$\mathrm{Im}\, \mathcal{A}_1^{123 \to 456} = \tfrac{1}{2} \sum_{78} \mathcal{A}_0^{12 \to 78} \mathcal{A}_0^{378 \to 456} + \mathrm{perm}$$
$$- \tfrac{i}{2} \sum_{789} \mathcal{A}_0^{12 \to 78} \mathcal{A}_0^{38 \to 49} \mathcal{A}_0^{79 \to 56} + \mathrm{perm}. \tag{7.4}$$

The first line contains the familiar normal thresholds in all possible channels. In the second line, the triangle anomalous thresholds contribute, where three constituent $2 \to 2$ amplitudes are glued together with two cuts to form the overall $3 \to 3$ process with three on-shell legs. Kinematic support of this term can be determined using the same techniques as in Section 5.1. More complicated unitarity equations occur at higher multiplicity and genus. It would be interesting to prove the existence of these contributions directly from the worldsheet.

**Real part of the amplitude.** Perhaps most intriguing is the possibility to extend the techniques explored in this paper to the real part of the amplitude. The full integration contour admits a Rademacher expansion in terms of infinitely many circles such as those in Figure 1 that allows for a convergent closed form expression of the full amplitude. We plan to return to this problem in the future [16].

# Acknowledgments

We thank Nima Arkani-Hamed, Pinaki Banerjee, Hofie Hannesdottir, Aaron Hillman, Giulio Salvatori, Oliver Schlotterer, and Edward Witten for useful discussions. L.E. and S.M. are supported by the grant DE-SC0009988 from the U.S. Department of Energy. S.M. gratefully acknowledges the funding provided by Frank and Peggy Taplin.

# A  Decay widths

The imaginary part of the first few double residues of the planar annulus amplitude are

$$\operatorname*{DRes}_{s=1} \operatorname{Im} A_{\text{an}}^{\text{p}} = \frac{\pi^2}{420}, \tag{A.1}$$

$$\operatorname*{DRes}_{s=2} \operatorname{Im} A_{\text{an}}^{\text{p}} = \frac{\pi^2(t+1)}{420}, \tag{A.2}$$

$$\operatorname*{DRes}_{s=3} \operatorname{Im} A_{\text{an}}^{\text{p}} = \frac{10883\pi^2(t+1)(t+2)}{8981280}, \tag{A.3}$$

$$\operatorname*{DRes}_{s=4} \operatorname{Im} A_{\text{an}}^{\text{p}} = \frac{17\pi^2(t+2)\left(480201t^2 + 1920804t + 1440704\right)}{19926466560}, \tag{A.4}$$

$$\operatorname*{DRes}_{s=5} \operatorname{Im} A_{\text{an}}^{\text{p}} = \frac{\pi^2}{9729720000000}\Big[ \left(988083963 + 4425000\sqrt{5}\right)t^4$$
$$+ \left(9880839630 + 44250000\sqrt{5}\right)t^3$$
$$+ \left(34536479825 + 175675000\sqrt{5}\right)t^2$$
$$+ \left(49171903750 + 325250000\sqrt{5}\right)t$$
$$+ 23455600625 + 221875000\sqrt{5}\Big]. \tag{A.5}$$

Higher decay widths up to $n = 14$ can be found in the ancillary file `decaywidths.txt`.

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
