# Peer review of "Unitarity Cuts of the Worldsheet"

_SciPost Physics, doi:SciPost Phys. 14, 015 (2023)_

## Round 1 · Referee Report · Anonymous (Referee 1) · 2022-10-5

Report

The manuscript under review offers a variety of exciting results and novel perspectives on four-point genus-one amplitudes for massless states of open and closed superstrings. While the supersymmetry cancellations and integral representations of these superstring amplitudes are known from the early 1980's, the literature of the last 40 years has very little to say about their discontinuity structure or exact evaluations of the integrals. Moreover, numerical evaluations for a given kinematic point (i.e. Mandelstam $s$ and $t$ variables) were completely out of reach before the present work by the divergent integrals in the conventional string-amplitude representation.

The authors present an elegant integral representation for the imaginary parts of four-point superstring amplitudes at genus one -- separately for the torus, Moebius strip and the planar/non-planar cylinder diagram. Thanks to a clever implementation of the Witten $i \epsilon$ prescription, the integrals for the imaginary parts are manifestly convergent and suitable to plot in large kinematic regions beyond the low-energy regime. The authors derive numerous physical properties of the amplitudes from their new integral representation and in many cases perform highly non-trivial cross-checks against older or recent results in the literature. In particular, they give a detailed and beautiful account on how the unitarity properties expected from field-theory intuition are implemented via worldsheet techniques.

I am deeply impressed by both the technical strength and the innovative power of the manuscript: The authors overcome several 40-year old barriers in making genus-one string amplitudes more explicit by their mastery of integration contours in moduli space and a novel worldsheet perspective on unitarity. Their method is full of potential for the challenges of higher-point or higher-genus string amplitudes and will definitely have lasting impact on the field -- I am really looking forward to the follow-up work planned in reference [16]. At the same time, the manuscript is remarkably well written and pedagogically structured. The wealth of new results and methodological progress justifies the length of the paper, and the authors succeeded in making it a motivating read until the end.

Based on the above, I enthusiastically recommend publication of the manuscript in SciPost. Once the authors prepare a version 2, they could take care of the minor points listed below.

Requested changes

  • The inline equation in the middle of page 3 starting with $- \frac{t_8 g_s }{ t}$ (multiplying a beta integral) is lacking an $\alpha'$ in the denominator of its prefactor for consistency with (1.1), thought the authors may want to multiply both by $(\alpha')^2$ to get a smooth limit as $\alpha'\rightarrow 0$.

  • The notation for the gauge-group generators accidentally changed from $t^a$ to $T^a$ in (3.6) and (3.7), please harmonize

  • One of the equation references to (3.6) and (3.7) in the 2nd line below (3.12) went wrong

  • In the 2nd line of (3.72), one of the $\Gamma(-t_{\rm L})$ in the numerator should be a $\Gamma(-t_{\rm R})$

  • The short explanation that $\varepsilon$ denotes an 8th root of unity is doubled below (4.36) and (4.38); it is probably simplest to erase the first 5/4 lines below (4.38)

  • One of the first two equations in (4.83) should probably refer to $y_{42}$ rather than $y_{41}$

---

## Round 1 · Referee Report · Anonymous (Referee 2) · 2022-10-6

Report

In this paper, the authors undertake a systematic investigation of how to implement unitarity in higher-genus string theory scattering amplitudes. At genus zero, all degenerations of the string worldsheet are controlled by collisions of vertex operator insertions, which are directly linked with on-shell poles of the amplitude, but the presence of non-separating degenerations (i.e., pinching a cycle) significantly complicates matters at higher genus. Furthermore, the moduli integrals arising from worldsheet correlation functions are generally divergent; while tricks like analytic continuation can easily deal with these problems at genus zero, the situation at higher-genus is much more difficult.

Building on Witten's idea that these issues could be resolved by analytically continuing to Lorentzian signature on the worldsheet in a neighbourhood of boundary divisors in moduli space, the authors give a comprehensive and practical treatment of the imaginary part of higher genus string amplitudes. Extremely detailed calculations are provided for four-point, genus one amplitudes, but it is made clear how the formalism should extend to more general higher-genus amplitudes. Heuristically, the main result is a stringy version of the Cutkosky rules, expressing the imaginary part in terms of a sum over exchange thresholds of the twice-cut amplitude. In practical terms, this provides a manifest convergent expression for the imaginary part of the amplitude which can be evaluated numerically. The authors observe many remarkable properties from this result, including a 'low-spin dominance' phenomenon, where the imaginary part is well-captured at each threshold by the partial waves of lower spin.

The paper is phenomenally well-written, and the early sections in particular manage to make a subject with a long, somewhat convoluted history easily accessible to non-experts. This result will be of broad interest to the scattering amplitudes community, and represents a breakthrough in understanding how to implement unitarity in perturbative string theory. This certainly merits publication in SciPost Physics. I spotted some very minor typos (in addition to those pointed out by the other referee), and one small suggestion which the authors may want to consider prior to publication.

Requested changes

(1.) Page 3, 3rd paragraph: "...so much more appealing that, say,..." should have "that" --> "than".
(2.) Page 3, 3rd paragraph: "Gamma functions have fast-convergent sum representation[s]..."
(3.) Page 4, 3rd line: "...metrics are use in the [first] place..."
(4.) Page 4: When the authors are talking about tree-level and low-energy being very forgiving with respect to analytic continuation, they say that, "tree-level only features meromorphic functions." Perhaps it would be helpful to clarify that what is meant is that at tree-level/genus zero, the worldsheet correlation function features only meromorphic functions of the moduli. I slightly worry that a non-expert reader could get confused and think that this is referring to the correlator as a function of the kinematics, for which it is certainly not a rational function at genus zero (the kinematics entering through Koba-Nielsen factors). It's a minor point, I know, but the introduction is so wonderfully accessible and this was the only place where I felt a bit of missing clarity.
(5.) Page 37, 1st line: "...the terms divergent slower than..." should have "divergent" --> "diverge".

---

## Round 2 · Author Response

We would like to thank the two referees for their careful reading of the paper and their remarks. We addressed all their comments, see the list of changes below.

---

## Round 2 · List of Changes

All of the requested changes of referee 1 were typos and we fixed them according to their suggestions. Similarly, point (1.), (2.), (3.) and (5.) of referee 2 are typos and we fixed them accordingly. Regarding point (4.) of referee 2, we agree that this can be a bit misleading and have reformulated it accordingly. The relevant sentence now reads
``Tree-level amplitudes only feature meromorphic functions of the kinematics (with isolated poles corresponding to propagators going on-shell), while in the low-energy limit the answers can be usually matched with the field-theory intuition for placement of branch cuts.''

---

## Editorial Decision

published